# EvoControl: Multi-Frequency Bi-Level Control for High-Frequency Continuous Control

**Samuel Holt** [† 1]  **Todor Davchev** [2]  **Dhruva Tirumala** [2]  **Ben Moran** [2]  **Atil Iscen** [2]  **Antoine Laurens** [2]  **Yixin Lin** [2]  **Erik Frey** [2]  **Markus Wulfmeier** [‡ 2]  **Francesco Romano** [‡ 2]  **Nicolas Heess** [2]

## Abstract

High-frequency control in continuous action and state spaces is essential for practical applications in the physical world. Directly applying end-to-end reinforcement learning to high-frequency control tasks struggles with assigning credit to actions across long temporal horizons, compounded by the difficulty of efficient exploration. The alternative, learning low-frequency policies that guide higher-frequency controllers (e.g., proportional-derivative (PD) controllers), can result in a limited total expressiveness of the combined control system, hindering overall performance. We introduce *EvoControl*, a novel bi-level policy learning framework for learning both a slow high-level policy (using PPO) and a fast low-level policy (using Evolution Strategies) for solving continuous control tasks. Learning with Evolution Strategies for the lower-policy allows robust learning for long horizons that crucially arise when operating at higher frequencies. This enables *EvoControl* to learn to control interactions at a high frequency, benefitting from more efficient exploration and credit assignment than direct high-frequency torque control without the need to hand-tune PD parameters. We empirically demonstrate that *EvoControl* can achieve a higher evaluation reward for continuous-control tasks compared to existing approaches, specifically excelling in tasks where high-frequency control is needed, such as those requiring safety-critical fast reactions.

[†]Work done during an internship at Google DeepMind. [‡]Equal advising. [1]University of Cambridge [2]Google DeepMind. Correspondence to: Samuel Holt <sih31@cam.ac.uk>.

## 1. Introduction

High-frequency control is paramount for ensuring the safety and reliability of real-world robotic systems (Hogan, 1984; Oh et al., 2014; Venkataraman & Gulati, 1993; Dantec et al., 2022). Failures to respond in real-time to unexpected collisions, disturbances, or human interactions can lead to catastrophic consequences in safety-critical applications such as surgery (Kuchenbecker et al., 2010), autonomous driving (Guo et al., 2019), and industrial automation (Vasic & Billard, 2013). To achieve reliable performance, practical robotic systems commonly rely on low-level fixed controllers (e.g., proportional-derivative (PD) controllers) operating at high frequencies. However, this approach raises the fundamental question of whether we can remove these lower-level fixed controllers entirely and instead directly control motor torques at high frequency, potentially enabling faster reactions and finer control.

Recent work falls into two main categories. First, directly learning end-to-end high-frequency (often torque-based (Aljalbout et al., 2024)) policies, labeled *direct torque control*, presents significant learning challenges as this increases the number of state transitions within a fixed time window, resulting in longer trajectories with complex temporal dependencies that hinder exploration and credit assignment (Peng & Van De Panne, 2017; Sutton & Barto, 2018; Dabney et al., 2020)—and can often lead to suboptimal behavior (Section 5.2).

Second, a common alternative is to learn a slower high-level policy that outputs target positions or velocities, which are then tracked by a faster (higher-frequency) low-level fixed controller, such as a PD controller (Wei, 2019). This bi-level approach, labeled as *fixed controllers*, is prevalent in real-world continuous control tasks (Aljalbout et al., 2024; Song et al., 2019) and simplifies learning by reducing the effective time horizon of the high-level policy (Sutton et al., 1999). For real-time control, this composition affords the high-level policy larger inference time, key for handling rich observations, such as images (i.e., at 30Hz) or using larger networks, such as Visual-Language-Models (Ma et al., 2024). While composing with a low-level PD controller that operates at higher-frequencies $\sim 500 Hz$ on robotic plat-

forms, often only observing at high-frequency direct robot proprioceptive observations such as robot joint positions, velocities and torques (Borase et al., 2021). However, *fixed controllers* are unable to produce fast interaction behavior beyond simple state-goal-tracking limiting their expressiveness, and require manual tuning of their PD parameters for each task.

An effective method for high-frequency control therefore aims to have the following three core properties:
**(P1) Efficient Exploration:** Throughout learning, be able to efficiently explore the state space as well as a high-level policy with a *fixed controller*.
**(P2) High-Frequency Interaction Control:** Enable the learning of a low-level controller capable of complex, adaptive behaviors at high frequencies.
**(P3) Automate Controller Tuning:** Reduce manual tuning of the low-level PD controller parameters.

With these considerations, we introduce *EvoControl*, a novel bi-level policy learning framework for learning both a slow high-level policy and a fast low-level policy for continuous control robotic tasks. EvoControl learns a high-level policy with proximal policy optimization (PPO) and a low-level proprioceptive policy with Evolution Strategies (ES) in alternating stages. The low-level policy is initialized as a fixed PD controller and gradually transitions to a learned neural network policy throughout training via an annealing parameter $\alpha$. This staged training process enables stable learning and effective exploration. ES at the low level provides robust learning, particularly beneficial for long horizon credit assignment inherent in high-frequency control. These components enable EvoControl to learn high-frequency interaction control, automate controller tuning, and more efficiently explore than direct high-frequency torque control.

**Contributions:** ① We introduce *EvoControl*, a novel bi-level policy learning framework for learning both a slow (e.g. 30Hz) high-level policy and a fast (e.g. 500Hz) low-level robot proprioceptive controller using PPO and ES, respectively, for continuous-control tasks (Section 3). ② Theoretically, we show that there exist some continuous-time Markov Decision Processes (CTMDPs) in which acting at higher frequencies can yield strictly higher expected cumulative reward (Proposition 2.1). Empirically, we demonstrate that EvoControl can achieve a higher evaluation reward for standard continuous-control tasks at high frequency compared to existing approaches, excelling in tasks where high-frequency control is needed, such as in safety-critical applications of unmodeled interactions (Section 5.1). ③ We gain insight and understanding of how EvoControl can achieve efficient exploration compared to direct torque control at high-frequency, learn fast interactions, and demonstrate robustness to mistuned PD parameter settings.

## 2. Problem

We follow the standard continuous control reinforcement learning (RL) setting with the inclusion of an optional low-level controller.

**States & Actions.** We denote the environments state space as $\mathcal{S} \subset \mathbb{R}^{d_s}$ and its action space as $\mathcal{U} \subset \mathbb{R}^{d_u}$. At time $t \in \mathbb{R}$, the system's state is represented by $\mathbf{s}_t \in \mathcal{S}$, and its action by $\mathbf{u}_t \in \mathcal{U}$. Considering action (e.g., actuator) limits, the action space is constrained to a box in Euclidean space: $\mathcal{U} = [\mathbf{u}_{\min}, \mathbf{u}_{\max}]$.

**Environment Dynamics.** The transition dynamics for continuous control environments can be described by an underlying unknown *differential equation* of $\dot{s}_t = \frac{ds_t}{dt} = f(s_t, u_t)$. The transition function, which describes the evolution of the state over a discrete time step $\Delta_t$, can be approximated using the Euler method $s_{t+\Delta_t} \approx s_t + \Delta_t f(s_t, u_t)$. Given an action $u_t$ and current state $s_t$, $s_{t+\Delta_t} \sim P(s_{t+\Delta_t}|s_t, u_t)$ is implicitly defined by this approximation. More sophisticated numerical integration methods can also be used. We expand on the problem setup in Appendix A.

**Policies.** The agent can be represented as a single policy $\pi : \mathbb{R}^{d_s} \to \mathbb{R}^{d_u}$, that observes the current observation at time $t$ and samples an action $u_t \sim \pi(s_t)$ and then applies this action to the environment at a given fixed $\Delta_t$. To formalize

---

**Algorithm 1** Bi-Level Policy Interaction (Single High-Level Step)

---

1: $a_k \sim \rho(s_k)$ {High-level action}
2: **for** $i = 0$ to $G - 1$ **do**
3:      $u_{k+i} \sim \beta(s_{k+i}, a_k)$ {Low-level action}
4:      $s_{k+i+1} \sim f(s_{k+i}, u_{k+i}, \Delta_t)$
5: **end for**

---

a bi-level policy, we decompose $\pi$ into two components: a slow, high-level policy, $\rho$, and a fast, low-level policy, $\beta$. Both policies interact with the environment as described in Algorithm 1. At time step $k$, $\rho$ outputs a latent action $a_k \sim \rho(s_k)$ (e.g., a target position or velocity). Operating at a higher frequency, $\beta$ receives $a_k$ and generates low-level actions $u_{k+i}$ (e.g., motor torques) at a finer-grained time step $i$ to achieve the target specified by $\rho$. With $\beta$ operating at frequency $1/\Delta_t$, $\rho$'s latent action is executed by $\beta$ for $G$ steps, making $\rho$'s effective frequency $1/(G\Delta_t)$.

**Objective.** The environment produces a reward $r_t$ sampled from an unknown reward function $r(s_t, u_t)$. The overall objective of the agent is to maximize the expected future discounted reward $\mathbb{E}_{s_{0:T}, u_{0:T-1}, r_{0:T-1}} \left[ \sum_{i=0}^{T-1} \gamma^i r_i \right]$, where $0 \leq \gamma < 1$ is the discount factor.

## 2.1. Existence of CTMDPs Requiring Higher-Frequency Control for Optimality

In certain continuous-time Markov Decision Processes (CT-MDPs), taking actions at a higher frequency strictly increases the achievable reward—a point we make formally in Proposition 2.1.

**Proposition 2.1.** *Consider a continuous-time Markov Decision Process (CTMDP) with a finite state space and action space. Let this CTMDP be discretized with time step $\Delta_t$. There exist CTMDPs such that for any fixed discretization step $\Delta_t > 0$, there exists a finer discretization $\Delta_t' < \Delta_t$ where the optimal policy for the $\Delta_t'$-discretized MDP achieves a higher expected cumulative reward over a fixed time horizon $T$ than the optimal policy for the $\Delta_t$-discretized MDP.*

*Proof.* Full proof is in Appendix B; however, we present the following sketch. Consider a CTMDP with states $\{s_{\text{good}}, s_{\text{bad}}\}$ and actions $\{a_{\text{maintain}}, a_{\text{recover}}\}$. Under $a_{\text{maintain}}$, the system transitions from $s_{\text{good}}$ to $s_{\text{bad}}$ at rate $p > 0$, while $a_{\text{recover}}$ immediately transitions $s_{\text{bad}}$ to $s_{\text{good}}$. Let rewards be $r(s_{\text{good}}) = r_{\text{good}} > 0$ and $r(s_{\text{bad}}) = r_{\text{bad}} < 0$, with episode length $T$. Discretizing at $\Delta_t$, the optimal policy uses $a_{\text{maintain}}$ when in $s_{\text{good}}$ and $a_{\text{recover}}$ when in $s_{\text{bad}}$, yielding expected return $R(\Delta_t) = T\, r_{\text{good}} - T\, p\, \Delta_t \big(r_{\text{good}} - r_{\text{bad}}\big)$. Now refine to $\Delta_t' = \Delta_t/n$ where $(n > 1)$. Recovery happens more quickly, reducing time spent in $s_{\text{bad}}$ each $\Delta_t$-interval, so the finer-discretized return is $R(\Delta_t') = T\, r_{\text{good}} - \frac{T\, p\, \Delta_t}{n}\big(r_{\text{good}} - r_{\text{bad}}\big)$. The difference $R(\Delta_t') - R(\Delta_t) = T\, p\, \Delta_t \big(r_{\text{good}} - r_{\text{bad}}\big)\big(1 - \frac{1}{n}\big) > 0$, so a finer discretization leads to a strictly higher expected cumulative reward under its optimal policy. $\square$

This highlights the need for higher-frequency control in certain environments. This is analogous to the Pulse Width Modulation (PWM) sampling theorem (Huang et al., 2011), where variable pulse widths enable perfect signal reconstruction from discrete samples, similar to how high-frequency actions enable optimal control in our MDP setting. We further explore a continuous control safety-critical intuitive example in Appendix B.1.

### 2.2. Background: Fixed PD Controllers

In continuous control and robotics, hierarchical structures composed of a learned high-level policy ($\rho$) and a fixed low-level controller (e.g., a PD controller (Oku et al., 2018; Find-eisen et al., 1980)) are common. This hierarchical decomposition reduces the number of decision steps for the high-level policy by a factor of $G$ within a fixed episode duration $T$, where $G$ is the number of low-level actions executed per high-level action. The high-level policy outputs a target $a_k$,

Table 1: Common Fixed Low-Level PD Controllers

| Method | $a_k$ | Control Law |
|---|---|---|
| PD Absolute Position | $q^d = a_k$ | $\tau(t) = K_p(q^d - q) + K_d(\dot{q}^d - \dot{q})$ |
| PD Delta Position | $\delta q^d = a_k$ | $\tau(t) = K_p((q_k + \delta q^d) - q) + K_d(\dot{q}^d - \dot{q})$ |
| PD Velocity | $\dot{q}^d = a_k$ | $\tau(t) = K_p(\dot{q}^d - \dot{q}) + K_d(0 - \ddot{q})$ |
| PD Integrated Velocity | $\dot{q}^d = a_k$ | $\tau(t) = K_p((q^d + \int \dot{q}^d dt) - q) + K_d(0 - \dot{q})$ |
| PD Position & $K_p$ | $\{q^d, K_p\} = a_k$ | $\tau(t) = K_p(q^d - q) + K_d(\dot{q}^d - \dot{q})$ |

often a desired position or velocity[1], which the low-level PD controller then tracks. The controller computes a control signal $u_t$ based on the error between the target $a_k$ and the measured system state $s_t$, $u_t = K_p(a_k - s_t) + K_d(\dot{a}_k - \dot{s}_t)$ where $e_t = a_k - s_t$ is the tracking error, and $K_p, K_d \in \mathbb{R}^+$ are constant proportional and derivative gains. Common PD controller designs using proprioceptive states (joint positions $q_t$, velocities $\dot{q}_t$, and torques $\tau_t$) are summarized in Table 1. We provide an expanded background on PD controllers in Appendix C.

## 3. EvoControl: Evolved Low-Level Controller Framework

We now propose *EvoControl*, a novel bi-level policy learning framework for learning both a slow high-level policy and a fast low-level policy for continuous control tasks. The key idea is to stabilize the bi-level on-policy learning of a higher-level policy by initially learning with a fixed-low-level PD controller and then annealing to a gradually ES-learned high-frequency controller—Figure 1 provides a block diagram.

First, we formulate the bi-level policy optimization problem and its challenges. Then, we discuss our approach and the advantages of using ES for lower-level policy optimization. Specifically our framework can be applied starting with different semantically meaningful high-level actions $a_k$ from the high-level policy, such as position/velocity targets, commonly seen in existing PD controllers, as outlined in Table 1.

### 3.1. Promise and Challenges of Policy Hierarchies

Hierarchical reinforcement learning (HRL), employs multiple levels of policies with increasing temporal abstraction. It tackles complex tasks through improved exploration and long-horizon planning (Parr & Russell, 1997; Sutton et al., 1999; 2011). If high-level actions induce diverse low-level trajectories, exploration can become more effective (Sutton et al., 1999; Li et al., 2021; McClinton et al., 2021). For instance, fixed low-level PD controllers, combined with a learned high-level policy, act as temporally extended actions, facilitating efficient exploration if aligned with the

---

[1]The state $s_t$ can encompass a wide range of proprioceptive information beyond joint positions ($q_t$). We present the target $a_k$ and tracking error in terms of position/velocity to align with common PD controller formulations.

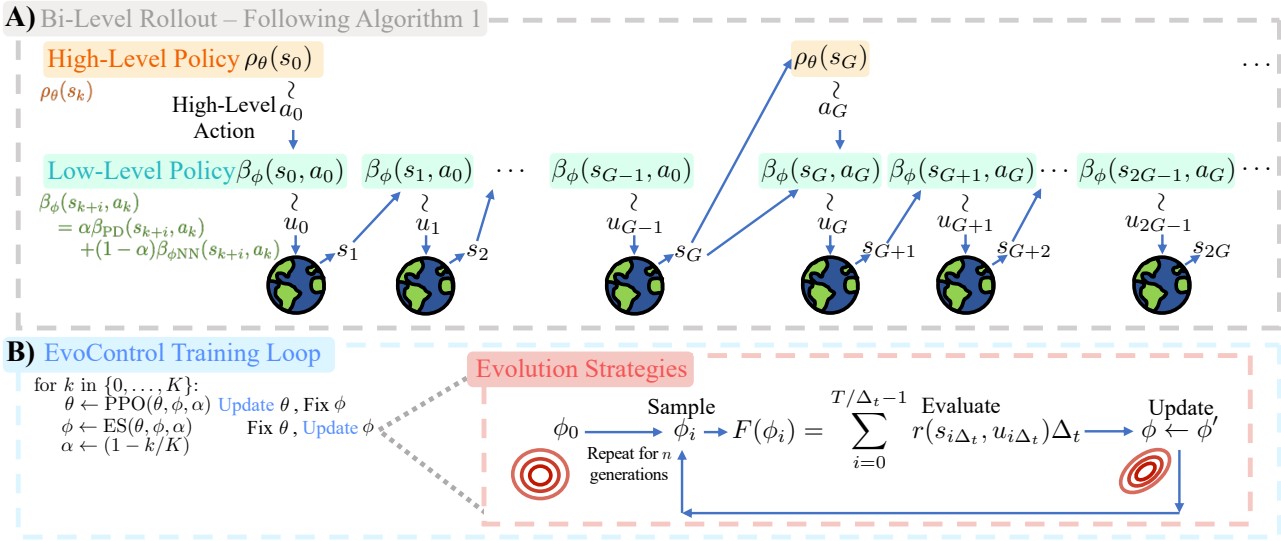

Figure 1: **A) Bi-level Policy Interaction.** The high-level policy $\rho_\theta$ outputs latent action $a_k$, which guides the low-level policy $\beta_\phi$ for $G$ steps. **B) EvoControl Training Loop.** EvoControl trains both the high-level ($\rho_\theta$, with parameters $\theta$) and low-level ($\beta_\phi$, with parameters $\phi$) policies over the course of training divided into $K$ discrete sections. Each section $k$ first optimizes $\theta$ with PPO (fixed $\phi$), then optimizes $\phi$ with Evolution Strategies (fixed $\theta$). ES maintains a population of $\phi$ parameters, evaluates their fitness (episodic return, $F$), and updates the parameter distribution to maximize average fitness. This process, robust to long horizons, enables learning of complex low-level behaviors. To stabilize learning, $\beta_\phi$ is initially a PD controller ($\beta_{\text{PD}}$, $\alpha = 1$) and transitions to a learned controller as $\alpha$ anneals towards 0.

task[2] (Chiaverini et al., 1999).

However, simultaneously learning high- and low-level policies with an intermediate latent action can destabilize learning (Yang et al., 2021; Nachum et al., 2018a; Wöhlke et al., 2021). As the low-level policy updates, the same high-level latent action may produce different low-level action sequences, creating a non-stationary learning environment for the high-level policy. Conversely, effective low-level policy learning requires informative high-level latent actions, creating a co-dependency that can hinder both levels of the hierarchy.

Bi-level learning presents further challenges: determining optimal latent actions (Nachum et al., 2018a) and the appropriate low-level policy reward function to optimize. While subgoal-based rewards (e.g., $r_t = -||s_{\text{goal}} - s_t||_2$) are promising and relate to PD control, they can struggle to capture complex behaviors best optimized through overall episodic return ($R = \sum_{i=0}^{T/\Delta_t - 1} r(s_{i\Delta_t}, u_{i\Delta_t})\Delta_t$). Furthermore, optimizing $R$ requires long-term credit assignment, exacerbated by high-frequency low-level policies (Peng & Van De Panne, 2017). This motivates our framework for optimizing the episodic return $R$ directly.

---

[2]For example, a PD absolute position controller can serve as an effective prior for goal-reaching tasks, where the low-level control actions $u_t$ correspond to torques applied to the robot's actuators.

### 3.2. Efficient High-Level Policy Exploration

In the following, we outline how we train both the slow high-level policy $\rho$ and the fast low-level policy $\beta$ by training each level in $K \in \mathbb{Z}_+$ stages, with the other fixed, which assists in mitigating the issues of instability and long-horizon credit assignment.

To deliver on the key advantage of efficient high-level state-action exploration for the bi-level control approach, as outlined in Figure 1, we seek a stable way of initially learning the high-level policy to overcome the non-stationary challenge of learning with a continually updating lower-level policy. A key approach we choose, which, as we will see later, also assists in stably learning a lower-level policy, is to represent the lower-level policy as a convex combination of a fixed PD controller $\beta_{\text{PD}}$ and a learned neural network actor policy $\beta_{\phi\text{NN}}$ with parameters $\phi$, and start initially training with only the PD controller, with $\alpha = 1, \alpha \in [0, 1]$ and anneal $\alpha$ to 0 over the course of training. Specifically, we formulate the bi-level policies during training in Algorithm 1 as:

$$\rho_\theta(s_k),$$
$$\beta_\phi(s_{k+i}, a_k) = \alpha\beta_{\text{PD}}(s_{k+i}, a_k) + (1 - \alpha)\beta_{\phi\text{NN}}(s_{k+i}, a_k)$$

This brings two immediate advantages: 1) the high-level action $a_k$ output from the high-level policy has an initial semantic meaning, grounding it and allowing a user to select the most appropriate PD controller for the given task, and 2)

as $\alpha$ is annealed throughout training from 1 to 0, we inherit the effective state-action exploration properties of having a fixed PD controller initially (Section 5.2), and yet can still retain the flexibility of learning more complex lower-level behavior, beyond just sub-goal/state tracking.

To optimize the high-level policy, we employ Proximal Policy Optimization (PPO) (Schulman et al., 2017), a highly effective on-policy reinforcement learning algorithm for continuous control tasks. The high-level policy is represented by a continuous control agent consisting of a neural network with separate critic and actor heads. The actor head parameterizes a multivariate Gaussian distribution with a diagonal covariance matrix, with parameters $\theta$. Specifically, the high-level policy is defined as:

$$a_k \sim \rho_\theta(s_k) = \mathcal{N}(\mu_\theta(s_k), \Sigma_\theta(s_k))$$

where $\mu_\theta(s_k)$ is the mean vector and $\Sigma_\theta(s_k) = \text{diag}(\sigma_{\theta,1}^2(s_k), \sigma_{\theta,2}^2(s_k), ..., \sigma_{\theta,n}^2(s_k))$ is the diagonal covariance matrix, with $\sigma_{\theta,l}^2(s_k)$ representing the variance for the $l$-th action dimension at time step $k$.

### 3.3. ES-learning a Fast Low-level Policy

We seek to learn more complex lower-level behavior beyond prior work of simple goal-reaching low-level policies (Nachum et al., 2018a). However, directly optimizing the episodic return ($R$) is challenging due to the extended credit assignment horizon, exacerbated by the higher-frequency low-level controller and its increased number of steps ($G$) per high-level latent action $a_k$.

Policy gradient methods, while a natural choice for policy optimization, are known to struggle with the long horizons encountered in high-frequency control (Peng & Van De Panne, 2017)[3]. This difficulty is compounded by the credit assignment problem (Sutton & Barto, 2018), where the impact of individual low-level actions on the overall return becomes increasingly diffused over longer horizons (Dayan & Hinton, 1992). Furthermore, the increased temporal density of actions at higher frequencies can lead to heightened sensitivity to policy parameter variations, making the optimization landscapes more challenging to navigate and potentially leading to suboptimal solutions. Empirical evidence supporting these challenges is presented in Appendix J.2.

Instead, we seek a learning method that is invariant to the long-horizon credit assignment issue and that can discover a globally optimal low-level policy. Motivated by this, we adopt an *Evolutionary Strategies (ES)* approach (often termed *Neuroevolution* when applied to training neural

networks) (Rechenberg, 1973; Wierstra et al., 2014; Salimans et al., 2017). ES is a gradient-free, black box, global optimization method that optimizes the lower-level policy neural network parameters $\phi$, by maintaining a population of parameters represented by a distribution $p_\eta(\phi)$, itself parameterized by $\eta$ and maximizes the average fitness value $\mathbb{E}_{\phi \sim p_\eta} F(\phi)$ over the population by searching for $\phi$ with stochastic gradient ascent (Salimans et al., 2017). The core idea rests upon optimizing the score function estimator of

$$\nabla_\phi \mathbb{E}_{\epsilon \sim \mathcal{N}(0,I)}[F(\phi + \sigma\epsilon)] = \frac{1}{\sigma} \mathbb{E}_{\epsilon \sim \mathcal{N}(0,I)}[F(\phi + \sigma\epsilon)\epsilon],$$

where $\mathcal{N}(0, I)$ is the standard multivariate normal distribution, and $\sigma$ is a step size parameter. This estimator allows for gradient estimation without explicit backpropagation by sampling perturbations of $\epsilon$. Therefore, ES maintains a population of parameters, evaluates their fitness, and generates a new population through selection, mutation (adding noise $\sigma\epsilon$), and recombination, as determined by a specific ES algorithm used (Salimans et al., 2017). ES, while less sample efficient compared to RL, are particularly well-suited for scenarios where gradient-based methods struggle, such as those with delayed rewards, noisy environments, or long-horizon tasks (Salimans et al., 2017). The EvoControl framework supports different ES algorithms, and we empirically evaluate many competitive ES approaches in Appendix J.1 and find that the competitive approach of Policy Gradients with Parameter-Based Exploration (PGPE) (Sehnke et al., 2010) is both effective within EvoControl (Section 5.1) and straightforward to implement.

One consideration is what to select as the fitness function $F(\phi)$ for the parameters. Given the bi-level setup, we seek to optimize the episodic return for the combined bi-level policy as a rollout in the environment, i.e. $F(\phi) = R$. Doing so becomes a long horizon optimization problem, especially when the lower-level operates at a higher frequency. Interestingly, directly optimizing the parameters with gradient descent is infeasible due to stochastic noise on the state of the environment and potentially many steps of gradient propagation through an entire bi-level policy rollout of the environment. Crucially, to reduce the variance of the fitness function $F$ and improve learning convergence of the lower-level policy, we sample the mode of the probabilistic high-level actor, and parameterize the lower-level actor as a deterministic low-level policy neural network.

Another advantage of using ES for lower-level policy optimization is the inherent parallelism of fitness evaluations. This parallelism can lead to faster wall-clock time convergence compared to gradient-based methods, even though ES generally requires more samples (Salimans et al., 2017). In our experiments, we find that this trade-off is beneficial: the increased sample complexity is outweighed by the ability to stably learn a high-frequency low-level policy

---

[3]Low-level controllers often operate at 500Hz-1KHz, resulting in $G = 50 - 100$ low-level actions respectively for each high-level action at 10Hz.

(Appendix J.3).

The process of annealing with a PD controller further improves learning a lower-level policy that can be directed with a high-level latent action as input $a_k$ that is directed towards solving the task. Although initially the higher-level policy will provide a latent-action $a_k$ that would be applicable to a particular PD controller that it was initially trained with in the early stages of training, by enabling full optimization of the lower-level policy we can evolve a better performing lower-level policy, lessening the reliance on a tuned PD controller. In practice we find this effective, even if the PD controller is mistuned (Section 5.2). We provide pseudocode for EvoControl in Appendix G.1.

## 4. Related Work

Here we provide the existing approaches to continuous control, and provide an extended related work in Appendix D.

**Fixed Low-Level Controllers:** Commonly involve combining a learned high-level policy (often operating at low frequency, e.g., 10-30Hz) with a fixed or analytical high-frequency low-level controller (e.g., a Proportional-Derivative (PD) controller operating at 500Hz or higher) (Song et al., 2019). Whilst prevalent, due to the ease of state-action space exploration (Peng & Van De Panne, 2017; Pateria et al., 2021), this approach suffers from several drawbacks. The low-level controller's PD parameters require careful tuning per task, and its fixed nature limits its ability to handle high-frequency interactions, such as unexpected collisions or disturbances, that involve more complex behavior than just reaching a given goal state or emergency braking. Furthermore, recent work applying RL algorithms to the physical world often restricts itself to relatively low-frequency control (∼20Hz) due to the reliance on analytical impedance controllers (Martín-Martín et al., 2019; Luo et al., 2018; Johannink et al., 2019; Davchev et al., 2022). EvoControl, in contrast, aims to achieve efficient exploration while also enabling the learning of flexible and complex high-frequency behaviors in the low-level policy.

**Direct Torque Control:** Methods learn end-to-end policies that output joint torques at high frequency (Peng & Van De Panne, 2017; Wahlström et al., 2015; Watter et al., 2015). While offering greater flexibility, this approach suffers from the curse of dimensionality imposed by the increased number of time steps in long horizons. The resulting explosion in the number of possible action sequences significantly hinders exploration and can lead to suboptimal policies (Martín-Martín et al., 2019; Peng & Van De Panne, 2017). EvoControl mitigates this challenge by employing a hierarchical structure, enabling more efficient exploration while retaining the adaptability afforded by direct torque control at the low level.

Table 2: EvoControl Ablation of PD Controllers.

| Controller Variant | $\beta_{\text{NN}}$ Obs. | $\beta_{\text{NN}}$ action |
|---|---|---|
| EvoControl (Full State) | $s_t, a_k, e_t, q_t, \dot{q}_t, t/T$ | $\tau$ |
| EvoControl (Residual State) | $e_t, t/T$ | $\tau$ |
| EvoControl (Target + Proprioceptive) | $a_k, q_t, \dot{q}_t, e_t, t/T$ | $\tau$ |
| EvoControl (Target) | $a_k, q_t, \dot{q}_t, t/T$ | $\tau$ |
| EvoControl (Learned Gains) | $s_t, a_k, q_t, \dot{q}_t, t/T$ | $K_p, K_d$ |
| EvoControl (Delta Position) | $s_t, a_k, e_t, q_t, \dot{q}_t, t/T$ | $\tau$ |

**Hierarchical Reinforcement Learning (HRL):** Methods, including options frameworks (Sutton et al., 1999; Bacon et al., 2017), hierarchical actor-critic architectures (Riedmiller et al., 2018; Vezzani et al., 2022), and recent extensions to effectively train deep policies and critics (Rao et al., 2021; Salter et al., 2022; Wulfmeier et al., 2020a;b), decompose complex tasks into simpler subtasks. While these methods have demonstrated success in improving exploration and learning, they typically focus on discrete skill/subgoal decomposition. Furthermore, HRL subgoal methods often learn simpler lower-level policies limited by subgoal attainment, rather than optimizing overall episode return for complex behavior. EvoControl, inspired by HRL, addresses the challenges of continuous high-frequency control with semantically meaningful exploration. Unlike typical HRL, which focuses on skill discovery, EvoControl targets learning a fast low-level policy that complements the slow high-level policy. Uniquely, EvoControl combines PPO and ES within its hierarchical framework for efficient exploration and complex high-frequency control.

## 5. Experiments and Evaluation

In this section, we evaluate EvoControl and verify that it can achieve a higher evaluation reward for both the same number of high-level policy steps and the same number of low-level enviornment steps compared to the existing training of a high-level policy either with *fixed controllers* or *direct torque control*.

**Benchmark Environments.** We evaluate performance on thirteen high-dimensional continuous control environments. Ten environments are adapted from standard Gym MuJoCo tasks (Brockman et al., 2016a; Freeman et al., 2021), including locomotion (e.g., Ant, HalfCheetah, Humanoid) and manipulation tasks (e.g., Reacher, Pusher). Crucially, we substantially modify these benchmarks by increasing the control frequency to 500Hz (with episodes lasting 1000 steps or 2 seconds of real-time) and removing the typical control-cost term. Typical Gym MuJoCo tasks operate at control frequencies between 12.5–100Hz, whereas our modifications explicitly examine whether directly controlling motor torques at significantly higher frequencies can offer advantages such as faster reactions and finer motor control. Additionally, we introduce two novel safety-critical variants—Reacher and HalfCheetah—with randomly posi-

Table 3: Normalized evaluation returns ($\mathcal{R}$) for benchmarks trained for an equivalent number of 1M high-level ($\rho$) steps per environment. EvoControl consistently outperforms baseline methods (*fixed controllers* and *direct torque control*), with results averaged over 384 random seeds (95% confidence intervals shown). Scores are normalized between 0 (random policy) and 100 (best-performing non-EvoControl baseline). Bold values exceed the best baseline performance (scores >100). Corresponding unnormalized results are provided in Table 28.

| Same PPO high-level alg. $\rho$ with a Low-Level Policy $\beta$ of | Ant $\mathcal{R}\uparrow$ | Halfcheetah $\mathcal{R}\uparrow$ | Hopper $\mathcal{R}\uparrow$ | Humanoid $\mathcal{R}\uparrow$ | Humanoid Standup $\mathcal{R}\uparrow$ | Inverted Double Pend. $\mathcal{R}\uparrow$ | Inverted Pend. $\mathcal{R}\uparrow$ | Pusher $\mathcal{R}\uparrow$ | Reacher $\mathcal{R}\uparrow$ | Reacher 1D $\mathcal{R}\uparrow$ | Walker2D $\mathcal{R}\uparrow$ |
|---|---|---|---|---|---|---|---|---|---|---|---|
| *Fixed Cont. - PD Position* | 100±6.56 | 61.2±0.441 | 88.1±1.18 | 100±2.96 | 100±0.974 | 99.9±0.03 | 100±2.86e-15 | 72.5±6.14 | 100±1.8 | 85.2±2.87 | 61.1±0.51 |
| *Fixed Cont. - PD Position Delta* | 2.4±1.91 | 2.76±0.0888 | 96.2±1.30 | 96.6±1.71 | 2.96±0.0397 | 53.8±1.57 | 100±2.86e-15 | 0.0±0.0 | 40.9±3.23 | 15.2±7.6 | 72.7±0.19 |
| *Fixed Cont. - PD Int. Velocity* | 3.59±1.78 | 2.46±0.0932 | 71.8±0.87 | 83.4±1.13 | 0±0 | 49.7±1.55 | 86.5±2 | 0.0±0.0 | 0±0 | 0±0 | 69.3±2.06 |
| *Fixed Cont. - PD : Position & $K_p$* | 3.55±2.54 | 16.7±0.151 | 100±1.00 | 90.9±1.19 | 29.8±1.42 | 97.5±0.751 | 100±2.86e-15 | 100±6.65 | 50.7±3.9 | 81.8±4.11 | 100±0.24 |
| *Fixed Cont. - Random* | 0.0±0.0 | 0.0±0.0 | 0.0±0.0 | 0.0±0.0 | 0.0±0.0 | 0.0±0.0 | 0.0±0.0 | 0.0±0.0 | 0.0±0.0 | 0.0±0.0 | 0.0±0.0 |
| *Direct Torque Cont. - High Freq. (500Hz)* | 0±0 | 17.2±0.316 | 1.37±0.51 | 10.4±2.19 | 10.3±0.586 | 0±0 | 0±0 | 0.97±5.72 | 2.08±5.84 | 45.3±6.74 | 0.0±0.0 |
| *Direct Torque Cont. - Low Freq. (31.25Hz)* | 54.5±7.15 | 100±1.21 | 69.2±0.62 | 98±2.55 | 80.6±2.56 | 100±0.0311 | 100±2.86e-15 | 53.0±9.35 | 59.2±3.72 | 100±1.94 | 80.7±2.16 |
| **EvoControl (Full State)** | **368±10.6** | **157±1.1** | **263±1.46** | **123±2.7** | **116±0.609** | **101±0.0487** | 100±2.86e-15 | **262±8.04** | **114±0.973** | **106±0.936** | **163±3.72** |
| **EvoControl (Residual State)** | **182±8.58** | **182±1.02** | 97±0.51 | **170±1.14** | **212±4.95** | 99.2±0.054 | 100±2.86e-15 | **271±7.75** | **106±1.29** | **104±1.19** | **165±2.19** |
| **EvoControl (Target + Proprio.)** | **319±14.1** | **168±1.41** | **164±5.08** | **165±1.77** | **165±4.94** | 99.7±0.0417 | 100±2.86e-15 | **255±7.61** | 96.8±3.54 | **105±0.776** | **143±1.86** |
| **EvoControl (Target)** | **293±13.2** | **162±1.58** | **272±1.84** | **164±1.8** | **205±5.04** | 99.6±0.0377 | 100±2.86e-15 | **255±6.93** | **112±0.785** | **105±0.78** | **151±2.44** |
| **EvoControl (Learned Gains)** | **266±14.1** | **113±1.6** | **198±9.62** | **150±2.55** | **117±0.205** | 99.5±0.0947 | 100±2.86e-15 | **239±7.61** | **116±0.747** | **105±1.21** | **158±3.63** |
| **EvoControl (Delta Position)** | **362±12.8** | **133±1.82** | **216±2.88** | **119±2.78** | **105±0.285** | **101±0.0364** | 100±2.86e-15 | **193±8.41** | 65.5±3.71 | 99.1±2.44 | **147±1.90** |

Table 4: Normalized evaluation returns ($\mathcal{R}$) for benchmarks trained over an equivalent number of low-level ($\beta$) environment steps (fixed environment duration of 2s per rollout). EvoControl here uses an evolutionary strategy with population size ($es\_pop\_size = 64$), detailed further in Appendix J.4.1. Results, averaged over 6400 random seeds (95% confidence intervals shown), demonstrate EvoControl's superior normalized returns compared to baseline methods (*fixed controllers* and *direct torque control*). Scores are normalized to a 0–100 scale, where 0 indicates a random policy and 100 represents the highest score achieved by any non-EvoControl baseline. Bold values highlight scores surpassing this baseline (scores >100).

| Same PPO high-level alg. $\rho$ with a Low-Level Policy $\beta$ of | Ant $\mathcal{R}\uparrow$ | Halfcheetah $\mathcal{R}\uparrow$ | Hopper $\mathcal{R}\uparrow$ | Humanoid $\mathcal{R}\uparrow$ | Humanoid Standup $\mathcal{R}\uparrow$ | Inverted Double Pend. $\mathcal{R}\uparrow$ | Inverted Pend. $\mathcal{R}\uparrow$ | Pusher $\mathcal{R}\uparrow$ | Reacher $\mathcal{R}\uparrow$ | Reacher 1D $\mathcal{R}\uparrow$ | Walker2D $\mathcal{R}\uparrow$ |
|---|---|---|---|---|---|---|---|---|---|---|---|
| *Fixed Cont. - PD Position* | 100±5.17 | 66±0.555 | 90.2±0.489 | 92.8±2.9 | 87.8±1.3 | 99.6±0.0357 | 100±1.53e-06 | 100±9.4 | 100±1.24 | 88.7±2.6 | 68.7±0.285 |
| *Fixed Cont. - PD Position Delta* | 4.93±1.67 | 2.88±0.0891 | 84.3±0.854 | 100±1.79 | 3.16±0.024 | 57.1±1.6 | 100±1.53e-06 | 3.4±9.98 | 42.8±2.99 | 27.9±7.48 | 85.4±0.513 |
| *Fixed Cont. - PD Int. Velocity* | 5.94±1.8 | 2.59±0.0938 | 56.9±2.3 | 72.6±0.872 | 0±0 | 61.3±1.59 | 99±0.45 | 0±0 | 0±0 | 9.58±7.72 | 100±0.681 |
| *Fixed Cont. - Random* | 0.0±0.0 | 0.0±0.0 | 0.0±0.0 | 0.0±0.0 | 0.0±0.0 | 0.0±0.0 | 0.0±0.0 | 0.0±0.0 | 0.0±0.0 | 0.0±0.0 | 0.0±0.0 |
| *Direct Torque Cont. - High Freq. (500Hz)* | 0±0 | 44±0.794 | 42.9±0.53 | 63.6±2.37 | 53±2.15 | 97.8±0.387 | 49.9±2.36 | 9.18±7.69 | 10.8±5.42 | 81.1±4.24 | 32.2±1.27 |
| *Direct Torque Cont. - Low Freq. (31.25Hz)* | 70.4±6.52 | 100±1.22 | 100±1.26 | 87.2±1.35 | 100±0.204 | 100±0.0178 | 100±1.53e-06 | 83.1±10.2 | 67.2±3.05 | 100±1.79 | 69.2±1.38 |
| **EvoControl (Full State)** | **188±12** | **165±1.15** | **118±22.4** | **111±15.9** | 90.2±6.6 | 100±0.306 | 100±0 | **296±42.6** | **109±1.86** | **104±2.14** | **127±126** |
| **EvoControl (Residual State)** | **152±117** | **166±16.9** | **125±118** | **108±43** | **112±14.2** | 99.2±0.304 | 100±0 | **302±18.1** | **102±1.08** | **104±2.07** | **169±59.6** |
| **EvoControl (Target + Proprio.)** | **183±69.2** | **135±11.1** | **130±55.2** | **126±12.1** | **124±58.8** | 87.1±53.3 | 100±0 | **300±42.6** | **107±2.77** | **104±2.23** | **130±12.7** |
| **EvoControl (Target)** | **201±27.5** | **164±10.6** | **131±93.5** | **125±4.52** | **109±12.2** | 99.5±1.7 | 100±0 | **286±27** | 83.5±63.4 | **103±3.2** | **122±38** |
| **EvoControl (Learned Gains)** | **103±23.1** | 90±20.6 | **141±845** | **111±28.9** | **102±1.5** | 93.9±26 | 100±0 | **263±46.8** | **104±11.4** | **102±4.85** | **149±35.2** |
| **EvoControl (Delta Position)** | **168±29.9** | **123±75.2** | 95.9±40.8 | **104±9.23** | 94.2±8.68 | 100±0.491 | 100±0 | **300±61.6** | 59±5.93 | 99.9±7.07 | **160±12.4** |

tioned obstacles penalizing collisions, explicitly designed to test the value of rapid, high-frequency control. Environment implementation details are provided in Appendix E.

**Benchmark Methods.** We compare EvoControl against established baselines, using the same high-level PPO policy ($\rho$) learning algorithm across all, varying only the low-level policy ($\beta$). We consider *fixed controllers*: PD Position, PD Position Delta, PD Integrated Velocity and PD Position & $K_p$ (Aljalbout et al., 2024); *direct torque control* at both high (500Hz) and low (31.25Hz) frequencies (Chen et al., 2023); a Random policy (30Hz); and several EvoControl ablations with varying state information provided to the low-level neural network controller (Table 2). Here, the EvoControl variants using position-based controllers are annealed from their corresponding PD controllers. Method implementation details are provided in Appendix F.

**Evaluation.** Unless otherwise stated we train each policy (high-level $\rho$ and low-level $\beta$) for 1M high-level steps. Post-training, we evaluate performance using 128 rollouts (different random seeds) per trained policy, calculating the return for each 1,000-step episode. We repeat this process

for three training seeds per baseline. Results are reported as the mean normalized score $\mathcal{R}$ (Yu et al., 2020) across all 384 evaluation rollouts (3 training seeds x 128 evaluation rollouts), scaled from 0 (random policy performance) to 100 (best non-EvoControl baseline)—detailed in Appendix H.

## 5.1. Main Results

We evaluated all benchmark methods across our environments, with results tabulated in Table 3 and Table 4, for the same number of high-level policy steps ($\rho$) and the same number of low-level environment steps, respectively. EvoControl, on average, achieves higher normalized evaluation return $\mathcal{R}$ on all environments. Specifically, EvoControl can both achieve a high average return $\mathcal{R}$ while learning a slow (31.25Hz) high-level policy and be able to solve an environment task with a fast (500Hz) learned low-level policy. Furthermore, EvoControl can outperform direct torque control at high frequency, and outperform the same high-level policy learning algorithm with position PD controllers, and we provide insights in Section 5.2. Moreover, we also show that EvoControl can support other ES for optimizing the low-level policy in Appendix J.1.

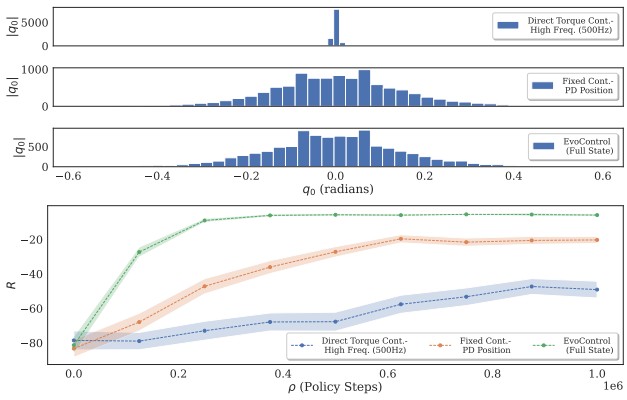

Figure 2: **Top 3 Sub-Plots: State visitation histogram for Reacher 1D.** Empirically demonstrating that PPO with high-frequency-direct torque control suffers from less efficient exploration compared to using PPO at a low-frequency with a fixed-high-frequency PD controller. EvoControl can achieve the same efficient exploration as a PD controller. **Bottom: Evaluation return $R$ versus $\rho$ policy steps on Reacher 1D.** PPO at high-frequency with direct-torque control experiences slower convergence compared to learning PPO at a low-frequency with a fixed PD controller. EvoControl can evolve its lower-level controller throughout training, leading to a higher evaluation reward in comparison—additional plots are provided in Appendix I.1.

## 5.2. Insight Experiments

Next, we analyze why EvoControl performs better than learning a policy with standard *fixed controllers* or *direct torque control*. We highlight the importance of controlling an environment at high frequency when required without sacrificing learning convergence ability.

**Does EvoControl Possess Efficient Exploration? (P1).** To explore if the benchmarked methods during training possess efficient exploration we analyze the learning curves for the Reacher 1D environment and their state space visitation frequency histograms in Figure 2 (with implementation details in Appendix I.1). We observe that EvoControl can initially achieve the same efficient state exploration due to temporal abstractions, similar to that of a fixed PD position controller, and then can further achieve a higher evaluation reward throughout training. This suggests that the higher-frequency control enabled by EvoControl, as theoretically motivated by Proposition 2.1, contributes to an increased reward. This is reminiscent of the principle behind Pulse Width Modulation (PWM), where higher frequency allows for finer control and more accurate signal representation (Huang et al., 2011). While a direct equivalence to PWM is not claimed, the ability of high-frequency actions to improve control, as demonstrated in Proposition 2.1, provides a theoretical

Table 5: Normalized evaluation returns ($\mathcal{R}$) on the Safety Critical Reacher environment, using the same normalization from Table 3. EvoControl effectively learns a low-level controller capable of rapid, adaptive responses at a high frequency, surpassing the fixed-frequency limitations of traditional PD position controllers and enabling quicker reactions to unexpected collisions. In contrast, baseline controllers at lower frequencies fail to detect collisions quickly enough, resulting in substantially lower performance.

| Same PPO high-level alg. $\rho$ with a Low-Level Policy $\beta$ of | Safety Critical Reacher $\mathcal{R}\uparrow$ |
|---|---|
| *Fixed Cont.* - PD Position | $100\pm19.2$ |
| *Fixed Cont.* - PD Position Delta | $0\pm0$ |
| *Fixed Cont.* - PD Int. Velocity | $67.1\pm11.4$ |
| *Fixed Cont.* - Random | $0.0\pm0.0$ |
| *Direct Torque Cont.* - High Freq. (500Hz) | $0\pm0$ |
| *Direct Torque Cont.* - Low Freq. (31.25Hz) | $35.9\pm17.1$ |
| **EvoControl (Full State)** | $205\pm14.1$ |
| **EvoControl (Residual State)** | $124\pm24.3$ |
| **EvoControl (Target + Proprio.)** | $237\pm11$ |
| **EvoControl (Target)** | $121\pm24.9$ |
| **EvoControl (Learned Gains)** | $169\pm7.15$ |
| **EvoControl (Delta Position)** | $213\pm14.6$ |

underpinning for EvoControl's improved performance. Crucially, we observe performing direct high-frequency torque control suffers from poor state exploration, given the same number of training steps (Appendix I.1).

**Can EvoControl Learn High-Frequency Interaction Control? (P2).** High-frequency control can be crucial for safety-critical tasks like collision avoidance, where rapid responses to unexpected contacts are paramount. To investigate such a setting, we adapted the Reacher 1D environment to introduce a random object in 25% of the episodes which block the arm from reaching its intended goal and add both an observation for any measured contact force and a reward penalty for this contact force (*Safety Critical Reacher*). We tabulate the normalized performance of all baselines in Table 5. We observe that EvoControl is able to observe and react faster at high-frequency to un-modelled collisions, compared to a low-frequency policy with a fixed-high-frequency state tracking PD controller. Critically such a collision detection and avoidance environment exemplifies our intuition from Proposition 2.1, that higher-frequency actions can achieve a higher reward. Intuitively, a well-performing policy requires a change in behavior as soon as any un-modeled collision is detected, intuitively similar to fast automatic reflexes for a low-level system controller with a high-level system, beyond simple goal-state tracking. We provide experimental details in Appendix I.2.

This finding is further supported by experiments on the higher-dimensional *Safety Critical HalfCheetah* task, detailed in Appendix J.14, which features a more complex 19-dimensional state space. In this task, EvoControl sim-

ilarly excels, leveraging high-frequency control to rapidly detect and adapt to collisions, reinforcing the robustness and generalizability of the approach.

Furthermore, we validated EvoControl's practicality for real-world deployment by demonstrating zero-shot sim-to-real transfer on a 7-DoF Franka Emika Panda robot for tabletop manipulation tasks. Our real-world validation showed Evo-Control can operate effectively at high-frequency (200Hz low-level control) with rapid inference (average 64 μs per step), comfortably surpassing typical robotic control rates. Importantly, EvoControl reduced collision forces compared to a tuned PD controller, highlighting benefits for safety-critical applications. We include representative results and visualizations in Figure 3 and Figure 4, and refer to Appendix K for comprehensive details of these real-robot experiments.

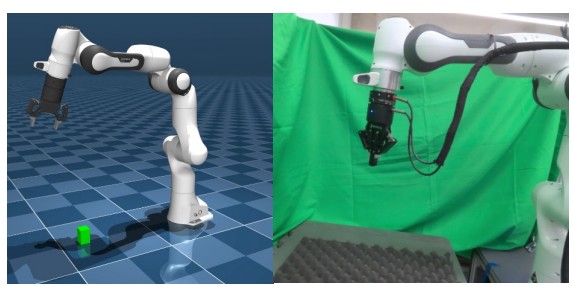

Figure 3: **Real-World Robot Setup.** Left: MuJoCo simulation of the Franka Emika Panda robot setup. Right: Corresponding real hardware configuration for sim-to-real validation.

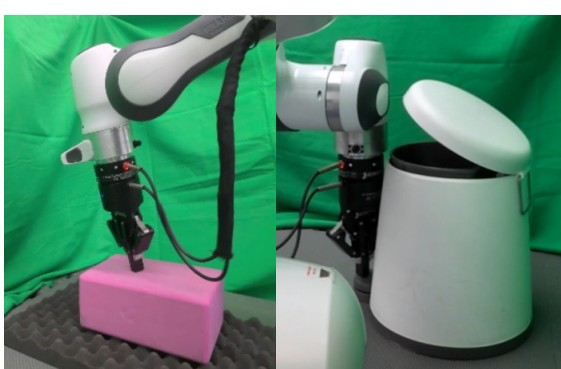

Figure 4: **Real-Robot Tasks.** Left: Block collision experiment measuring collision forces. Right: Bin-opening task demonstrating gentle high-frequency torque control.

**Can EvoControl Automate Tuning of PD parameters? (P3).** A widespread limitation of any fixed-PD-low-level controller is the inherent sensitivity of its state-tracking performance to that of its fixed gains $(K_p, K_d)$. In practical scenarios such gains require careful manual tuning to each task and environment of operation. Therefore having an approach that can be more robust to tuning PD gains than just their inherent sensitivity is practically useful. Empiri-

Table 6: Normalized evaluation returns ($\mathcal{R}$) for benchmark methods on the Reacher 1D environment, using the same normalization from Table 3. EvoControl is more robust to the tuning of the underlying PD controller than existing fixed PD controllers, which can degrade as their PD controller parameter ($K_p$) becomes less tuned for the task.

| Same PPO high-level alg. $\rho$ with a Low-Level Policy $\beta$ of | $K_P = 0.001$ $\mathcal{R}\uparrow$ | $K_P = 0.1$ $\mathcal{R}\uparrow$ | $K_P = 1.0$ $\mathcal{R}\uparrow$ | $K_P = 10.0$ $\mathcal{R}\uparrow$ |
|---|---|---|---|---|
| *Fixed Cont.* - PD Position | 0±0 | 13.5±7.39 | 81.8±2.81 | 100±0.893 |
| *Fixed Cont.* - PD Position Delta | 0±0 | 0±0 | 14.6±7.31 | 78.8±3.58 |
| *Fixed Cont.* - PD Int. Velocity | 0±0 | 0±0 | 0±0 | 80.5±4.17 |
| *Fixed Cont.* - Random | 0.0±0.0 | 0.0±0.0 | 0.0±0.0 | 0.0±0.0 |
| *Direct Torque Cont.* - High Freq. (500Hz) | 43.6±6.48 | 43.6±6.48 | 43.6±6.48 | 43.6±6.48 |
| *Direct Torque Cont.* - Low Freq. (31.25Hz) | 96.2±1.87 | 96.2±1.87 | 96.2±1.87 | 96.2±1.87 |
| **EvoControl (Full State)** | **99.1±3.13** | **99.2±7.07** | **102±3.25** | **101±0.36** |
| **EvoControl (Residual State)** | **100±4.55** | **99.5±5.62** | **100±3.29** | **102±2.18** |
| **EvoControl (Target + Proprio.)** | **98.6±3.5** | **97.9±1.48** | **101±4.43** | **101±0.894** |
| **EvoControl (Target)** | 94.6±6.85 | 96.5±10.1 | 101±1.58 | 100±0.995 |
| **EvoControl (Learned Gains)** | 91.8±5.37 | 93.4±6.69 | 101±2.35 | 100±3.04 |
| **EvoControl (Delta Position)** | 95.8±8.29 | 97.4±1.92 | 95.4±12.2 | 97.6±8.52 |

cally, evaluating on the Reacher 1D task, with varying the $K_p = \{0.001, 0.1, 1.0, 10.0\}$ gain, we observe EvoControl achieving a higher average normalized return $\mathcal{R}$ than a PD controller, crucially as the PD controllers become less tuned $K_p \to 0$ their performance decreases, highlighting the sensitivity of these fixed controllers. This highlights EvoControls robustness to PD parameters, which could arise due to the PD controller providing a semantically meaningful initial latent action $a_k$ that the lower-level policy can then refine. Ablating the initial PD controller annealing in EvoControl destabilizes learning of both the high-level and low-level policies, confirming its importance (Appendix J.5).

## 6. Conclusion and Future Work

In this paper, we present *EvoControl*, a novel bi-level policy learning framework for learning both a slow high-level policy and a fast low-level controller using PPO and ES, respectively, for continuous-control tasks. Theoretically, we show that there exist some CTMDPs in which acting at higher frequencies can yield a strictly higher expected cumulative reward. Empirically, EvoControl outperforms existing high-frequency control methods, particularly in tasks requiring fast reactions. Moreover the limitations of the current approach, are that EvoControl still relies on the existence of a fixed-PD controller for the task (common in robotics applications, Appendix J.5) and can use more computational complexity (Appendix J.4) compared to only performing PPO, which can be readily parallelized in practice with modern accelerated compute platforms, both could be readily improved. In addition, promising future directions include exploring more complex nested hierarchies, direct low-level to high-level information flow, and ensembles of policies (Appendix L).

## Acknowledgements

We thank the anonymous reviewers, and area chairs, and specifically Francesco Nori, Leonard Hasenclever, Steven Bohez, Thomas Lampe, Baruch Tabanpour, Antoine Moulin, Nimrod Gileadi, Jose Enrique Chen and Taylor Howell for their insightful comments and suggestions that ultimately improved this work.

## Impact Statement

Our novel bi-level policy learning framework, which trains a fast-reacting low-level controller alongside a slower high-level policy, offers the promise of safer, better control in areas such as surgical robotics, autonomous driving, and industrial automation. By enabling high-frequency decision-making, it can better handle unexpected events and reduce risks. However, leveraging learned control for safety-critical systems requires rigorous testing and transparency to prevent unexpected failures or behaviors. Organizations adopting this technology should address potential biases, prioritize robustness, and maintain thorough oversight to ensure reliable, ethical deployment.

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

# Appendix

## Table of Contents

# A. Expanded Problem

In the following we expand the problem setup from the main paper.

**States & Actions.** We denote the environments state space as $\mathcal{S} \subset \mathbb{R}^{d_s}$ and its action space as $\mathcal{U} \subset \mathbb{R}^{d_u}$. At time $t \in \mathbb{R}$, the system's state is represented by $\mathbf{s}_t \in \mathcal{S}$, and its action by $\mathbf{u}_t \in \mathcal{U}$. Considering action (e.g. actuator) limits the action space is constrained to a box in Euclidean space: $\mathcal{U} = [\mathbf{u}_{\min}, \mathbf{u}_{\max}]$.

**Environment Dynamics.** The transition dynamics for continuous control environments can be described by an underlying unknown *differential equation* of $\dot{s}_t = \frac{ds_t}{dt} = f(s_t, u_t)$. The transition function, which describes the evolution of the state over a discrete time step $\Delta_t$, can be approximated using the Euler method $s_{t+\Delta_t} \approx s_t + \Delta_t f(s_t, u_t)$. Given an action $u_t$ and current state $s_t$, $s_{t+\Delta_t} \sim P(s_{t+\Delta_t}|s_t, u_t)$ is implicitly defined by this approximation. More sophisticated numerical integration schemes (e.g., Runge-Kutta methods) can be employed for higher accuracy. In stochastic environments, the dynamics function $f$ can be considered to be stochastic, leading to a probability distribution over next states given the current state and action. We consider the setting where there is an additional observation function that maps the current environment state to an underlying observation $z_t = g(s_t) + \epsilon$, where $\epsilon$ is optional observation noise, e.g. Gaussian noise with zero mean $\epsilon_t \sim \mathcal{N}(0, \sigma_\epsilon^2)$[4]. To simplify notation we use observation and state interchangeably and clarify the specifics when needed.

**Policies.** The agent can be represented as a single policy $\pi : \mathbb{R}^{d_s} \to \mathbb{R}^{d_u}$, that observes the current observation at time $t$ and samples an action $u_t \sim \pi(s_t)$ and then applies this action to the environment at a given fixed $\Delta_t$. In the case of a stochastic policy, the action is sampled from a distribution conditioned on the state: $u_t \sim \pi(\cdot|s_t)$.

To formalize a bi-level policy, we decompose $\pi$ into two components: a slow, high-level policy, $\rho$, and a fast, low-level policy, $\beta$. Both policies interact with the environment as described in Algorithm 1. The high-level policy $\rho$ operates at a lower frequency and outputs a high-level (latent) action, $a_k \sim \rho(s_k)$, at time step $k$. This latent action often represents a desired high-level target, such as a target position or velocity. The low-level policy $\beta$ operates at a higher frequency and receives the high-level action $a_k$ as input. $\beta$ then generates the low-level actions, $u_{k+i}$, at a finer-grained time step $i$, corresponding to direct motor torques or other low-level control signals. These low-level actions aim to achieve the high-level target specified by $\rho$. We denote the high-level time steps as $k$ and low-level time steps as $i$ to maintain this distinction. The fast, low-level policy $\beta$ operates at a frequency of $1/\Delta_t$, where $\Delta_t$ is the low-level time step. The slow, high-level policy $\rho$ guides the low-level policy over a longer horizon. Specifically, $\rho$ issues a latent action which is executed by $\beta$ for $G$ steps. This means $\rho$ effectively operates at a frequency of $1/(G\Delta_t)$ with a time step of $G\Delta_t$.

**Objective.** The environment produces a reward $r_t$ sampled from an unknown reward function $r(s_t, u_t)$, $r : \mathcal{S} \times \mathcal{U} \to \mathbb{R}$. The overall objective of the agent is to maximize the expected future discounted reward $\mathbb{E}_{s_{0:T}, u_{0:T-1}, R_{0:T-1}} \left[ \sum_{i=0}^{T-1} \gamma^i r_i \right]$, where $0 \leq \gamma < 1$ is the discount factor.

**Markov Decision Process (MDP).** We can model the environment as a Markov Decision Process (MDP), defined by the tuple $\mathcal{M} = \langle \mathcal{S}, \mathcal{U}, P, r, \gamma \rangle$, where:

- $\mathcal{S} \subset \mathbb{R}^{d_s}$ is the continuous state space.

- $\mathcal{U} \subset \mathbb{R}^{d_u}$ is the continuous action space.

- $P(s_{t+\Delta_t}|s_t, u_t)$ is the state transition probability distribution, implicitly defined by the dynamics function $f$ and the discretization scheme (e.g., Euler method).

- $r : \mathcal{S} \times \mathcal{U} \to \mathbb{R}$ is the reward function, providing a scalar reward $r_t = r(s_t, u_t)$ at each time step.

- $\gamma \in [0, 1)$ is the discount factor, determining the importance of future rewards.

---

[4]For the standard MuJoCo Brax environments we use, the joint velocity observation has Gaussian noise added to it, following the standard implementation of the environments (Freeman et al., 2021).

# B. Proof of Proposition 2.1: Existence of CTMDPs Requiring Higher-Frequency Control for Optimality

**Proposition B.1.** *Consider a continuous-time Markov Decision Process (CTMDP) with a finite state space and action space. Let this CTMDP be discretized with time step $\Delta_t$. There exist CTMDPs such that for any fixed discretization step $\Delta_t > 0$, there exists a finer discretization $\Delta_t' < \Delta_t$ where the optimal policy for the $\Delta_t'$-discretized MDP achieves a higher expected cumulative reward over a fixed time horizon $T$ than the optimal policy for the $\Delta_t$-discretized MDP.*

**Full Set of Assumptions:**

We consider a CTMDP with state space $S$ and action space $A$. When discretized with a time step $\Delta_t$, the transition probabilities are governed by a transition rate matrix $Q(a)$, where $Q_{ij}(a)$ denotes the transition rate from state $i$ to state $j$ under action $a$. The transition probabilities in the discretized MDP are then given by:

$$P(s_j|s_i, a, \Delta_t) = \begin{cases} [e^{Q(a)\Delta_t}]_{ij} + O(\Delta_t^2) & i \neq j \\ [e^{Q(a)\Delta_t}]_{ii} + O(\Delta_t^2) & i = j \end{cases}$$

where $e^{Q(a)\Delta_t}$ is the matrix exponential. This is a standard assumption when discretizing CTMDPs, justifiable by the Taylor expansion of the matrix exponential (see, e.g., Norris (1998), Chapter 2, Theorem 2.8.2). We assume that the error terms $O(\Delta_t^2)$ converge uniformly to zero as $\Delta_t \to 0$. The reward function $r(s, a)$ is the same for both the continuous and discretized cases. The time horizon $T > 0$ is fixed, and we assume $T$ is an integer multiple of both $\Delta_t$ and $\Delta_t'$.

*Proof.* **CTMDP and MDP Definition:** We consider a CTMDP with the following properties:

- **State Space:** $S = \{s_{\text{good}}, s_{\text{bad}}\}$

- **Action Space:** $A = \{a_{\text{maintain}}, a_{\text{recover}}\}$

- **Transition Rates:**

    - Under $a_{\text{maintain}}$: transition rate from $s_{\text{good}}$ to $s_{\text{bad}}$ is $p > 0$. No other transitions.
    - Under $a_{\text{recover}}$: instantaneous transition from $s_{\text{bad}}$ to $s_{\text{good}}$. No other transitions.

- **Reward Function:** $r(s_{\text{good}}, a) = r_{\text{good}} > 0$; $r(s_{\text{bad}}, a) = r_{\text{bad}} < 0$.

- **Discount Factor:** $\gamma = 1$ (undiscounted)

- **Episode Duration:** $T$

- **Initial State:** $s_{\text{good}}$

Discretizing this CTMDP with time step $\Delta_t$ yields an MDP with transition probabilities as described in the assumptions above, derived from the transition rates. Specifically, for small $\Delta_t$:

$$P(s_{\text{bad}}|s_{\text{good}}, a_{\text{maintain}}, \Delta_t) \approx p\Delta_t$$
$$P(s_{\text{good}}|s_{\text{good}}, a_{\text{maintain}}, \Delta_t) \approx 1 - p\Delta_t$$
$$P(s_{\text{good}}|s_{\text{bad}}, a_{\text{recover}}, \Delta_t) = 1$$

**Cumulative Reward Definition:** The cumulative reward $R(\pi, T, \Delta_t)$ for a policy $\pi$ over a time horizon $T$ in the MDP with discretization $\Delta_t$ is:

$$R(\pi, T, \Delta_t) = \mathbb{E}_\pi \left[ \sum_{t=0}^{T/\Delta_t - 1} r(s_t, a_t)\Delta_t \right]$$

**Proof Steps:**

**1) Optimal Policies ($\pi^*(\Delta_t)$):** For any discretization $\Delta_t$, the optimal policy $\pi^*(\Delta_t)$ is to apply $a_{\text{recover}}$ in $s_{\text{bad}}$ and $a_{\text{maintain}}$ in $s_{\text{good}}$.

**2) Expected Cumulative Reward (Finer Discretization $\Delta'_t = \frac{\Delta_t}{n}$):** Consider a finer discretization $\Delta'_t = \frac{\Delta_t}{n}$ for some integer $n > 1$. Under $\pi^*(\Delta'_t)$, the probability of transitioning to $s_{bad}$ and *staying* there for a time $k\Delta'_t$ within a $\Delta_t$ interval is approximately $\binom{n}{k}(p\frac{\Delta_t}{n})^k(1 - p\frac{\Delta_t}{n})^{n-k}$. Since recovery is immediate, the expected time in $s_{bad}$ within a $\Delta_t$ interval is $\sum_{k=1}^{n} k\Delta'_t \binom{n}{k}(p\frac{\Delta_t}{n})^k(1 - p\frac{\Delta_t}{n})^{n-k} = p\frac{(\Delta_t)^2}{n}$. The expected time spent in $s_{good}$ is then $\Delta_t - \frac{p(\Delta_t)^2}{n}$. Thus the expected reward per $\Delta_t$ interval is:

$$(\Delta_t - \frac{p(\Delta_t)^2}{n})r_{good} + \frac{p(\Delta_t)^2}{n}r_{bad} = \Delta_t r_{good} - \frac{p(\Delta_t)^2}{n}(r_{good} - r_{bad})$$

And with $\frac{T}{\Delta_t}$ such intervals, the total expected cumulative reward is:

$$R(\pi^*(\Delta'_t), T, \Delta'_t) = \frac{T}{\Delta_t}(\Delta_t r_{good} - \frac{p(\Delta_t)^2}{n}(r_{good} - r_{bad}))$$
$$= Tr_{good} - \frac{Tp\Delta_t}{n}(r_{good} - r_{bad})$$

**3) Expected Cumulative Reward (Coarser Discretization $\Delta_t$):** For a fixed $\Delta_t$, the stationary distribution probabilities under $\pi^*(\Delta_t)$ can be approximated by solving the balance equations: $\mu_{good}(1 - p\Delta_t) + \mu_{bad}(1) = \mu_{good}$ and $\mu_{good}(p\Delta_t) + \mu_{bad}(0) = \mu_{bad}$. Also $\mu_{good} + \mu_{bad} = 1$. This gives $\mu_{bad} = p\Delta_t$ and $\mu_{good} = 1 - p\Delta_t$.

Thus, the expected reward per $\Delta_t$ step is:

$$R_{step} = \mu_{good}r_{good} + \mu_{bad}r_{bad}$$
$$= (1 - p\Delta_t)r_{good} + (p\Delta_t)r_{bad}$$
$$= r_{good} - p\Delta_t r_{good} + p\Delta_t r_{bad}$$
$$= r_{good} - p\Delta_t(r_{good} - r_{bad})$$

Over the entire horizon $T$, with $\frac{T}{\Delta_t}$ steps, this gives:

$$R(\pi^*(\Delta_t), T, \Delta_t) = \frac{T}{\Delta_t}(r_{good} - p\Delta_t(r_{good} - r_{bad}))\Delta_t$$
$$= Tr_{good} - Tp\Delta_t(r_{good} - r_{bad})$$

**4) Comparison:** Comparing the rewards, we have:

$$R(\pi^*(\Delta'_t), T, \Delta'_t) - R(\pi^*(\Delta_t), T, \Delta_t)$$
$$= (Tr_{good} - \frac{Tp\Delta_t}{n}(r_{good} - r_{bad})) - (Tr_{good} - Tp\Delta_t(r_{good} - r_{bad}))$$
$$= Tr_{good} - \frac{Tp\Delta_t}{n}(r_{good} - r_{bad}) - Tr_{good} + Tp\Delta_t(r_{good} - r_{bad})$$
$$= -\frac{Tp\Delta_t}{n}(r_{good} - r_{bad}) + Tp\Delta_t(r_{good} - r_{bad})$$
$$= Tp\Delta_t(r_{good} - r_{bad})(1 - \frac{1}{n})$$

Since $r_{good} > r_{bad}$, $p > 0$, $\Delta_t > 0$, and $n > 1$, this difference is positive. Therefore, for any $\Delta_t$, a finer discretization $\Delta'_t = \frac{\Delta_t}{n}$ with $n > 1$ leads to a higher expected cumulative reward under its optimal policy. $\qquad\square$

### B.1. Intuitive Continuous Control Safety-Critical Example

Consider a safety-critical task involving a one-degree-of-freedom robot arm. The arm's state is its joint angle $\theta_t$, and the action is the motor torque $\tau_t$. The goal is to reach a target angle $\theta_{\text{goal}}$ from an initial angle $\theta_0$ within a fixed episode duration $T$. An immovable obstacle may appear in a random subset of episodes (e.g., 25% of the time), obstructing the direct path to the goal. The reward function encourages reaching the target angle while penalizing contact forces with the obstacle:

$$R = \int_0^T \left( r_{\text{goal}} e^{-|\theta_t - \theta_{\text{goal}}|} - r_{\text{collision}} F_t \right) dt$$

where $r_{\text{goal}}$ and $r_{\text{collision}}$ are positive weighting constants, and $F_t$ is the magnitude of the contact force between the arm and the obstacle at time $t$ (0 if no contact). The policy receives the observation $O_t = (\theta_t, \dot{\theta}_t, \tau_{t,\text{measured}})$, where $\tau_{t,\text{measured}}$ is the measured torque, reflecting contact forces if any.

A standard approach might use a position PD controller with a low-frequency high-level policy that provides the target angle $\theta_{\text{target}}$. However, this approach faces limitations. If the obstacle is present, the PD controller will exert a continuous force against it, incurring significant penalties. The low-frequency policy might only detect the collision after a substantial delay, making it difficult to react effectively.

A high-frequency policy, on the other hand, can detect the collision much faster and take corrective action. Upon detecting a sudden increase in $\tau_{t,\text{measured}}$, it can immediately reduce the motor torque, minimizing the contact force $F_t$. Furthermore, a sophisticated high-frequency policy can learn to approach the target cautiously, probing for the obstacle with small torques. If contact is detected, it can adjust its trajectory to reach the goal while avoiding further collisions.

This intuitive example illustrates how high-frequency control can be crucial for safety-critical tasks. It enables faster reaction to unexpected events and allows for more nuanced control strategies that consider the full reward structure, including collision avoidance. This motivates the development of methods like EvoControl, capable of effectively learning such high-frequency policies. This example highlights scenarios where high-frequency control offers a significant advantage over traditional low-frequency control coupled with fixed controllers, especially in tasks requiring rapid responses and nuanced interaction behaviors.

## C. Expanded Background: Fixed PD Controllers

Low-level PD controllers are extensively used within robotics applications, specifically when combined with a learned high-level policy $\rho$. This hierarchical structure simplifies the learning problem and effectively reduces the number of decision steps for the high-level policy. This reduction is achieved by allowing the high-level policy to operate at a timestep of $G\Delta_t$, where $G$ is the number of low-level actions executed per high-level action, effectively reducing the number of high-level actions within a fixed episode duration $T$.

In continuous control and robotics, these hierarchical structures, composed of a learned high-level policy ($\rho$) and a fixed low-level controller (e.g., a PD controller (Oku et al., 2018)), are common (Peng & Van De Panne, 2017; Song et al., 2019; Chentanez et al., 2018; Peng et al., 2018; Xie et al., 2020). The high-level policy outputs a target $a_k$ which the low-level PD controller tracks using a control signal based on the error between $a_k$ and the measured system state $s_t$.

Commonly, PD control is designed to track a second-order signal, such as position or velocity. The control signal, $u_t$, is given by:

$$u_t = K_p(a_k - s_t) + K_d(\dot{a}_k - \dot{s}_t), \tag{1}$$

where $K_p, K_d \in \mathbb{R}^+$ are constant proportional and derivative gains, and $e_t = a_k - s_t$ represents the tracking error.

Specifically for robotics, proprioceptive observed states can be represented as joint positions ($q_t$), joint velocities ($\dot{q}_t$), and torques ($\tau_t$). This leads to several common PD control designs, summarized in Table 1. These designs differ in how the high-level target $a_k$ is interpreted and used in the control law. For instance, in "PD Absolute Position," $a_k$ directly specifies the desired joint position ($q^d$), while in "PD Delta Position," $a_k$ represents a change in joint position ($\delta q^d$) relative to the current position. The state $s_t$ in the control law can encompass a wider range of proprioceptive information beyond just joint positions ($q_t$). We present the target $a_k$ and tracking error in terms of position/velocity for clarity and to align with common PD controller formulations.

# D. Extended Related Work

Table 7: **Comparison with related bi-level learning approaches in RL.** Our method, EvoControl, can achieve efficient state-action space exploration, whilst learning high-frequency interaction behavior, and avoids tuning of PD parameters.

| Approach | Ref. | $\pi$ | Low-level $\beta$ Reward | High-level Action Duration $\Delta_\rho$ | High-level Action $\rho$ | (P1) Efficient Exploration | (P2) High-Frequency Interaction Control | (P3) Automate Controller Tuning PD Parameters |
|---|---|---|---|---|---|---|---|---|
| Fixed Controllers | (Song et al., 2019) | $\{\rho_{\text{nn}}, \beta_{\text{pd\_controller}}\}$ | $-\|s(t) - s_{\text{desired}}(t)\|_2$ | $G\Delta_t$ | $\{q_d, \dot{q}_d, \tau_d\}$ | ✓ | ✗ | ✗ |
| Direct Torque Control | (Peng & Van De Panne, 2017) | $\{\rho\}$ | $R = \sum_{i=0}^{T/\Delta_t - 1} r(s_{i\Delta_t}, u_{i\Delta_t})\Delta_t$ | $\Delta_t$ | $a(t)$ | ✗ | ✓ | ✓ |
| HRL: Skills | (Sutton et al., 1999; Rao et al., 2021) | $\{\rho_{\text{manager}}, \{\beta_0, \beta_1, \ldots, \beta_n\}\}, n \in \mathcal{Z}_n$ | $R = \sum_{i=0}^{T/\Delta_t - 1} r(s_{i\Delta_t}, u_{i\Delta_t})\Delta_t$ | $z_n \in \mathcal{Z}_n$ | ✗ | ✓ | ✓ | |
| HRL: Sub Goals | (Nachum et al., 2018b) | $\{\rho, \beta\}$ | $-\|s(t) - s_{\text{desired}}(t)\|_2$ | $G\Delta_t$ | $s_{\text{desired}}(t)$ | ✓ | ✗ | ✓ |
| **EvoControl** | **(Ours)** | $\{\rho, \beta\}$ | $R = \sum_{i=0}^{T/\Delta_t - 1} r(s_{i\Delta_t}, u_{i\Delta_t})\Delta_t$ | $G\Delta_t$ | $z \in \{q_d, \dot{q}_d, \tau_d\}$ | ✓ | ✓ | ✓ |

Existing approaches to continuous control in robotics primarily fall into two categories: those employing fixed low-level controllers and those utilizing direct torque control learned end-to-end. EvoControl aims to addresses limitations inherent in both approaches, and we summarize the key differences in Table 7.

**Fixed Low-Level Controllers:** A common strategy involves combining a learned high-level policy (often operating at low frequency, e.g., 10-30Hz) with a fixed, high-frequency low-level controller (e.g., a Proportional-Derivative (PD) controller operating at 500Hz or higher) (Song et al., 2019; Chentanez et al., 2018; Peng et al., 2018; Xie et al., 2020). The high-level policy generates setpoints (e.g., desired positions or velocities), and the low-level controller tracks these setpoints by adjusting actuator torques. While prevalent, this approach suffers from several drawbacks. The low-level controller's parameters require careful tuning, and its fixed nature limits its ability to handle high-frequency interactions such as unexpected collisions or disturbances. Furthermore, recent work applying RL algorithms to the physical world often restricts itself to relatively low-frequency control ($\sim$20Hz) due to the reliance on analytical impedance controllers (Martín-Martín et al., 2019; Luo et al., 2018; Johannink et al., 2019). Even hierarchical approaches employing analytical controllers often limit high-level policy frequencies (Davchev et al., 2022). EvoControl, in contrast, aims to achieve efficient exploration while also enabling the learning of flexible and complex high-frequency behaviors in the low-level policy.

**Evolutionary Strategies:** Direct evolutionary strategies (ES approaches, (Bäck et al., 2013)) have been shown to provide an alternative for solving reinforcement learning environments; however the direct application of them, as shown by others are that they can be sample inefficient, get stuck in global minima; however excel at discovering good performing long-horizon tasks, sparse reward tasks and delayed reward tasks, as they often optimize the episodic return, rather than the intermediate temporal difference return (Salimans et al., 2017). There exist works formulating hierarchical ES for both levels, however still under-perform gradient-based RL policy methods (Abramowitz & Nitschke, 2022). EvoControl through it's novel combination of a PPO learned high-level policy, and a ES-learned low-level policy empirically outperforms the ablation version of using ES for both the high-level and low-level in EvoControl, as shown in Appendix J.9.

**Direct Torque Control:** Alternatively, some methods learn an end-to-end policy that directly outputs joint torques at a high frequency (Peng & Van De Panne, 2017; Wahlström et al., 2015; Watter et al., 2015). This approach, while potentially offering greater adaptability, faces significant challenges. High-frequency control suffers from the curse of dimensionality imposed by the increased number of time steps in long horizons. The resulting explosion in the number of possible action sequences significantly hinders exploration and can lead to suboptimal policies (Martín-Martín et al., 2019; Peng & Van De Panne, 2017). EvoControl mitigates these challenges by employing a hierarchical structure, enabling more efficient exploration while retaining the adaptability afforded by direct torque control at the low level. Moreover, the related work of Peng & Van De Panne (2017) compares learning policies with four different action spaces of direct torque control, PD position control, PD velocity control and a muscle activation's for the task of imitating gaits for planar walking robot environments (continuous control). Their findings correlate with ours, in that they observed on average faster learning convergence and higher task reward using a low-level high-frequency (fast) controller, such as PD controller compared to performing direct torque control. Additionally Peng & Van De Panne (2017) due to having no prior controller parameters for the environments that they wanted to control, Peng & Van De Panne (2017) similarly performed a related approach where they optimized the fixed low-level controller parameters throughout training a high-level policy. However, all of their low-level controllers used are simple, few parameter (2-7) controllers, such as a PD controller, and such fixed simple controllers are all only capable of sub-goal simple tracking behavior. Whereas EvoControl, can represent the fast lower-level policy with a neural network policy and learn this throughout training the high-level policy, learning fast adaptive behavior of the low-level policy, that goes beyond simple sub-goal tracking behavior. Furthermore, the related work of Reda et al. (2020) studied environment design for continuous control tasks, and found that varying the control frequency of performing direct torque control in standard Mujoco Gym like environments (e.g. Ant, Hopper) could yield better learning and overall policy return, however requires tuning the control frequency (or discrete action repeats of the simulation timestep $\Delta_t$)

for each environment and task to get the best performance—likely due to matching the inherent control frequency of the dynamics of the environment. They also studied the use of learning with PD controllers, and determined that PD controllers can aid in converging faster to good policy, however can get stuck in lower-reward solutions (local minima), motivating the need for a method to practically perform high-frequency torque control. In summary, EvoControl can overall learn a high-frequency policy $\pi$ by learning both a slow-high-level policy $\rho$ combined with a fast-low-level policy $\beta$, learning adaptive low-level behavior of an equivalent high-frequency policy, avoiding the difficulties of learning a direct torque control high-frequency policy directly. Furthermore, we provide empirical evidence for the difficulty of learning a direct torque control high-frequency policy, as even with an ever increasing number of training steps, such a policy may converge to local minima Appendix J.3.

**Hierarchical Reinforcement Learning (HRL):** EvoControl draws inspiration from the HRL paradigm, which decomposes complex tasks into simpler subtasks managed by separate policies. Existing HRL methods such as options frameworks (Sutton et al., 1999; Bacon et al., 2017) and hierarchical actor-critic architectures (Riedmiller et al., 2018; Vezzani et al., 2022) have been successfully applied to improve exploration and learning efficiency. However, these methods typically focus on discrete skill selection or subgoal decomposition (Heess et al., 2016), while EvoControl explicitly addresses the challenges of learning a low-level controller for continuous high-frequency control, enabling semantically meaningful exploration in different control modes. Related work in the RHPO/HO2/MO2/HeLMS family (Rao et al., 2021; Salter et al., 2022; Wulfmeier et al., 2020a;b) has also explored hierarchical approaches. However, unlike typical HRL, which focuses on skill discovery, EvoControl targets learning a fast low-level policy that complements the slow high-level policy. Uniquely, EvoControl combines PPO and ES within its hierarchical framework for efficient exploration and complex high-frequency control.

**Hybrid Combinations of RL and ES:** Existing related work has looked into combining evolutionary strategies ES to improve RL algorithms, specifically using them to collect diverse data as ES methods show superior exploration capabilities compared to on-policy and off-policy RL algorithms, and also take updates for the RL agent itself (Sigaud, 2023; Suri et al., 2020; Khadka & Tumer, 2018; Conti et al., 2018; Li et al., 2024; Bodnar et al., 2020). Specifically, Suri et al. (2020); Khadka & Tumer (2018) use RL (SAC/DDPG) with ES data collection to collect diverse trajectories into the replay buffer to update the RL agent. Suri et al. (2020) propose automatic mutation tuning to improve the ES component, and demonstrate improved performance on 10 out of 15 continuous-control environments compared to the equivalent RL method baselines. Khadka & Tumer (2018) also uses ES (Neuroevolution) to collect diversified trajectories, and use these trajectories in a replay buffer to train an off-policy RL agent. They further, update the ES data collection agent with snapshots of the trained RL agent throughout training, and demonstrate on continuous control tasks that this can lead to higher reward evaluation and faster convergence in higher-dimensional state-action challenging environments. Furthermore, Zheng et al. (2020) proposes a transfer approach that has a pool of agents containing three classes of agents: on-policy agents, off-policy agents, and a population-based ES agents. All agents explore and collect trajectories into a replay buffer, with the on-policy and ES agent initially transferred from the weights off-policy (global) agent; the trajectories of the on-policy agents are then more frequently sampled when used to update the off-policy global agent, and use a threshold to control the frequency of policy parameter transfer. There also exist alternative solutions to combine ES and RL, such as seeding ES with an RL agent for symbolic regression (Mundhenk et al., 2021). Moreover, Elfwing et al. (2007) proposed a task decomposition method, using MAXQ, to break down a complex task into a hierarchy of subtasks on small dimensional state-action space problems, and used a genetic programming algorithm to learn the hierarchical task decomposition automatically. Specifically in Elfwing et al. (2007), each hierarchy corresponds to a different subtask that can be performed. Unlike EvoControl, all of these related works do not consider the problem of learning at higher frequencies, operating their continuous control environments at default large discrete time steps (e.g. 20-100Hz (Brockman et al., 2016a)), and do not focus on an hierarchical approach of having a high-level policy outputting a latent action and a low-level policy following this latent action for $G$ steps (providing temporal action abstractions)—which limits the practical deployment of their agent, as if rich features are used as inputs to the agent such as images or the use of larger architectures, such as Vision Language Transformers, the inference time of the agent would increase (e.g. $\approx 30Hz$ for images from video), limiting the agents use where high-frequency control is necessary for an environment. Conversely, EvoControl enabled from its hierarchical structure decomposing a high-level policy and a low-level policy, the low-level policy can remain a simple neural network agent of a smaller size being able to run with a fast inference time, and hence fast control operation of $500 - 1KHz$, and still gain the benefit of having a higher-level policy that can still take as input rich features arriving at a lower-frequency such as images.

**Continuous-Time Control & Planning:** A surge of recent work tackles reinforcement learning in *continuous-time* settings where observations arrive irregularly and actions experience latency. Neural Laplace Control marries a Laplace–domain

dynamics model with model predictive control (MPC) to handle *unknown* delays (Holt et al., 2023a). Complementarily, Active Observing in Continuous-Time Control shows that *when* to measure the state is as critical as *what* to do, proving that irregular sampling can strictly outperform uniform sensing while remaining computationally tractable (Holt et al., 2023b). These ideas resonate with event-triggered control in multi-agent systems (Garcia & Antsaklis, 2012) and with delay-robust *model-based* RL methods, e.g. Delay-Aware Model-Based RL (Chen et al., 2021a). For long horizons, sequence-modelling planners—including Decision Transformer (Chen et al., 2021b) and diffusion-based policy refinement (Chi et al., 2023)—provide strong baselines that can be embedded inside MPC loops, yielding hybrid RL–MPC schemes with favourable stability guarantees (Reiter et al., 2025). While orthogonal to EvoControl, continuous-time control methods naturally connect to high-frequency control—reducing the discretization timestep brings discrete-time methods closer to continuous-time behavior. Future extensions of EvoControl could incorporate these continuous-time approaches, enabling extremely fast-reacting low-level neural policies that outperform traditional MPC or planning methods constrained by computational budget at very high control frequencies.

**Interpretable Dynamics and Generative Simulation:** Replacing black-box predictors with symbolic or sparse representations improves transparency and extrapolation. SINDy uncovers compact governing equations from data (Brunton et al., 2016), and recent work extends this philosophy to causal inference, which learns closed-form differential systems that remain valid under irregular sampling, enabling counterfactual policy evaluation (Kacprzyk et al., 2024). Beyond inference, high-fidelity simulators are indispensable for safe planning. Recent work automates simulator construction by letting a large language model propose causal structure which is then *empirically calibrated* via gradient-free optimisation, achieving robust generalisation beyond historical support (Holt et al., 2025). Such approaches complement physics-grounded generative agents (Battaglia et al., 2018) and underscore the trend toward marrying interpretable models with powerful planners for trustworthy, data-efficient control. In relation to EvoControl, leveraging automated simulator construction enables the generation of diverse, open-ended training environments (Team et al., 2021), facilitating scalable online policy learning. Future versions of EvoControl could integrate these capabilities, significantly enhancing the approach's applicability to more complex, varied tasks and objectives.

## E. Environment Selection and Implementation Details

**Benchmark environments.** We compare against ten standard continuous-control environments (Brockman et al., 2016a; Freeman et al., 2021), and also a safety critical continuous control environment. Specifically we use the continuous control suite from Brax[5] (Freeman et al., 2021), which consists of ten standard continuous control environments, such as locomotion based robot control tasks such as Ant, HalfCheetah and larger state-action space environments such as Humanoid (e.g. controlling a humanoid robot with a state-action dimension of 60 to walk forwards with a given velocity). All the Brax environments are released under the Apache-2.0 license. Furthermore, within this standard suite of tasks is manipulation based environments such as Reacher and Pusher, where pusher is a 7 degree of freedom (DOF) robotic arm, with the task to push a movable object on a table to a desired goal location. Moreover, we also construct a safety inspired environment, adapting a single arm version of the Reacher environment, where introduce a random un-modeled contact that incurs a large negative reward when the robot manipulator collides with the object. To compare the frequency element, we set the frequency of each environment to 500Hz, and then motivated by a low-lever controller running at lower frequency such as 31.25Hz ($G = 16$) (a realistic assumption when involving cameras to determine state), we set this as the low-level frequency.

### E.1. Standard Gym MuJuCo Tasks

We use Brax (Freeman et al., 2021), a differentiable physics engine, which provides efficient implementations of the Ant, HalfCheetah, Hopper, Humanoid, HumanoidStandup, InvertedDoublePendulum, Pusher, Reacher, and Walker2d environments. These environments encompass a range of locomotion and manipulation tasks, providing a diverse testbed for evaluating EvoControl. For each environment, we set the simulation timestep $\Delta_t$ to 0.002 (500Hz operation). High-level policies operate at a frequency of 31.25Hz, achieved by executing each high-level action for $G = 16$ simulation steps. To ensure a fair comparison across different control modes, we remove the action magnitude penalization from the default reward function of each environment. The low-level policy receives the high-level action concatenated to a subset of the environment observation state as its own observation, and the exact input specification for each EvoControl variation is provided in Table 2. This allows the low-level controller to condition its actions on the target specified by the high-level policy. The low-level action space is the same as the high-level action space. All environments have a fixed episode length

---

[5]The Brax continuous control environments are all publicly available from https://github.com/google/brax.

of low-level timesteps of 1,000 environment steps. To increase the realism of the simulation, we run the Brax environments with the backend of MJX, that is a MuJoCo environment in Jax with XLA. This enables us to even modify the MuJoCo xml definition file (to create the Safety Critical Reacher) environment. For all MuJoCo environments, we incorporated fixed PD controllers. We tuned the PD gains for each environment individually. Specifically, we set the proportional gain ($K_p$) to 1.0. This value was chosen as the environments, by default, accept actions with a magnitude of 1, representing a normalized torque input. To determine the optimal derivative gain ($K_d$), we leveraged MuJoCo's dampratio parameter, setting it to 1.0 (critically damped). We then empirically observed the $K_d$ value that corresponds to this dampratio within the simulation. These tuned $K_p$ and $K_d$ values were used consistently throughout our experiments unless explicitly stated otherwise, providing a standardized and well-tuned PD baseline for comparison with EvoControl. This approach ensured that the PD controllers were appropriately configured for each environment's dynamics, providing a strong benchmark for evaluating the performance of learned low-level policies.

### E.2. Reacher 1D

The Reacher 1D environment is a simplified version of the standard Reacher environment. We remove the second arm link, creating a 1DOF task suitable for detailed analysis. The goal is randomly placed within the reachable workspace of the single arm link. The high-level state space consists of the angle and angular velocity of the arm, and the 2D position of the target. The high-level action is the desired angle. The low-level state comprises the high-level state concatenated with the high-level action, and the low-level action is the torque applied to the joint. To ensure reproducibility we provide the full environment MuJoCo xml specification below.

```xml
<mujoco model="reacher_1d">
  <compiler angle="radian" inertiafromgeom="true"/>
  <default>
    <joint armature="1" damping="1.0" limited="true"/>
    <geom conaffinity="0" contype="0" friction="1 0.1 0.1" rgba="0.4 0.33 0.26 1.0"/>
  </default>
  <option gravity="0 0 0" timestep="0.002" />
  <custom>
    <!-- brax custom params -->
    <numeric data="0 0.1 -0.1" name="init_qpos"/>
    <numeric data="1000 1000" name="constraint_stiffness"/>
    <numeric data="1000" name="constraint_limit_stiffness"/>
    <numeric data="3 0.1" name="constraint_vel_damping"/>
    <numeric data="0.1" name="constraint_ang_damping"/>
    <numeric data="0.0" name="ang_damping"/>
    <numeric data="0" name="spring_mass_scale"/>
    <numeric data="1" name="spring_inertia_scale"/>
    <numeric data="5" name="solver_maxls"/>
  </custom>
  <worldbody>
    <light diffuse=".5 .5 .5" pos="0 0 3" dir="0 0 -1"/>
    <!-- Arena -->
    <geom conaffinity="0" contype="0" name="ground" pos="0 0 0" size="1 1 10" type="plane" rgba="1 1 1 1"/>
    <geom conaffinity="0" fromto="-.3 -.3 .01 .3 -.3 .01" name="sideS" size=".02" type="capsule"/>
    <geom conaffinity="0" fromto=" .3 -.3 .01 .3  .3 .01" name="sideE" size=".02" type="capsule"/>
    <geom conaffinity="0" fromto="-.3  .3 .01 .3  .3 .01" name="sideN" size=".02" type="capsule"/>
    <geom conaffinity="0" fromto="-.3 -.3 .01 -.3 .3 .01" name="sideW" size=".02" type="capsule"/>
    <!-- Arm -->
    <geom conaffinity="0" contype="0" fromto="0 0 0 0 0 0.02" name="root" size=".011" type="capsule"/>
    <body name="body0" pos="0 0 0.01">
      <joint axis="0 0 1" limited="true" name="joint0" pos="0 0 0" type="hinge" range="-3.13 3.13"/>
      <geom fromto="0 0 0 0.2 0 0" name="link0" size=".01" type="capsule"/>
      <body name="fingertip" pos="0.11 0 0">
        <geom name="fingertip" pos="0 0 0" size=".01" type="sphere"/>
      </body>
    </body>
    <!-- Target -->
    <body name="target" pos="0 0 0.01">
      <joint armature="0" axis="1 0 0" damping="0" limited="true" name="target_x" pos="0 0 0" range="-.2 .2" stiffness="0" type="slide"/>
      <joint armature="0" axis="0 1 0" damping="0" limited="true" name="target_y" pos="0 0 0" range="-.2 .2" stiffness="0" type="slide"/>
      <geom conaffinity="0" contype="0" name="target" pos="0 0 0" size=".009" type="sphere"/>
    </body>
  </worldbody>
  <actuator>
    <motor ctrllimited="true" ctrlrange="-1.0 1.0" gear="200.0" joint="joint0"/>
  </actuator>
</mujoco>
```

### E.3. Safety Critical Reacher

The Safety Critical Reacher environment builds upon the Reacher 1D environment by introducing a safety aspect. In 25% of the episodes, a randomly positioned obstacle is introduced, which the arm must avoid. A contact force sensor is added to the observations, and a penalty is applied to the reward for any contact force exceeding a threshold. This encourages the development of low-level controllers capable of reacting quickly to avoid collisions. The high-level state space adds a contact force sensor to the Reacher 1D state, while action spaces for both high and low level controllers remain the same as the Reacher 1D environment. This environment directly tests the hypothesis that higher-frequency actions can lead to significantly better performance in safety-critical scenarios, aligning with the intuition presented in Proposition 2.1. The

faster reaction time allowed by a high-frequency low-level controller is crucial for effective collision avoidance. To ensure reproducibility we provide the full environment MuJoCo xml specification below. We also use the following reward:

$$r = -\left\|q_0 - \frac{\pi}{2}\right\| - 3.1415927 \cdot \mathbb{I}(\|f_c\| > 0) \tag{2}$$

where $q_0$ is the joint angle (where $q_0$ indicates the first dimension of $q$ at time $t$), $f_c$ is the contact force between the arm and the obstacle, and $\mathbb{I}(\cdot)$ is an indicator function that equals 1 if the condition inside is true, and 0 otherwise. Where we used a fixed goal location of $\pi/2$, and initial starting state of $q_0 = 0$. To ensure reproducibility we provide the full environment MuJoCo xml specification below.

```xml
<mujoco model="safety_critical_reacher">
  <compiler angle="radian" inertiafromgeom="true"/>
  <default>
    <joint armature="1" damping="1" limited="true"/>
    <geom friction="1 0.1 0.1" rgba="0.4 0.33 0.26 1.0"/>
  </default>
  <option gravity="0 0 0" timestep="0.002" />
  <custom>
    <!-- brax custom params -->
    <numeric data="-1.57 0.11 0.0 -0.3" name="init_qpos"/>
    <numeric data="1000 1000" name="constraint_stiffness"/>
    <numeric data="1000" name="constraint_limit_stiffness"/>
    <numeric data="3 3" name="constraint_vel_damping"/>
    <numeric data="0.1" name="constraint_ang_damping"/>
    <numeric data="0.0" name="ang_damping"/>
    <numeric data="0" name="spring_mass_scale"/>
    <numeric data="1" name="spring_inertia_scale"/>
    <numeric data="5" name="solver_maxls"/>
  </custom>
  <worldbody>
    <light diffuse=".5 .5 .5" pos="0 0 3" dir="0 0 -1"/>
    <!-- Arena -->
    <geom conaffinity="0" contype="0" name="ground" pos="0 0 0" size="1 1 10" type="plane" rgba="1 1 1 1"/>
    <geom conaffinity="0" contype="0" fromto="-.3 -.3 .01 .3 -.3 .01" name="sideS" size=".02" type="capsule"/>
    <geom conaffinity="0" contype="0" fromto=" .3 -.3 .01 .3 .3 .01" name="sideE" size=".02" type="capsule"/>
    <geom conaffinity="0" contype="0" fromto="-.3 .3 .01 .3 .3 .01" name="sideN" size=".02" type="capsule"/>
    <geom conaffinity="0" contype="0" fromto="-.3 -.3 .01 -.3 .3 .01" name="sideW" size=".02" type="capsule"/>
    <!-- Arm -->
    <geom conaffinity="0" contype="0" fromto="0 0 0 0 0 0.02" name="root" size=".011" type="capsule"/>
    <body name="body0" pos="0 0 0.01">
      <joint axis="0 0 1" limited="true" name="joint0" pos="0 0 0" range="-1.570 3.1415" type="hinge"/>
      <geom fromto="0 0 0 0.2 0 0" name="link0" size=".01" type="capsule"/>
      <body name="fingertip" pos="0.11 0 0">
        <geom conaffinity="0" contype="0" name="fingertip" pos="0 0 0" size=".01" type="sphere"/>
      </body>
    </body>
    <!-- Random Collision Capsule -->
    <body name="obstacle-body" pos="0 0 0.01">
      <joint axis="1 0 0" damping="0" limited="true" name="obstacle_x" pos="0 0 0" range="-.2 .2" stiffness="0" type="slide" armature="1e10"/>
      <joint axis="0 1 0" damping="0" limited="true" name="obstacle_y" pos="0 0 0" range="-.2 .2" stiffness="0" type="slide" armature="1e10"/>
      <joint axis="0 0 1" damping="0" limited="true" name="obstacle_z" pos="0 0 0" range="-.3 .3" stiffness="0" type="slide" armature="1e10"/>
      <geom pos="0 0 0" size=".02" fromto="0 0 -0.1 0 0 0.1" type="capsule" name="obstacle"/>
    </body>
  </worldbody>
  <contact>
    <pair geom1="obstacle" geom2="fingertip" condim="1" />
    <pair geom1="obstacle" geom2="link0" condim="1" />
  </contact>
  <actuator>
    <motor ctrllimited="true" ctrlrange="-1.0 1.0" gear="200.0" joint="joint0"/>
  </actuator>
</mujoco>"""
```

## E.4. Safety Critical HalfCheetah

The Safety Critical HalfCheetah environment augments the standard HalfCheetah task with a single, vertically–oriented capsule obstacle that the agent can collide with. At the start of each episode the obstacle is placed directly on the cheetah's trajectory with probability 25%; otherwise it is translated below the arena so that no contact can occur. The observation vector is extended by the scalar magnitude of the measured contact force, while the 6–DoF action space remains unchanged.

Whenever any of the `torso`, `fthigh`, `fshin` or `ffoot` geoms touches the obstacle, a reward of $-200$ is applied. We omit the usual quadratic control cost so that collision-avoidance is the dominant optimisation signal. The per–timestep reward therefore becomes

$$r_t = \underbrace{\frac{x_{t+1} - x_t}{\Delta t}}_{\text{run}} - 200 \cdot \mathbb{I}(\|f_c\| > 0), \tag{3}$$

where $x_t$ is the torso's $x$-position, $\Delta t$ is the physics timestep, $f_c$ is the contact force between the agent and the obstacle, and $\mathbb{I}(\cdot)$ is the indicator function. As argued in Proposition 2.1, this large, impulse–shaped penalty rewards policies that can react at high frequency to avoid dangerous contacts. To ensure reproducibility we provide the full environment MuJoCo xml specification below.

```xml
<mujoco model="cheetah">
```

```xml
<compiler angle="radian" coordinate="local" inertiafromgeom="true" settotalmass="14"/>
<default>
  <joint armature=".1" damping=".01" limited="true" solimplimit="0 .8 .03" solreflimit=".02 1" stiffness="8"/>
  <geom conaffinity="0" condim="3" contype="0" friction=".4 .1 .1" solimp="0.0 0.8 0.01" solref="0.02 1" rgba="0.4 0.33 0.26 1.0"/>
  <motor ctrllimited="true" ctrlrange="-1 1"/>
</default>
<size nstack="300000" nuser_geom="1"/>
<option gravity="0 0 -9.81" timestep="0.002" iterations="4" />
<custom>
  <!-- brax custom params -->
  <numeric data="0 0 0 0 0 0 0 0 0 -10" name="init_qpos"/>
  <numeric data="10000" name="constraint_limit_stiffness"/>
  <numeric data="20000" name="constraint_stiffness"/>
  <numeric data="10 2 2 10 2 1 1" name="constraint_ang_damping"/>
  <numeric data="10" name="constraint_vel_damping"/>
  <numeric data="-0.01" name="ang_damping"/>
  <numeric data="0.2" name="baumgarte_erp"/>
  <numeric data="0.3" name="spring_mass_scale"/>
  <numeric data="0.8" name="spring_inertia_scale"/>
  <numeric data="50" name="solver_maxls"/>
</custom>
<asset>
  <texture builtin="gradient" height="100" rgb1="1 1 1" rgb2="0 0 0" type="skybox" width="100"/>
  <texture builtin="flat" height="1278" mark="cross" markrgb="1 1 1" name="texgeom" random="0.01" rgb1="0.8 0.6 0.4" rgb2="0.8 0.6 0.4" type="cube" width="127"/>
  <texture builtin="checker" height="100" name="texplane" rgb1="0 0 0" rgb2="0.8 0.8 0.8" type="2d" width="100"/>
  <material name="MatPlane" reflectance="0.5" shininess="1" specular="1" texrepeat="60 60" texture="texplane"/>
  <material name="geom" texture="texgeom" texuniform="true"/>
</asset>
<worldbody>
  <light cutoff="100" diffuse="1 1 1" dir="-0 0 -1.3" directional="true" exponent="1" pos="0 0 1.3" specular=".1 .1 .1"/>
  <geom conaffinity="1" condim="3" material="MatPlane" name="floor" pos="0 0 0" size="40 40 40" type="plane" rgba="0.5 0.5 0.5 1.0"/>
  <body name="torso" pos="0 0 .7">
    <camera name="track" mode="trackcom" pos="0 -3 0.3" xyaxes="1 0 0 0 0 1"/>
    <joint armature="0" axis="1 0 0" damping="0" limited="false" name="rootx" pos="0 0 0" stiffness="0" type="slide"/>
    <joint armature="0" axis="0 0 1" damping="0" limited="false" name="rootz" pos="0 0 0" stiffness="0" type="slide"/>
    <joint armature="0" axis="0 1 0" damping="0" limited="false" name="rooty" pos="0 0 0" stiffness="0" type="hinge"/>
    <geom fromto="-.5 0 0 .5 0 0" name="torso" size="0.046" type="capsule"/>
    <geom contype="1" axisangle="0 1 0 .87" name="head" pos=".6 0 .1" size="0.046 .15" type="capsule"/>
    <!-- <site name='tip'  pos='.15 0 .11'/>-->
    <body name="bthigh" pos="-.5 0 0">
      <joint axis="0 1 0" damping="6" name="bthigh" pos="0 0 0" range="-.52 1.05" stiffness="240" type="hinge"/>
      <geom contype="1" axisangle="0 1 0 -3.8" name="bthigh" pos=".1 0 -.13" size="0.046 .145" type="capsule"/>
      <body name="bshin" pos=".16 0 -.25">
        <joint axis="0 1 0" damping="4.5" name="bshin" pos="0 0 0" range="-.785 .785" stiffness="180" type="hinge"/>
        <geom axisangle="0 1 0 -2.03" name="bshin" pos="-.14 0 -.07" size="0.046 .15" type="capsule"/>
        <body name="bfoot" pos="-.28 0 -.14">
          <joint axis="0 1 0" damping="3" name="bfoot" pos="0 0 0" range="-.4 .785" stiffness="120" type="hinge"/>
          <geom contype="1" axisangle="0 1 0 -.27" name="bfoot" pos=".03 0 -.097" size="0.046 .094" type="capsule"/>
        </body>
      </body>
    </body>
    <body name="fthigh" pos=".5 0 0">
      <joint axis="0 1 0" damping="4.5" name="fthigh" pos="0 0 0" range="-1 .7" stiffness="180" type="hinge"/>
      <geom contype="1" axisangle="0 1 0 .52" name="fthigh" pos="-.07 0 -.12" size="0.046 .133" type="capsule"/>
      <body name="fshin" pos="-.14 0 -.24">
        <joint axis="0 1 0" damping="3" name="fshin" pos="0 0 0" range="-1.2 .87" stiffness="120" type="hinge"/>
        <geom axisangle="0 1 0 -.6" name="fshin" pos=".065 0 -.09" size="0.046 .106" type="capsule"/>
        <body name="ffoot" pos=".13 0 -.18">
          <joint axis="0 1 0" damping="1.5" name="ffoot" pos="0 0 0" range="-.5 .5" stiffness="60" type="hinge"/>
          <geom contype="1" axisangle="0 1 0 -.6" name="ffoot" pos=".045 0 -.07" size="0.046 .07" type="capsule"/>
        </body>
      </body>
    </body>
  </body>
  <!-- Random Collision Capsule -->
  <body name="obstacle-body" pos="2.0 0 0.01">
    <joint axis="0 0 1" damping="0" limited="true" name="obstacle_z" pos="0 0 0" range="-10.0 10.0" stiffness="0" type="slide"/>
    <geom pos="0 0 0" size="1.0" fromto="0 0 -5.0 0 0 5.0" type="capsule" name="obstacle"/>
  </body>
</worldbody>
<contact>
  <pair geom1="obstacle" geom2="torso" condim="1" />
  <pair geom1="obstacle" geom2="fshin" condim="1" />
  <pair geom1="obstacle" geom2="ffoot" condim="1" />
  <pair geom1="obstacle" geom2="fthigh" condim="1" />
</contact>
<actuator>
  <motor gear="120" joint="bthigh" name="bthigh"/>
  <motor gear="90" joint="bshin" name="bshin"/>
  <motor gear="60" joint="bfoot" name="bfoot"/>
  <motor gear="120" joint="fthigh" name="fthigh"/>
  <motor gear="60" joint="fshin" name="fshin"/>
  <motor gear="30" joint="ffoot" name="ffoot"/>
</actuator>
</mujoco>
```

### E.5. Low-Level Controller (PD) Parameters

For every benchmark environment we release the exact proportional ($K_p$) and derivative ($K_d$) gains used to initialise the fixed PD controllers[6]. These values were obtained with the critically–damped heuristic described in Appendix E and are reproduced verbatim to enable reproducibility.

**Position-Delta percentage.**  All position-delta controllers use a common relative percentage step size of 15% of the low-level action from the low-level policy multiplied by the joint range.

---

[6]Identical values are used for the initial $\beta_{PD}$ in *EvoControl* (§3); for the annealing schedule see Appendix G.1.

Table 8: PD gains for locomotion tasks (joint order as in the simulator).

| Environment | $K_p$ | $K_d$ |
|---|---|---|
| Ant | [1.0, 1.0, 1.0, 1.0, 1.0, 1.0, 1.0, 1.01] | [0.013503286298071678, 0.013387106015517087, 0.013503286298071678, 0.013387106015517087, 0.013503286298071678, 0.013387106015517087, 0.013503286298071678, 0.013387106015517087] |
| HalfCheetah | [1.0, 1.0, 1.0, 1.0, 1.0, 1.0] | [0.013068637140440764, 0.012039072923754321, 0.011433132942037051, 0.011834836266943663, 0.015236008072342119, 0.022076070073966402] |
| Hopper | [1.0, 1.0, 1.0] | [0.027671617948625667, 0.016434757102351864, 0.010611226997820387] |
| Walker2D | [1.0, 1.0, 1.0, 1.0, 1.0, 1.0] | [0.04316262945905774, 0.021009366320377088, 0.005125036607986897, 0.04316262945905774, 0.021009366320377088, 0.005125036607986897] |
| Humanoid | [1.0 × 17] | [0.06294467526760232, 0.010484943278527236, 0.05253240964007362, 0.029011285582051016, 0.0032262763580096407, 0.009664385891402158, 0.006034450945849556, 0.029011285582051016, 0.0032262763580096407, 0.009668983583956587, 0.006034450945849556, 0.029129550027586302, 0.03411799312574108, 0.019725189664746758, 0.029129550027586305, 0.03411799312574109, 0.01972518966474677] |
| Humanoid Stand-up | [1.0 × 17] | [0.054437005767013026, 0.054733973353533194, 0.014337546020293841, 0.0032262763580096425, 0.02901128558205103, 0.00966438589140216, 0.005682180594001962, 0.0032262763580096425, 0.02901128558205103, 0.009668983583956589, 0.005682180594001962, 0.02912955002758632, 0.03411799312574108, 0.019725189664746755, 0.02912955002758632, 0.03411799312574109, 0.019725189664746758] |
| Inv. Double Pendulum | [1.0] | [0.017375593307861] |
| Inverted Pendulum | [1.0] | [0.07871611563924959] |

Table 9: PD gains for manipulation and reacher-style tasks.

| Environment | $K_p$ | $K_d$ |
|---|---|---|
| Pusher[1] | [1.0] | [0.009740730108475674, 0.007738485648275393, 0.002148051582366496, 0.003025365021942605, 0.0020592919232439886, 0.0020104689753536694, 0.0020340762132783027] |
| Reacher (2-DoF) | [1.0] | [0.01, 0.01] |
| Reacher 1D | [1.01] | [0.1] |
| Safety-Critical Reacher | [1.0] | [0.1] |

## F. Benchmark Method Implementation Details

**Benchmark methods.** We seek to compare against competitive established baselines, using the same high-level PPO policy ($\rho$) learning algorithm with the same high-level architecture across all baselines, varying only the low-level policy ($\beta$). We compare with *fixed controllers* baselines, which are deterministic PD controllers of: PD Position, PD Position Delta, and PD Integrated Velocity (Aljalbout et al., 2024). We also compare against *direct torque control* baselines at both high (500Hz, i.e. the simulation timestep) and low (31.25Hz) frequencies; and a Random policy (31.25Hz). Moreover, we seek to investigate the EvoControl framework, and hence benchmark against several different variations from varying the observation for the low-level policy, from the full-state to a restricted partially observed state (only observing the robot joint positions $q$ or velocities $\dot{q}$), following EvoControl types with their corresponding observations as outlined in Table 2. Additionally, the EvoControl variants using position-based controllers are annealed from their corresponding PD controllers. We provide more detailed implementation information for each benchmark method in the following.

### F.1. High-Level Policy and PPO Implementation

The focus of the paper is on enabling high-frequency control with a learning based method, therefore to provide a thorough competitive implementation of all the benchmark methods we use the same high-level policy neural network architecture and learning algorithm of PPO (Schulman et al., 2017) across all the benchmark methods for all the main results. We did also perform additional ablations of training the high-level policy with ES instead for all the benchmark methods, which can be seen in Appendix J.9.

We use the standard PPO implementation (Schulman et al., 2017), specifically using the implementation from PureJaxRL[7] (Lu et al., 2022), a Jax (Bradbury et al., 2021) implementation of PPO. We used the fixed PPO hyper-parameters from PureJaxRL, which are derived from the PPO continuous-control environment parameters from CleanRL (Huang et al., 2022) which are themselves derived from those from stable baselines (Raffin et al., 2021). These hyper-parameters have been determined to provide good performance across a range of continuous-control environments. These parameters are specifically 'learning_rate'=3e-4,'num_envs'=1024,'num_steps'=10 (number of environment steps per rollout),'total_timesteps'=1e6, 'update_epochs'=4 (number of PPO update epochs per iteration), 'num_minibatches'=8 (number of minibatches for each PPO update),'gamma'=0.99 (discount factor),'gae_lambda'=0.95 (generalized Advantage Estimation parameter),'clip_eps'=0.2,'ent_coef'=0.0,'vf_coef'=0.5, and 'max_grad_norm'=0.5 (gradient clipping threshold).

The PPO implementation uses batched environments for efficient data collection, accumulating 'num_envs' $\times$ 'num_steps' transitions before performing updates. This facilitates parallel environment interaction and accelerated training.

The high-level policy architecture (the same for all benchmark methods) $\rho_\theta$ (with parameters $\theta$) is represented by an actor-critic network implemented using Flax (a Jax based neural network library). Both the actor and critic share a common base network consisting of two hidden layers with 256 units each and tanh activation's (this architecture was initially provided by PureJaxRL to provide effective performance). The actor head outputs the parameters of a multivariate Gaussian distribution (mean and diagonal covariance)—as outlined in Section 3. The critic head outputs a scalar value estimating the state-value function.

---

[7] The PureJaxRL PPO implementation can be found here https://github.com/luchris429/purejaxrl.

## F.2. PD Controller Implementation

We implement standard PD controllers as described in Table 1. For all environments, we tune the PD gains as described in Section E. Briefly, $K_p$ is set to 1.0 and $K_d$ is selected to correspond to a MuJoCo 'dampratio' of 1.0 (critically damped). For the PD Position controller, the high-level action $a_k$ is interpreted as the desired absolute joint position ($q^d$). The PD Delta Position controller interprets $a_k$ as a change in joint position ($\delta q^d$) relative to the joint position at the time of the high-level action $q_k$, such that $q^d = q_k + \delta q^d$. The Integrated Velocity controller interprets $a_k$ as the desired joint velocity and integrates it to obtain a target position. This integration is performed numerically using the trapezoidal rule. These controllers provide a variety of baseline behaviors for comparison.

## F.3. Fixed Controllers

The fixed controllers (PD Position, PD Position Delta, PD Integrated Velocity and PD Position & $K_p$) are implemented as deterministic policies. Given a state and the high-level action $a_k$, they directly compute the low-level control action $u_t$ based on the corresponding control law as described in Table 1. Specifically, in PD Position & $K_p$, the high-level policy outputs both a target position and a $K_p$ gain, which are then used by a fixed PD controller with a fixed derivative gain—this allows the high-level policy to directly control the responsiveness of the low-level controller.

## F.4. Direct Torque Control

For direct torque control, we use two variants: high-frequency (500Hz) and low-frequency (31.25Hz). In the high-frequency variant, the policy operates at the simulation frequency, outputting a torque command at every simulation step. The low-frequency variant operates at the same frequency as the high-level policy in the hierarchical setting. It outputs a torque command every $G = 16$ simulation steps, which is held constant during the intervening steps. Both variants are trained using PPO with the same hyperparameters as the high-level policy, except for the number of environment steps which is adapted based on the direct torque control policy frequency. This allows for a direct comparison of the performance of direct torque control at different frequencies. The same high-level PPO implementation is used to train both the high and low-frequency policies, ensuring that any performance differences are due to the control frequency and not the learning algorithm itself.

# G. EvoControl Implementation Details

In the following we provide implementation details for EvoControl. We used JAX (Bradbury et al., 2021) to implement EvoControl, and present the core training loop in Algorithm 2.

**Network Architectures.** We use the exact same high-level policy architecture and learning algorithm as the baselines use, from Appendix F.1. Therefore the high-level policy $\rho_\theta$ (with parameters $\theta$) is represented by an actor-critic network implemented using Flax. Both the actor and critic share a common base network consisting of two hidden layers with 256 units each and tanh activations. The actor head outputs the parameters of a multivariate Gaussian distribution (mean and diagonal covariance). The critic head outputs a scalar value estimating the state-value function. The low-level policy $\beta_{\phi NN}$ (with parameters $\phi$) is a separate neural network, also implemented using Flax. It consists of three hidden layers with 256 units each and tanh activations. The output layer produces the low-level control actions (torques), $\tau = u_t$ (unless otherwise specified, for example $K_p, K_d$). Specifically the low-level takes as an input observation the EvoControl variant observation, as detailed in Table 2.

Table 10: EvoControl Ablation of PD Controllers.

| Controller Variant | $\beta_{\text{NN}}$ Obs. | $\beta_{\text{NN}}$ action |
|---|---|---|
| EvoControl (Full State) | $s_t, a_k, e_t, q_t, \dot{q}_t, t/T$ | $\tau$ |
| EvoControl (Residual State) | $e_t, t/T$ | $\tau$ |
| EvoControl (Target + Proprioceptive) | $a_k, q_t, \dot{q}_t, e_t, t/T$ | $\tau$ |
| EvoControl (Target) | $a_k, q_t, \dot{q}_t, t/T$ | $\tau$ |
| EvoControl (Learned Gains) | $s_t, a_k, q_t, \dot{q}_t, t/T$ | $K_p, K_d$ |
| EvoControl (Delta Position) | $s_t, a_k, e_t, q_t, \dot{q}_t, t/T$ | $\tau$ |

**Low-level Observation.** For clarity we reproduce Table 2, here as Table 10. Specifically, the observation for the low-level policy can consist of the current state $s_t$, the high-level policy latent action $a_k$, the PD controller error $e_t$ (that is used during the annealing), the robots generalized positions $q_t$, the generalized velocities $\dot{q}_t$, and the ratio of the percentage of the low-level steps that the current high-level action is being followed for—for example with $G = 16$, $T = G = 16$, and hence $t = i$ (Algorithm 1) or the number of low-level steps out of $G$ that the low-level policy is currently on whilst following the high-level policy.

**Annealing Strategy.** The annealing parameter $\alpha$ controls the convex combination of the fixed PD controller ($\beta_{\text{PD}}$) and the learned low-level policy ($\beta_{\phi NN}$). We use a linear decay schedule, starting at $\alpha = 1.0$ and decreasing linearly to $\alpha = 0.0$ over the $K$ training sections, of $\alpha_k = 1 - \frac{k}{K}$, where $k$ is the current training section.

**ES Details.** For ES we use Policy Gradients with Parameter-Based Exploration (PGPE) (Sehnke et al., 2010) algorithm to optimize the low-level policy $\beta_{\phi NN}$. The neural network's parameter vector $\phi$ is directly optimized. We use a population size of es_pop_size $= 512$, and each individual is evaluated over es_rollouts $= 16$ rollouts to estimate its fitness (episodic return $R$). Adam (Kingma & Ba, 2014) is used within PGPE, and we we use the PGPE hyper-parameters of a center learning rate of 0.05 and a standard deviation learning rate of 0.1. We use es_sub_generations $= 8$ generations per training section $k$. The parameter distribution's initial standard deviation is 0.1. We use the implementation of PGPE provided by EvoJax (Tang et al., 2022), in Jax, and their recommended hyper-parameters for PGPE, which were empirically found to work well for continuous control tasks. Furthermore, we set $K = 8$ per 1M high-level $\rho$ steps used to train the high-level policy for, and this was empirically determined to work well in practice.

## G.1. EvoControl PseudoCode

For the following pseudocode; we used the same parameters as described above, specifically setting the total number of training sections to $K = 8$ (per 1M high-level $\rho$ policy steps), and then training the PPO high-level policy for 1M (i.e. 1 Million) high-level steps, therefore performing $N$ PPO updates of $N = \lfloor$ 1e6/(num_envs$\times$num_steps$\times$K)$\rfloor$ = 12. We note that each step of the slow high-level policy when operating with a fast low-level policy is effectively $G = 16$ high-frequency environment timesteps of $\Delta_t$.

---

**Algorithm 2** EvoControl Training

---

**Require:** Environment $f(s_t, u_t)$, reward function $r(s_t, u_t)$, high-level policy $\rho_\theta(s_k)$, initial low-level policy $\beta_{PD}(s_i, a_k)$, total training sections $K$, steps per section $N$, annealing strategy for $\alpha$, ES parameters $\eta$, population size $P$, generations per section $G_{evo}$, rollouts per individual $R_{evo}$.

**Ensure:** Trained high-level policy $\rho_\theta(s_k)$, trained low-level policy $\beta_\phi(s_i, a_k)$.

1: Initialize $\alpha \leftarrow 1.0$
2: Initialize low-level policy $\beta(s_i, a_k) \leftarrow \alpha\beta_{\text{PD}}(s_i, a_k) + (1 - \alpha)\beta_{\phi NN}(s_i, a_k)$
3: Initialize ES strategy (e.g., PGPE) with parameters $\eta$
4: **for** $k = 1$ to $K$ **do**
5:     // Train high-level policy $\rho_\theta$ with PPO
6:     **for** $n = 1$ to $N$ **do**
7:         Collect rollout data using $\rho_\theta$ and $\beta$ (Algorithm 1)
8:         Update $\rho_\theta$ using PPO
9:     **end for**
10:    // Train low-level policy $\beta_\phi$ with ES
11:    **for** $g = 1$ to $G_{evo}$ **do**
12:       $\phi_{\text{pop}} \leftarrow$ Sample $P$ parameter sets from $p_\eta(\phi)$
13:       **for** $p = 1$ to $P$ **do**
14:          $F_p \leftarrow 0$
15:          **for** $r = 1$ to $R_{evo}$ **do**
16:             Collect rollout using $\rho_\theta$ (mode) and $\beta_{\phi_p}$ (Algorithm 1)
17:             $F_p \leftarrow F_p+$ rollout return
18:          **end for**
19:          $F_p \leftarrow F_p/R_{evo}$
20:       **end for**
21:       Update ES parameters $\eta$ using fitness values $F_{1:P}$ (e.g., PGPE update)
22:    **end for**
23:    $\phi \leftarrow$ best performing parameter set from ES
24:    $\beta_{\phi NN}(s_i, a_k) \leftarrow$ neural network with parameters $\phi$
25:    $\beta(s_i, a_k) \leftarrow \alpha\beta_{\text{PD}}(s_i, a_k) + (1 - \alpha)\beta_{\phi NN}(s_i, a_k)$
26:    $\alpha \leftarrow 1 - k/K$
27: **end for**

---

### G.2. Detailed Analysis of EvoControl

This appendix provides a detailed analysis of the EvoControl algorithm, addressing the mathematical setting, assumptions, complexity, and properties as requested.

#### G.2.1. MATHEMATICAL SETTING AND ASSUMPTIONS

EvoControl operates within the standard continuous control Reinforcement Learning (RL) framework. We consider a Markov Decision Process (MDP) defined by the tuple $\mathcal{M} = \langle \mathcal{S}, \mathcal{U}, P, r, \gamma \rangle$, with definitions provided in Appendix A.

**Assumptions:**

- *Markov Property:* The environment dynamics satisfy the Markov property, meaning the next state depends only on the current state and action, not on the history.

- *Stationarity:* The transition probabilities and reward function are stationary (do not change over time).

- *Differentiable Policy:* The high-level policy $\rho_\theta(s_k)$ is parameterized by $\theta$ and is differentiable with respect to $\theta$. This allows for gradient-based optimization.

- *Representable Low-Level Policy:* The low-level policy $\beta_\phi(s_i, a_k)$ can be adequately represented by the chosen neural network architecture with parameters $\phi$.

G.2.2. COMPLEXITY ANALYSIS

**Time Complexity:** The time complexity of EvoControl is dominated by the PPO updates for the high-level policy ($\rho$) and the rollout evaluations for ES of the low-level policy ($\beta$).

*PPO Updates:* The per-update complexity of PPO scales linearly with the number of environment interactions. For $N_\rho$ high-level steps, with 'num_envs' parallel environments running for 'num_steps' steps each, and PPO updates occurring every $K$ training sections, there are $N_\rho/(K \cdot \text{num\_envs} \cdot \text{num\_steps})$ PPO updates. Each PPO high-level environment step involves $G$ low-level environment steps.

*ES Rollouts:* Each training section involves ES of the low-level policy. With a population size of 'es_pop_size', 'es_rollouts' rollouts per individual, and 'es_sub_generations' generations per section, the number of rollouts per section is es_rollouts $\cdot$ es_sub_generations $\cdot$ es_pop_size. Each rollout has 'episode_length' low-level steps.

Let $\mathcal{T}_{\text{env}}$ be the time for a single low-level environment step of duration $\Delta_t$. Then, the total time complexity, *without* parallelization of environment rollouts, is:

$$O\left((N_\rho \cdot G + K \cdot \text{es\_rollouts} \cdot \text{es\_sub\_generations} \cdot \text{es\_pop\_size} \cdot \text{episode\_length}) \cdot \mathcal{T}_{\text{env}}\right)$$

We can also re-express this, as if we train the high-level policy for $N_\rho$ high-level $\rho$ steps, and we train EvoControl with $K = 8$ sections per 1M high-level $\rho$ steps (i.e. $K = (N_\rho \cdot 8)/(1e6)$), then the time complexity can also be expressed as:

$$O((N_\rho \cdot G + \frac{N_\rho \cdot 8}{1e6} \cdot \text{es\_rollouts} \cdot \text{es\_sub\_generations} \cdot \text{es\_pop\_size} \cdot \text{episode\_length}) \cdot \mathcal{T}_{\text{env}})$$

However, both of these time complexity measures are worst case, and do not account for any availability to parallelize environment rollouts, which is common in practice on modern GPUs. If we assume that a user has a GPU/CPU that can parallelize the environment rollouts, then the time complexity can approach:

$$O((N_m \cdot \text{num\_steps} \cdot G + \frac{N_\rho \cdot 8}{1e6} \cdot \text{es\_sub\_generations} \cdot \text{episode\_length}) \cdot \mathcal{T}_{\text{env}})$$

Where $N_m = (N_\rho)/(\text{num\_envs} \cdot \text{num\_steps})$.

We provide thorough additional experiments limiting the computational complexity of the above two approaches, as detailed in Appendix J.4.

**Space Complexity:** The space complexity is primarily determined by the size of the neural networks for the high-level and low-level policies, the size of the PPO buffer, and the population size for ES. It is $O(|\theta| + |\phi| + S_{PPO} + P \cdot |\phi|)$, where $|\theta|$ and $|\phi|$ are the number of parameters in the high-level and low-level policies respectively, and $S_{PPO}$ is the size of the PPO buffer.

### G.3. Computational Considerations

Building on the previous section, environment rollouts can be parallelized on modern GPUs. Specifically num_envs, es_rollouts, es_pop_size can all be parallelized. A benefit of ES here, is that the fitness evaluations (within PGPE) are highly parallelizable. We leverage JAX's 'vmap' function for vectorized rollouts, enabling efficient parallel execution on GPUs or CPUs. This can also be readily further optimized, such as distributing the population across multiple devices, to reduce training time (as ES is a gradient free approach) (Salimans et al., 2017).

Whilst the goal of our work is to provide an initial method that can learn a better low-level controller for use within a high-level policy learning environment, to achieve higher final evaluation reward, we acknowledge that doing so increases computational complexity compared to policy learning with a traditional fixed PD controller. Therefore to investigate, what happens if we make the number of low-level environment steps equivalent we provide a further ablation in this setting in Appendix J.4.

# H. Evaluation Metrics

For each environment, and for each baseline we train the joint policy $\pi$ consisting of a high-level $\rho$ and a low-level policy $\beta$ for the same number of high-level policy $\rho$ steps, here 1M steps. Once the policies have been trained, we perform 128 evaluation rollouts, each with a different random seed and compute the undiscounted cumulative sum of rewards for each rollout, i.e. the return for the episode, where each episode lasts 1,000 environment steps. We repeat training each baseline policy across three random seeds. We quote each result as the mean across it's random seeds and provide the corresponding 95% confidence intervals throughout for all metrics. Specifically we quote the normalized score $\mathcal{R}$ (Yu et al., 2020) of the policy in the environment, averaged over 384 random seeds—normalized to the interval of 0 to 100, where a score of 0 corresponds to a random policy performance, and 100 to an existing fixed controller expert policy—which is whichever non-EvoControl baseline scores the highest evaluation environment return.

All experiments were run on a NVIDIA H100 GPU, with 80GB VRAM with a 40 core CPU with 256 RAM. We detail the hyper-parameters in Appendix F for each benchmark method, and how the hyper-parameters were selected, and their origin of source. We did sweep over the learning rate for PPO with the fixed controller PD position baseline, however found the initial hyper-parameters already provided by prior work (PureJaxRL (Lu et al., 2022), and hence CleanRL (Huang et al., 2022)) to be the most performant, therefore they were kept constant throughout all experiments.

# I. Additional Experimental Setup

## I.1. Efficient Exploration Experimental Setup

To reproduce this experiment, we used the Reacher 1D environment, as detailed in Appendix E.2. Specifically to investigate the efficiency of exploration, we modified the Reacher 1D environment to have a deterministic goal across new random seeds, such that the goal location is $q_{\text{goal}} = \pi/2.0$, and the initial starting position to $q = 0$. This is to ensure that we can correctly measure exploration, otherwise starting in a random state with a random goal, could already explore the state-action space, just through environment resets—whereas the focus of this insight experiment is to compare the methods exploration instead, hence fixing the environment starting state and end goal state. Specifically we run each baseline approach for 10,240 low-level environment steps each, and collect the state throughout training for these initial steps. We then process the state collected, and plot the state visitation histograms, as shown in Figure 2.

## I.2. High-frequency Interaction Control in Safety Critical Reacher

To reproduce this experiment, we follow the same setup for the Safety Critical Reacher environment, as detailed in Appendix E.3.

# J. Additional Experiments

## J.1. EvoControl Supports Other Competitive ES Approaches

EvoControl supports using other competitive ES approaches to optimize the low-level policy neural network parameters. We provide full results for diverse competitive ES approaches, where for each ES approach we replicate our full main-table of results presented in the main paper. We provide this analysis empirically, as in Table 11, then follow with a discussion about which ES a user can select to use within the EvoControl framework.

We benchmarked against the following ES approaches of:

- **PGPE (Policy Gradients with Parameter-Based Exploration (Sehnke et al., 2010)):** A policy search algorithm that optimizes neural network parameters by maintaining a probability distribution over them and estimating gradients directly in parameter space. Unlike traditional policy gradient methods, which suffer from high variance due to per-timestep action sampling, PGPE samples policy parameters once per episode, ensuring deterministic rollouts within each trajectory. This leads to significantly reduced gradient variance and improved learning stability, particularly in reinforcement learning tasks with long time horizons and continuous action spaces. PGPE also supports symmetric sampling, which further refines gradient estimates by leveraging paired perturbations, akin to central difference approximations in finite difference methods. These properties make PGPE particularly effective for optimizing neural network policies in complex, high-dimensional control problems.

Table 11: EvoControl benchmarked with other ES approaches. Normalized evaluation return $\mathcal{R}$ for the benchmark methods, across each environment. EvoControl on average achieves a higher normalized evaluation return than the baselines of *fixed controllers* and *direct torque control*. Results are averaged over 384 random seeds, with ± indicating 95% confidence intervals. Returns are normalized to a 0-100 scale, where 0 represents a random policy, and 100 represents the highest reward achieved by a non-EvoControl baseline in each environment. Scores bolded are greater than 100.

| Same PPO high-level alg. $\rho$ with a Low-Level Policy $\beta$ of | Ant | Halfcheetah | Hopper | Humanoid | Humanoid Standup | Inverted Double Pendulum | Inverted Pendulum | Pusher | Reacher | Reacher 1D | Walker2D |
|---|---|---|---|---|---|---|---|---|---|---|---|
| | $\mathcal{R}\uparrow$ | $\mathcal{R}\uparrow$ | $\mathcal{R}\uparrow$ | $\mathcal{R}\uparrow$ | $\mathcal{R}\uparrow$ | $\mathcal{R}\uparrow$ | $\mathcal{R}\uparrow$ | $\mathcal{R}\uparrow$ | $\mathcal{R}\uparrow$ | $\mathcal{R}\uparrow$ | $\mathcal{R}\uparrow$ |
| *Fixed Cont.* - PD Position | 100±6.56 | 61.2±0.441 | 91.6±1.23 | 100±2.96 | 100±0.974 | 99.9±0.03 | 100±2.86e-15 | 100±8.47 | 100±1.8 | 85.2±2.87 | 75.7±0.633 |
| *Fixed Cont.* - PD Position Delta | 2.4±1.91 | 2.76±0.0888 | 100±1.35 | 96.6±1.71 | 2.96±0.0397 | 53.8±1.57 | 100±2.86e-15 | 0±0 | 40.9±3.23 | 15.2±7.6 | 90.2±0.239 |
| *Fixed Cont.* - PD Int. Velocity | 3.59±1.78 | 2.46±0.0932 | 74.7±0.903 | 83.4±1.13 | 0±0 | 49.7±1.55 | 86.5±2 | 0±0 | 0±0 | 0±0 | 85.9±2.55 |
| *Fixed Cont.* - Random | 0.0±0.0 | 0.0±0.0 | 0.0±0.0 | 0.0±0.0 | 0.0±0.0 | 0.0±0.0 | 0.0±0.0 | 0.0±0.0 | 0.0±0.0 | 0.0±0.0 | 0.0±0.0 |
| *Direct Torque Cont.* - High Freq. (500Hz) | 0±0 | 17.2±0.316 | 1.42±0.533 | 10.4±2.19 | 10.3±0.586 | 0±0 | 0±0 | 1.34±7.89 | 2.08±5.84 | 45.3±6.74 | 0±0 |
| *Direct Torque Cont.* - Low Freq. (31.25Hz) | 54.5±7.15 | 100±1.21 | 72±0.64 | 98±2.55 | 80.6±2.56 | 100±0.0311 | 100±2.86e-15 | 73.2±12.9 | 59.2±3.72 | 100±1.94 | 100±2.68 |
| **PGPE - EvoControl (Full State)** | **368±10.6** | **157±1.1** | **274±1.52** | **123±2.7** | **116±0.609** | **101±0.0487** | 100±2.86e-15 | **362±11.1** | **114±0.973** | **106±0.936** | **203±4.61** |
| **PGPE - EvoControl (Residual State)** | **182±8.58** | **182±1.02** | **101±0.53** | **170±1.14** | **212±4.95** | 99.2±0.054 | 100±2.86e-15 | **375±10.7** | **106±1.29** | **104±1.19** | **205±2.72** |
| **PGPE - EvoControl (Target + Proprio.)** | **319±14.1** | **168±1.41** | **171±5.28** | **165±1.77** | **165±4.94** | 99.7±0.0417 | 100±2.86e-15 | **353±10.5** | 96.8±3.54 | **105±0.776** | **178±2.31** |
| **PGPE - EvoControl (Target)** | **293±13.2** | **162±1.58** | **283±1.91** | **164±1.8** | **205±5.04** | 99.6±0.0377 | 100±2.86e-15 | **353±9.56** | **112±0.785** | **105±0.78** | **188±3.02** |
| **PGPE - EvoControl (Learned Gains)** | **266±14.1** | **113±1.6** | **206±10** | **150±2.55** | **117±0.205** | 99.5±0.0947 | 100±2.86e-15 | **330±10.5** | **116±0.747** | **105±1.21** | **196±4.5** |
| **PGPE - EvoControl (Delta Position)** | **362±12.8** | **133±1.82** | **225±2.99** | **119±2.78** | **105±0.285** | **101±0.0364** | 100±2.86e-15 | **267±11.6** | 65.5±3.71 | 99.1±2.44 | **183±2.36** |
| **CMA-ES - EvoControl (Full State)** | 97.8±61.8 | **104±15.9** | **139±0.484** | 93.5±12.9 | 78.4±7.85 | **100±0.11** | 100±0 | **263±52.2** | **109±2.34** | **106±3.64** | **131±2.25** |
| **CMA-ES - EvoControl (Residual State)** | 37.2±82.9 | **102±10.2** | **201±20.1** | 82±13.9 | 88.8±0.162 | **100±0.244** | 100±0 | **289±56.2** | **112±4.21** | **106±3.87** | **215±11.3** |
| **CMA-ES - EvoControl (Target + Proprio.)** | 62.6±50.4 | 89.5±16.4 | **261±11.6** | 74.7±13.4 | 99.7±4.94 | 99.9±0.261 | 100±0 | **318±41.4** | **115±2.93** | **106±3.66** | **201±2.56** |
| **CMA-ES - EvoControl (Target)** | 72.3±50.7 | 98.5±12.6 | **212±2.77** | 78.9±7.23 | 85.9±5.98 | 99.1±0.0161 | 100±0 | **307±50.1** | **115±5.12** | **106±4** | **209±2.1** |
| **CMA-ES - EvoControl (Learned Gains)** | 42.7±53.3 | 54.4±13.4 | **156±1.75** | **129±16.2** | **116±0.579** | 99.6±0.629 | 100±0 | **319±43.6** | **111±5.41** | **103±5.09** | **166±17.2** |
| **CMA-ES - EvoControl (Delta Position)** | 31.6±52.7 | 85.4±9.41 | **128±2.29** | 97.8±13.7 | **104±3.27** | **100±0.228** | 100±0 | **304±54.8** | 70.8±18 | 84.5±20.7 | **135±1.43** |
| **SEP-CMA-ES - EvoControl (Full State)** | 44.8±61.3 | 85.8±17.6 | **108±0.305** | 71.5±13.5 | **103±0.567** | **100±0.126** | 100±0 | **281±46.4** | **115±3.64** | **106±3.67** | **150±6.6** |
| **SEP-CMA-ES - EvoControl (Residual State)** | 91.8±75.9 | **128±4.15** | **246±23.4** | **122±13.6** | **108±2.06** | 99.6±0.16 | 100±0 | **305±34.4** | **112±2.78** | **106±4.28** | **171±22.6** |
| **SEP-CMA-ES - EvoControl (Target + Proprio.)** | 52.3±37.4 | 94.8±19.1 | **126±0.925** | 84.6±20 | **107±3.32** | 99±0.0269 | 100±0 | **298±54.3** | **115±2.31** | **106±3.74** | **195±1.88** |
| **SEP-CMA-ES - EvoControl (Target)** | 96.7±61.6 | 92.6±21.8 | **121±0.5** | 83±21.7 | **104±4.17** | 98.7±0.0231 | 100±0 | **271±39.1** | **112±5.76** | **106±3.65** | **196±4** |
| **SEP-CMA-ES - EvoControl (Learned Gains)** | 93.6±51.5 | 73±7.49 | **159±2.52** | **107±11.9** | **116±3.22** | **100±0.137** | 100±0 | **320±43.8** | **114±5.71** | **104±6.3** | **151±13.6** |
| **SEP-CMA-ES - EvoControl (Delta Position)** | 26.6±44.6 | 54±8.43 | **125±1.79** | 81.5±8.3 | **100±3.53** | 99.9±0.0987 | 100±0 | **252±48.4** | 63.2±18.6 | 85.5±29.8 | **176±13.5** |
| **CMA-ES-JAX - EvoControl (Full State)** | 82.9±57.7 | 98.3±16.2 | **113±0.0963** | 70.1±5.5 | 98.4±1.59 | **100±0.157** | 100±0 | **332±50.8** | **113±3.82** | **106±4.34** | **222±10.1** |
| **CMA-ES-JAX - EvoControl (Residual State)** | **161±70.7** | 99.8±3.83 | **148±57.2** | **114±16.7** | 98.5±0.926 | **100±0.156** | 100±0 | **237±54.1** | **111±6.67** | **106±4.1** | **183±5.28** |
| **CMA-ES-JAX - EvoControl (Target + Proprio.)** | 85.5±49.3 | 77.7±11.5 | **128±3.57** | 88.5±14.8 | 98.1±5.61 | 99.3±0.0247 | 100±0 | **312±57.7** | **112±3.12** | **106±3.88** | **175±7.15** |
| **CMA-ES-JAX - EvoControl (Target)** | **128±57.5** | 80±18.2 | **126±1.16** | 61.7±11 | 93.7±2.07 | 99±0.00698 | 100±0 | **248±40.3** | **110±3.44** | **106±3.8** | **152±2.42** |
| **CMA-ES-JAX - EvoControl (Learned Gains)** | 63.8±40.1 | 72.8±10.3 | **187±0.624** | **134±14.5** | **110±3.01** | **100±0.101** | 100±0 | **352±43.4** | **105±9.81** | **105±4.48** | **136±11.4** |
| **CMA-ES-JAX - EvoControl (Delta Position)** | 77±63.2 | 88±12.8 | **107±0.122** | 64.3±10 | 96.4±0.213 | 99.9±0.12 | 100±0 | **317±42.7** | 67.2±15.2 | 87.1±23.7 | **200±5.46** |
| **DE - EvoControl (Full State)** | 3.53±33.1 | 5.94±7.42 | 22.4±0.326 | 39.4±1.93 | 27.4±4.33 | 78.8±6.36 | 0±0 | **234±50.5** | **102±10.5** | 74.2±12.7 | 0±0 |
| **DE - EvoControl (Residual State)** | 7.54±5.79 | 23.3±5.98 | 0±0 | **142±11** | 7.71±0.364 | 45.7±18.2 | 0±0 | **235±55.8** | **107±7.37** | **106±3.87** | 1.37±0.158 |
| **DE - EvoControl (Target + Proprio.)** | 18.6±34.9 | 32.9±20.7 | 0±0 | 99.7±16.5 | 47.2±9.07 | **100±0.142** | 29.7±12.3 | **321±49.8** | 57.3±19.7 | **106±3.92** | 0±0 |
| **DE - EvoControl (Target)** | 0±0 | 23.3±6.18 | 0±0 | 20.9±6 | 2.35±0.652 | **100±0.0468** | 92.1±3.02 | **319±47.8** | 64.8±17.5 | **106±5.11** | 0±0 |
| **DE - EvoControl (Learned Gains)** | 0±0 | 22.1±8.32 | 40.2±6.75 | 0±0 | 22.1±5.02 | 52±2.83 | 100±0 | **220±49.7** | 53.7±13.4 | 59.1±15 | 1.08±2.76 |
| **DE - EvoControl (Delta Position)** | 0±0 | 46.2±10 | 41.3±0.693 | 48.8±1.46 | 65.7±10.7 | **100±2.2** | 99.8±0 | **278±41.7** | 54.1±18.8 | 93.7±19.4 | 0±0 |
| **iAMaLGaM - EvoControl (Full State)** | 64.6±51.4 | 75.2±8.66 | 96.2±0.122 | **109±15.8** | 94.5±1.35 | **100±0.0881** | 100±0 | **351±49.5** | 95.5±4.55 | **106±4.1** | **135±9.64** |
| **iAMaLGaM - EvoControl (Residual State)** | **110±60.3** | **101±9.38** | 58.8±0.524 | 99.4±14.3 | 96.2±0.993 | 99.4±0.162 | 100±0 | **333±52.5** | **109±3.49** | **106±3.6** | **161±10.4** |
| **iAMaLGaM - EvoControl (Target + Proprio.)** | 54.3±54.2 | 87.6±16.6 | 93±0.293 | **110±12.4** | **105±0.427** | 99.4±0.117 | 100±0 | **327±40** | 63±19.1 | **106±3.82** | **140±18.2** |
| **iAMaLGaM - EvoControl (Target)** | 47.2±57.9 | **133±8.06** | **140±5.71** | 90.7±10.9 | 98.5±5.56 | **100±0.148** | 100±0 | **299±44.7** | 59.6±17.9 | **104±11** | **101±2.15** |
| **iAMaLGaM - EvoControl (Learned Gains)** | 5.79±60.1 | 53.6±8.81 | **125±8.93** | **104±11.7** | **111±1.32** | **100±0.108** | 100±0 | **285±50.6** | **114±8.62** | **102±8.91** | 93.8±8.74 |
| **iAMaLGaM - EvoControl (Delta Position)** | 13.8±49.3 | 78.6±7.19 | **121±2.22** | 74±14.1 | 95.2±1.05 | **102±0.36** | 100±0 | **300±72.6** | 54.3±19.4 | **104±10.9** | **116±7.23** |
| **CR-FM-NES - EvoControl (Full State)** | 85.5±54.1 | 81±8.73 | 96.2±0.122 | 66.9±13.3 | 91.3±0.858 | **100±0.0881** | 100±0 | **351±49.5** | 95.5±4.55 | **106±4.1** | **123±4.44** |
| **CR-FM-NES - EvoControl (Residual State)** | 85.9±61.2 | **124±13.5** | 58.8±0.524 | **129±10.7** | 98.4±3.56 | 99.4±0.162 | 100±0 | **328±45.8** | **111±4.54** | **106±3.6** | **184±7.76** |
| **CR-FM-NES - EvoControl (Target + Proprio.)** | 61±48.2 | 81±15 | 93±0.293 | 97.1±18.9 | **106±1.11** | 99.4±0.117 | 100±0 | **327±40** | 63±19.1 | **106±3.82** | **110±11.4** |
| **CR-FM-NES - EvoControl (Target)** | 67±52.9 | 75.8±12.2 | **140±5.71** | **113±13** | **118±3.46** | **100±0.148** | 100±0 | **299±44.7** | 59.6±17.9 | **107±4.03** | **113±4.1** |
| **CR-FM-NES - EvoControl (Learned Gains)** | 45.9±67.2 | 58.7±6.94 | **125±8.93** | **120±14.3** | 97.9±3.28 | **100±0.108** | 100±0 | **313±44.3** | **114±5.33** | **102±8.91** | **106±6.75** |
| **CR-FM-NES - EvoControl (Delta Position)** | 85.3±54.4 | 66.4±14.7 | **121±2.22** | 89±10.3 | 86.9±0.435 | **102±0.36** | 100±0 | **300±72.6** | 54.3±19.4 | 95.6±31.1 | **120±5.53** |
| **OpenES - EvoControl (Full State)** | 99.2±55.1 | 10.4±3.53 | 46.7±0.295 | 68.2±3.38 | 61.7±0.316 | 43.1±3.24 | 100±0 | **204±48.8** | **113±6.44** | 98.4±10.3 | 35.6±20.3 |
| **OpenES - EvoControl (Residual State)** | 15.6±28.8 | 46.8±5.83 | 0±0 | **115±8.02** | 64.5±0.194 | 0±0 | 100±0 | **206±48** | **114±4.66** | **103±6.33** | 0±0 |
| **OpenES - EvoControl (Target + Proprio.)** | 36.3±52.8 | 30.7±8.83 | 87.2±0.144 | 87±3.14 | 81.2±0.335 | 0±0 | 100±0 | **214±46.9** | **113±7** | **105±5.8** | 89.2±1.26 |
| **OpenES - EvoControl (Target)** | 20.7±63 | 0±0 | 13±0.0809 | 42.1±2.17 | 59.6±0.651 | 0±0 | 100±0 | **203±47.6** | 85.1±9.9 | 35.7±24.2 | 30.3±1.31 |
| **OpenES - EvoControl (Learned Gains)** | 4.79±53.4 | 18.1±15.9 | 84.9±0.104 | 3.4±4.48 | 92.7±5.65 | **100±0.0948** | 100±0 | **229±48.2** | **104±13.2** | 45.5±15.4 | 31.3±6.6 |
| **OpenES - EvoControl (Delta Position)** | 22.1±30.7 | 6.22±0.488 | 9.28±0.156 | 76.7±3.63 | 73.6±0.162 | 72.9±9.62 | 100±0 | **220±60.5** | 56±20.1 | 0±0 | 11.7±6.91 |
| *PGPE - Direct Torque Cont.* - High Freq. (500Hz) | **106±31.8** | 94±7.51 | **138±4.82** | 95.4±6.46 | 88.5±3.53 | **100±1.34** | 100±0 | **302±21.3** | 57.8±12.1 | 8.98±14 | **126±3.66** |
| *CMA-ES - Direct Torque Cont.* - High Freq. (500Hz) | 70.4±29.1 | 80.6±6.53 | **118±3.48** | **100±7.04** | 95.6±3.73 | 99.7±1.17 | 100±0 | **268±27** | 55±11.8 | 0±0 | **111±2.74** |
| *SEP-CMA-ES - Direct Torque Cont.* - High Freq. (500Hz) | 50.6±30.8 | 88.4±10.4 | **102±9.51** | **110±8.82** | 85.3±3.82 | 84.7±5.96 | 100±0 | **273±22.9** | 55.8±11.8 | 0±0 | **110±3.85** |
| *DE - Direct Torque Cont.* - High Freq. (500Hz) | 0±0 | 57.3±15 | **114±5.35** | 62.5±7.87 | 79.4±4.41 | 43.4±8.48 | 57.9±17.5 | **288±26.9** | 57.2±11.7 | 20.9±19.8 | 87.9±11.6 |
| *iAMaLGaM - Direct Torque Cont.* - High Freq. (500Hz) | 82.8±29.4 | 87.7±10.1 | 77±4.58 | 70±7.58 | 90.9±1.36 | 83.9±4.67 | 100±0 | **266±25** | 57.6±11.8 | 0±0 | **114±7.32** |
| *OpenES - Direct Torque Cont.* - High Freq. (500Hz) | 17.9±11 | 12±1.74 | 9.4±0.445 | 41.8±4.51 | 42.4±10.9 | 24.1±1.78 | 6.86±2.34 | **112±25.6** | 25.1±16 | 0±0 | 59.6±11.5 |

- **CMA-ES (Covariance Matrix Adaptation Evolution Strategy ([Hansen & Ostermeier, 2001](#)))**: A widely used, competitive ES algorithm known for its efficiency in high-dimensional continuous optimization. It adapts the covariance matrix of the search distribution, guiding the exploration towards promising directions in the parameter space.

- **SEP-CMA-ES (Separable CMA-ES ([Ros & Hansen, 2008](#)))**: A variant of CMA-ES that utilizes a separable covariance matrix, reducing the computational complexity for high-dimensional problems.

- **DE (Differential Evolution ([Storn & Price, 1997](#)))**: Another popular ES algorithm that relies on vector differences

between population members to generate new candidate solutions. It is known for its robustness and ability to handle complex, multimodal objective functions.

- **iAMaLGaM (incremental AMaLGaM (Bosman et al., 2013)):** A JAX-based, parallelized variant of AMaLGaM that builds its probabilistic model incrementally for increased sample efficiency.

- **CR-FM-NES (Cross-Entropy Fitness Model based Natural Evolution Strategies (Nomura & Ono, 2022)):** An ES algorithm that builds a surrogate fitness model to improve the sample efficiency of the search process.

- **OpenES (OpenAI Evolution Strategies (Salimans et al., 2017)):** A simple, parallelizable ES algorithm that utilizes isotropic Gaussian mutations for exploration.

When evaluating these different ES approaches within EvoControl we used the open source implmentations from EvoJax (Tang et al., 2022) and EvoSax (Lange, 2023). Specifically using their respective defined implementations, and using the same broad ES hyper-parameters across all implementations as defined in Appendix G.

We empirically observe in Table 11 that EvoControl can support competitive ES approaches. Specifically PGPE achieves competitive performance within the EvoControl framework across all of the evaluated environments. Interestingly, within the EvoJax ES library PGPE achieves the highest evaluation reward in the environments that EvoJax benchmarked against[8] (Tang et al., 2022), which correlates with our empirical findings.

In practice EvoControl can be used with other ES approaches, however we select PGPE for the main table of results presented in the main paper, due to its good emprical performance (Table 11). Additioanlly, PGPE is computationally efficient and well-suited for large-scale, high-dimensional optimization problems typical in policy search for reinforcement learning. PGPE is straightforward to implementat and scales across multiple compute nodes, leveraging modern accelerators. Furthermore, PGPE has demonstrated robust performance in tasks requiring fine-grained control, making it a suitable choice for optimizing high-frequency low-level policies. Interestingly, more complicated ES algorithms like CMA-ES and DE offer sophisticated search mechanisms, they come with increased computational complexity, such as: CMA-ES: Adapts the covariance matrix but becomes computationally intensive in high-dimensional parameter spaces due to large covariance matrices; DE: Effective in continuous optimization but may struggle with the noisy and dynamic nature of reinforcement learning environments; Advanced ES Algorithms: Methods like iAMaLGaM and CR-FM-NES introduce additional computational overhead without consistently yielding better performance in our context. Our experiments show that while complex ES algorithms can perform well in certain tasks, PGPE offers a favorable balance between performance, computational cost, and implementation within the EvoControl framework.

### J.2. Ablation: Using PPO to Train the Lower-level Policy

We performed an additional ablation experiment, by training the low-level policy with PPO rather than ES. To be comparable we used the same architecture that our existing high-level PPO agent uses, as described in Section 3, and Appendix G. We follow the same setup, of training the high-level policy for 1M high-level environment steps, and now train the low-level policy for the same 1M high-level steps, now training for the low-level for $1M \times G = 16M$ low-level environment steps—to give this ablation the most competitive performance comparison to EvoControl and the non-EvoControl baselines. We perform a complete re-run across all environments as presented in the main paper main results table. The ablation with PPO training the lower-level policy can be seen in Table 12. We observe that using PPO to train the lower-level policy within this EvoControl ablation performs worse (achieves a lower average evaluation return) than using ES to train the lower-level policy $\beta$—thus justifying the use of Neurevolution for training the low-level policy.

### J.3. EvoControl Outperforms Direct Torque Control at High-frequency

In the following we provide empirical evidence for EvoControl outperforming the baseline of a high-frequency low-level direct torque control policy. To address any sample complexity concerns, we also find when we limit EvoControl to use the same computational complexity as all baselines, EvoControl still outperforms the baselines, which is evaluated in detail in Appendix J.4. To provide a thorough analysis of the ability to learn a high-frequency low-level direct torque control policy, we performed additional experiments of training the *Direct Torque Cont. - High Freq. (500Hz)* baseline for an

---

[8]The ES approach benchmark results for EvoJax are provided here: https://github.com/google/evojax/tree/main/scripts/benchmarks.

Table 12: Ablation. Training the lower-level policy with PPO instead of ES, training both the high-level policy and the low-level policy for 1M high-level environment steps each, to produce a competitive ablation. Normalized evaluation return $\mathcal{R}$ for the benchmark methods, across each environment. EvoControl on average achieves a higher normalized evaluation return than the baselines of *fixed controllers* and *direct torque control*. Results are averaged over 384 random seeds, with $\pm$ indicating 95% confidence intervals. Returns are normalized to a 0-100 scale, where 0 represents a random policy, and 100 represents the highest reward achieved by a non-EvoControl baseline in each environment. Scores bolded are greater than 100.

| Method Name | High-level $\rho$ with | Low-level $\beta$ with | Ant | Halfcheetah | Hopper | Humanoid | Humanoid Standup | Inverted Double Pendulum | Inverted Pendulum | Pusher | Reacher | Reacher 1D | Walker2D |
| --- | --- | --- | --- | --- | --- | --- | --- | --- | --- | --- | --- | --- | --- |
| | | | $\mathcal{R}\uparrow$ | $\mathcal{R}\uparrow$ | $\mathcal{R}\uparrow$ | $\mathcal{R}\uparrow$ | $\mathcal{R}\uparrow$ | $\mathcal{R}\uparrow$ | $\mathcal{R}\uparrow$ | $\mathcal{R}\uparrow$ | $\mathcal{R}\uparrow$ | $\mathcal{R}\uparrow$ | $\mathcal{R}\uparrow$ |
| *Fixed Cont. - PD Position* | PPO | PD Position | 100±6.56 | 61.2±0.441 | 91.6±1.23 | 100±2.96 | 100±0.974 | 99.9±0.03 | 100±1.53e-06 | 100±8.47 | 100±1.8 | 85.2±2.87 | 75.7±0.633 |
| *Fixed Cont. - PD Position Delta* | PPO | PD Position Delta | 2.4±1.91 | 2.76±0.0888 | 100±1.35 | 96.6±1.71 | 2.96±0.0397 | 53.8±1.57 | 100±1.53e-06 | 0±0 | 40.9±3.23 | 15.2±7.6 | 90.2±0.239 |
| *Fixed Cont. - PD Int. Velocity* | PPO | PD Int. Velocity | 3.59±1.78 | 2.46±0.0932 | 74.7±0.903 | 83.4±1.13 | 0±0 | 49.7±1.55 | 86.5±2 | 0±0 | 0±0 | 0±0 | 85.9±2.55 |
| *Fixed Cont. - Random* | Random | Direct Torque | 0.0±0.0 | 0.0±0.0 | 0.0±0.0 | 0.0±0.0 | 0.0±0.0 | 0.0±0.0 | 0.0±0.0 | 0.0±0.0 | 0.0±0.0 | 0.0±0.0 | 0.0±0.0 |
| *Direct Torque Cont. - High Freq. (500Hz)* | PPO | Direct Torque | 0±0 | 17.2±0.316 | 1.42±0.533 | 10.4±2.19 | 10.3±0.586 | 0±0 | 0±0 | 1.34±7.89 | 2.08±5.84 | 45.3±6.74 | 0±0 |
| *Direct Torque Cont. - Low Freq. (31.25Hz)* | PPO | Direct Torque | 54.5±7.15 | 100±1.21 | 72±0.64 | 98±2.55 | 80.6±2.56 | 100±0.0311 | 100±1.53e-06 | 73.2±12.9 | 59.2±3.72 | 100±1.94 | 100±2.68 |
| **Ablation: EvoControl (Full State)** | PPO | PPO | 16.4±59.6 | 25.5±53.6 | **102±55.7** | **142±20.3** | 82±78.8 | 83.8±40 | 69.4±132 | **105±209** | 55±27.1 | 80.1±31.5 | 82.4±9.64 |
| **Ablation: EvoControl (Residual State)** | PPO | PPO | 12±49.7 | 41.3±69.2 | 68.4±102 | 60.2±121 | 33.9±100 | 84.6±64.8 | 100±0 | 94.7±351 | 91.4±12.9 | **101±7.8** | **103±80.2** |
| **Ablation: EvoControl (Target + Proprio.)** | PPO | PPO | 0±0 | 25.5±34.6 | 54.1±72.9 | 56.6±34.2 | 38.5±52.6 | 34.6±138 | 29.8±22.3 | 91.3±15.3 | 44.7±44.8 | 62.3±85.1 | 20.8±119 |
| **Ablation: EvoControl (Target)** | PPO | PPO | 0±0 | 17.8±16.7 | 88.2±3.09 | 69.8±76.3 | 45±40.6 | 13.5±119 | 27.8±121 | **132±146** | 59.3±35.6 | 15.5±33.5 | 95.1±43.2 |
| **Ablation: EvoControl (Delta Position)** | PPO | PPO | 0±0 | 2.88±1.86 | 89±16.9 | 92.5±10.2 | 51.2±58.1 | 98.9±2.64 | 81.2±81 | 0±0 | 43.9±10.7 | 5.07±10.6 | 61.6±48.6 |

increasing number of high-level $\rho$ policy steps. Specifically, as tabulated in Table 13, we train for a larger number of $\rho$ steps, significantly greater than all the baselines were trained for (which is 1M $\rho$ steps)—here being from 1M $\rho$ steps to 10B $\rho$ steps. We observe that even with more high-level $\rho$ steps, which corresponds to significantly more low-level environment steps than that used in EvoControl, *Direct Torque Cont. - High Freq. (500Hz)* still on average achieves a lower normalized return compared to EvoControl. This could suggest that direct high-frequency control with PPO produces policies that get stuck in local minima, and fail to find a better performing global policy at high-frequency as EvoControl is able to do—leveraging ES for learning the lower-level high-frequency policy.

Table 13: Additional Experiment. Training *Direct Torque Cont. - High Freq. (500Hz)* baseline for an increased number of high-level $\rho$ policy steps—from from 1M $\rho$ steps to 10B $\rho$ steps. Normalized evaluation return $\mathcal{R}$ for the baseline, across each environment. Results are averaged over 384 random seeds, with $\pm$ indicating 95% confidence intervals. Returns are normalized to a 0-100 scale, where 0 represents a random policy, and 100 represents the highest reward achieved by a non-EvoControl baseline in each environment—using the normalization from the main table of results, Table 3.

| Same PPO high-level alg. $\rho$ with a Low-Level Policy $\beta$ of | Ant | Halfcheetah | Hopper | Humanoid | Humanoid Standup | Inverted Double Pendulum | Inverted Pendulum | Pusher | Reacher | Reacher 1D | Walker2D |
| --- | --- | --- | --- | --- | --- | --- | --- | --- | --- | --- | --- |
| | $\mathcal{R}\uparrow$ | $\mathcal{R}\uparrow$ | $\mathcal{R}\uparrow$ | $\mathcal{R}\uparrow$ | $\mathcal{R}\uparrow$ | $\mathcal{R}\uparrow$ | $\mathcal{R}\uparrow$ | $\mathcal{R}\uparrow$ | $\mathcal{R}\uparrow$ | $\mathcal{R}\uparrow$ | $\mathcal{R}\uparrow$ |
| 1,000,000 Train $\rho$ steps *Direct Torque Cont. - High Freq. (500Hz)* | 0±0 | 17.2±0.336 | 3.63±0.259 | 15.6±3.64 | 17±1.2 | 0±0 | 0±0 | 2.95±7.81 | 0±0 | 45.3±6.74 | 0±0 |
| 10,000,000 Train $\rho$ steps *Direct Torque Cont. - High Freq. (500Hz)* | 6.3±14.9 | 63±1.43 | 93.5±2.82 | 87.9±3.41 | 78.3±2 | 97.9±0.148 | 42.6±1.48 | 21.9±9.03 | 42.4±5.19 | 87.8±3.62 | 36.5±1.08 |
| 100,000,000 Train $\rho$ steps *Direct Torque Cont. - High Freq. (500Hz)* | 137±17 | 106±1.46 | 124±3.52 | 133±4.13 | 139±3.5 | 88.2±1.08 | 63.2±2.86 | 42.5±13.9 | 103±2.32 | 67.6±6.25 | 91.5±6.51 |
| 1,000,000,000 Train $\rho$ steps *Direct Torque Cont. - High Freq. (500Hz)* | 125±15.4 | 101±1.35 | 141±2.49 | 127±3.79 | 119±3.05 | 87.5±1 | 84.8±1.34 | 219±11.1 | 98.3±2.58 | 73.5±6.04 | 149±6.47 |
| 10,000,000,000 Train $\rho$ steps *Direct Torque Cont. - High Freq. (500Hz)* | 185±16.6 | 86.3±1.09 | 155±6.58 | 143±6.98 | 71.3±1.3 | 91.8±1.08 | 84.6±1.21 | 75.5±12.7 | 88.5±3.01 | 81.2±4.86 | 219±5.53 |

## J.4. Ablation: Equal Computational Complexity for All Baselines

We investigate making the computational complexity the same for all baselines and EvoControl, in two approaches. First, the most direct approach we set the budget of the number of low-level environment steps to be equivalent for all baselines, listed as *equivalent number of low-level environment steps* in Appendix J.4.1. Second, we recognize that modern GPUs allow for environment parallelization. Thus, we investigate only setting the same number of sequential low-level environment steps to be equivalent for all baselines—where the bottleneck for parallelized rollouts is the number of sequential steps of the parallelized environments. This is listed as *equivalent number of sequential low-level environment steps* in Appendix J.4.2.

### J.4.1. EQUIVALENT NUMBER OF LOW-LEVEL ENVIRONMENT STEPS

Here we explicitly set the total number of low-level environment steps for all the baselines to be the same. For EvoControl, that trains its high-level policy with PPO and its low-level policy with ES, this means that the high-level policy trained with PPO now receives less high-level update steps compared to the baselines, to account for the low-level step budget used by the low-level ES component. This is different from the main results within the paper (Section 5.1) which trained each baseline for 1M high-level steps, thus meaning that the high-level policy $\rho$ was trained for 1M steps, not accounting for the potentially differing number of low-level steps used by updating or using the lower-level policy.

To set the total number of low-level environment steps for all the baselines to be the same, we first compute the total number of low-level steps that EvoControl uses, where we train EvoControl's high-level policy for 1M steps, and then now train the baseline methods for this increased number of equivalent high-level steps. Specifically if we consider a PD

position baseline, originally training this for 1M high-level environment steps, with a lower-level PD position controller, operating at a higher frequency with $G = 16$, meaning that the number of low-level environment steps used in the environment are 1M $\times$ 16 = 16$M$. Here EvoControl, when the high-level is trained for 1M steps, the lower-level policy also receives updates, therefore the total number of low-level environment steps used within the training of EvoControl is $K \times$ es_rollouts $\times$ es_sub_generations $\times$ es_pop_size $\times$ episode_length. To simplify the comparison, we explicitly set es_rollouts $= 1$ and leave the other inputs the same as they were for the main results (that of $K = 8$, es_sub_generations $=$ 8, episode_length $= 1000$). This leaves the input parameter of es_pop_size that we can vary. Therefore the total number of low-level environment steps used by EvoControl is 1M $\times$ 16 + 64$K \times$ es_pop_size. Therefore, for the following experiments we train all the other non-EvoControl baselines for 1M $\times$ 16 + 64$K \times$ es_pop_size low-level environment steps, by specifically determining how many high-level steps this is by dividing by $G$ and using that as the input as the total number of high-level steps to train for each baseline.

For the results, as discussed, we vary es_pop_size $= \{16, 32, 64, 128, 256\}$ and re-run each non-EvoControl baseline with the equivalent number of low-level steps as EvoControl—which means as EvoControl uses ES to update the low-level policy, the high-level policy now receives less equivalent updates compared to the high-level policy of the non-EvoControl baselines. We observe in Tables 14 to 18 that EvoControl even when limited to use the same number of low-level environment steps as all the baselines, on average achieves a higher normalized evaluation return than all the baselines *fixed controllers* and *direct torque control*.

Table 14: Ablation. Equivalent Number of Low-Level Environment Steps, with es_pop_size $= 16$. Normalized evaluation return $\mathcal{R}$ for the benchmark methods, across each environment. EvoControl on average achieves a higher normalized evaluation return than the baselines of *fixed controllers* and *direct torque control*. Results are averaged over 6400 random seeds, with $\pm$ indicating 95% confidence intervals. Returns are normalized to a 0-100 scale, where 0 represents a random policy, and 100 represents the highest reward achieved by a non-EvoControl baseline in each environment. Scores bolded are greater than 100.

| Same PPO high-level alg. $\rho$ with a Low-Level Policy $\beta$ of | Ant | Halfcheetah | Hopper | Humanoid | Humanoid Standup | Inverted Double Pendulum | Inverted Pendulum | Pusher | Reacher | Reacher 1D | Walker2D |
|---|---|---|---|---|---|---|---|---|---|---|---|
| | $\mathcal{R}\uparrow$ | $\mathcal{R}\uparrow$ | $\mathcal{R}\uparrow$ | $\mathcal{R}\uparrow$ | $\mathcal{R}\uparrow$ | $\mathcal{R}\uparrow$ | $\mathcal{R}\uparrow$ | $\mathcal{R}\uparrow$ | $\mathcal{R}\uparrow$ | $\mathcal{R}\uparrow$ | $\mathcal{R}\uparrow$ |
| *Fixed Cont.* - PD Position | 100±1.75 | 62.1±0.115 | 97.5±0.246 | 100±0.629 | 100±0.144 | 100±0.00784 | 100±3.74e-07 | 100±2.91 | 100±0.337 | 86±0.672 | 83.1±0.0441 |
| *Fixed Cont.* - PD Position Delta | 4.11±0.503 | 2.84±0.022 | 100±0.679 | 96.7±0.391 | 2.73±0.0111 | 55.9±0.412 | 100±3.74e-07 | 33±3.84 | 42.2±0.705 | 18.6±1.74 | 93.8±0.052 |
| *Fixed Cont.* - PD Int. Velocity | 2.05±0.504 | 2.53±0.0229 | 76.1±0.319 | 86.5±0.315 | 0±0 | 48.1±0.388 | 86.6±0.496 | 0±0 | 0±0 | 0±0 | 73.1±0.633 |
| *Fixed Cont.* - Random | 0.0±0.0 | 0.0±0.0 | 0.0±0.0 | 0.0±0.0 | 0.0±0.0 | 0.0±0.0 | 0.0±0.0 | 0.0±0.0 | 0.0±0.0 | 0.0±0.0 | 0.0±0.0 |
| *Direct Torque Cont.* - High Freq. (500Hz) | 0±0 | 24.1±0.233 | 12.3±0.336 | 0±0 | 31.3±0.434 | 3.34±0.797 | 60.2±1.07 | 2.43±2.54 | 17.4±1.22 | 72.6±1.21 | 36.4±0.7 |
| *Direct Torque Cont.* - Low Freq. (31.25Hz) | 46.5±1.42 | 100±0.359 | 78.3±0.148 | 97.1±0.627 | 81.3±0.292 | 100±0.00823 | 99.7±0.0278 | 80.6±3.86 | 59.8±0.801 | 100±0.504 | 100±0.272 |
| **EvoControl (Full State)** | **145±2.51** | 89.7±0.661 | **127±0.596** | 96.2±0.651 | 88.9±0.111 | **100±0.0083** | 100±3.74e-07 | **423±3.41** | **107±0.192** | **106±0.183** | **134±0.491** |
| **EvoControl (Residual State)** | **124±2.45** | **125±0.595** | 83.5±0.719 | **117±0.768** | **101±0.159** | 99.6±0.0159 | 100±3.74e-07 | **445±3.61** | **105±0.178** | **105±0.227** | **188±0.831** |
| **EvoControl (Target + Proprio.)** | **148±2.81** | 99.2±0.719 | **146±0.998** | **122±0.707** | 99.1±0.452 | 99.7±0.0748 | 95.4±0.29 | **468±3.31** | **107±0.177** | **106±0.18** | **132±0.498** |
| **EvoControl (Target)** | **141±2.79** | **106±0.993** | **135±0.621** | **124±0.713** | **100±0.374** | 99.9±0.0148 | 100±3.74e-07 | **420±3.17** | 65.3±0.802 | **103±0.29** | **137±0.697** |
| **EvoControl (Learned Gains)** | 79.8±2.43 | 52.3±0.487 | **109±1.1** | 85.3±0.732 | 93.6±0.178 | 72.5±1.24 | 100±3.74e-07 | **409±3.45** | 99.5±0.421 | 97.9±0.449 | **114±0.909** |
| **EvoControl (Delta Position)** | **139±2.81** | 99.2±0.85 | **141±0.382** | 97.6±0.764 | 68.8±0.63 | 87.6±0.565 | 100±3.74e-07 | **398±3.32** | 63.2±0.802 | **102±0.337** | **150±0.529** |

Table 15: Ablation. Equivalent Number of Low-Level Environment Steps, with es_pop_size $= 32$. Normalized evaluation return $\mathcal{R}$ for the benchmark methods, across each environment. EvoControl on average achieves a higher normalized evaluation return than the baselines of *fixed controllers* and *direct torque control*. Results are averaged over 6400 random seeds, with $\pm$ indicating 95% confidence intervals. Returns are normalized to a 0-100 scale, where 0 represents a random policy, and 100 represents the highest reward achieved by a non-EvoControl baseline in each environment. Scores bolded are greater than 100.

| Same PPO high-level alg. $\rho$ with a Low-Level Policy $\beta$ of | Ant | Halfcheetah | Hopper | Humanoid | Humanoid Standup | Inverted Double Pendulum | Inverted Pendulum | Pusher | Reacher | Reacher 1D | Walker2D |
|---|---|---|---|---|---|---|---|---|---|---|---|
| | $\mathcal{R}\uparrow$ | $\mathcal{R}\uparrow$ | $\mathcal{R}\uparrow$ | $\mathcal{R}\uparrow$ | $\mathcal{R}\uparrow$ | $\mathcal{R}\uparrow$ | $\mathcal{R}\uparrow$ | $\mathcal{R}\uparrow$ | $\mathcal{R}\uparrow$ | $\mathcal{R}\uparrow$ | $\mathcal{R}\uparrow$ |
| *Fixed Cont.* - PD Position | 100±1.23 | 59.6±0.195 | 99.5±0.0651 | 83.2±0.473 | 100±0.275 | 99.9±0.00874 | 100±3.74e-07 | 100±2.71 | 100±0.258 | 86.3±0.645 | 74.8±0.128 |
| *Fixed Cont.* - PD Position Delta | 3.42±0.447 | 2.72±0.0204 | 100±0.266 | 96.2±0.411 | 3.03±0.00787 | 56.6±0.395 | 100±3.74e-07 | 50.7±3.73 | 42.6±0.682 | 18.5±1.72 | 93.3±0.072 |
| *Fixed Cont.* - PD Int. Velocity | 4.62±0.453 | 2.45±0.0214 | 82.5±0.702 | 79.4±0.275 | 0±0 | 50.6±0.404 | 96.9±0.226 | 0±0 | 0±0 | 0±0 | 100±0.519 |
| *Fixed Cont.* - Random | 0.0±0.0 | 0.0±0.0 | 0.0±0.0 | 0.0±0.0 | 0.0±0.0 | 0.0±0.0 | 0.0±0.0 | 0.0±0.0 | 0.0±0.0 | 0.0±0.0 | 0.0±0.0 |
| *Direct Torque Cont.* - High Freq. (500Hz) | 0±0 | 44.3±0.269 | 33.8±0.449 | 27.9±0.605 | 55±0.664 | 34.7±0.47 | 92.7±0.287 | 3.92±2.35 | 18.7±1.21 | 77±1.13 | 47.5±0.404 |
| *Direct Torque Cont.* - Low Freq. (31.25Hz) | 42.9±1.15 | 100±0.257 | 99.4±0.446 | 100±0.449 | 96.3±0.133 | 100±0.0082 | 100±3.74e-07 | 91.8±3.56 | 62.6±0.749 | 100±0.475 | 84.1±0.306 |
| **EvoControl (Full State)** | **158±2.4** | **143±0.502** | **142±0.195** | 93±0.663 | 96.5±0.144 | **100±0.00707** | 100±3.74e-07 | **411±2.87** | **107±0.163** | **106±0.17** | **149±0.484** |
| **EvoControl (Residual State)** | **158±2.18** | **112±0.433** | 94.2±0.114 | **112±0.68** | **123±0.683** | 99.5±0.0196 | 100±3.74e-07 | **424±3.31** | 97.4±0.435 | **105±0.209** | **200±0.515** |
| **EvoControl (Target + Proprio.)** | **152±2.39** | **120±0.848** | **127±0.427** | **123±0.614** | **116±0.405** | 99.3±0.0701 | 100±3.74e-07 | **444±3.27** | 57.3±0.813 | **106±0.165** | **145±0.792** |
| **EvoControl (Target)** | **154±2.33** | **103±0.521** | **141±0.396** | **122±0.579** | **117±0.609** | 88.4±0.494 | 100±3.74e-07 | **404±3.29** | 96.4±0.309 | **104±0.219** | **136±0.557** |
| **EvoControl (Learned Gains)** | **104±2.31** | 67.1±0.518 | **155±1.98** | 96.9±0.657 | **100±0.209** | 96.2±0.142 | 100±3.74e-07 | **360±2.9** | **102±0.281** | **102±0.31** | **161±0.705** |
| **EvoControl (Delta Position)** | **161±2.53** | **105±0.707** | **127±0.91** | **101±0.674** | 74±0.479 | **100±0.0103** | 100±3.74e-07 | **429±3.24** | 59.5±0.774 | **102±0.316** | **149±0.361** |

Table 16: Ablation. Equivalent Number of Low-Level Environment Steps, with es_pop_size = 64. Normalized evaluation return $\mathcal{R}$ for the benchmark methods, across each environment. EvoControl on average achieves a higher normalized evaluation return than the baselines of *fixed controllers* and *direct torque control*. Results are averaged over 6400 random seeds, with $\pm$ indicating 95% confidence intervals. Returns are normalized to a 0-100 scale, where 0 represents a random policy, and 100 represents the highest reward achieved by a non-EvoControl baseline in each environment. Scores bolded are greater than 100.

| Same PPO high-level alg. $\rho$ with a Low-Level Policy $\beta$ of | Ant | Halfcheetah | Hopper | Humanoid | Humanoid Standup | Inverted Double Pendulum | Inverted Pendulum | Pusher | Reacher | Reacher 1D | Walker2D |
|---|---|---|---|---|---|---|---|---|---|---|---|
| | $\mathcal{R}\uparrow$ | $\mathcal{R}\uparrow$ | $\mathcal{R}\uparrow$ | $\mathcal{R}\uparrow$ | $\mathcal{R}\uparrow$ | $\mathcal{R}\uparrow$ | $\mathcal{R}\uparrow$ | $\mathcal{R}\uparrow$ | $\mathcal{R}\uparrow$ | $\mathcal{R}\uparrow$ | $\mathcal{R}\uparrow$ |
| *Fixed Cont.* - PD Position | 100±5.17 | 66±0.555 | 90.2±0.489 | 92.8±2.9 | 87.8±1.3 | 99.6±0.0357 | 100±1.53e-06 | 100±9.4 | 100±1.24 | 88.7±2.6 | 68.7±0.285 |
| *Fixed Cont.* - PD Position Delta | 4.93±1.67 | 2.88±0.0891 | 84.3±0.854 | 100±1.79 | 3.16±0.024 | 57.1±1.6 | 100±1.53e-06 | 3.4±9.98 | 42.8±2.99 | 27.9±7.48 | 85.4±0.513 |
| *Fixed Cont.* - PD Int. Velocity | 5.94±1.8 | 2.59±0.0938 | 56.9±2.3 | 72.6±0.872 | 0±0 | 61.3±1.59 | 99±0.45 | 0±0 | 0±0 | 9.58±7.72 | 100±0.681 |
| *Fixed Cont.* - Random | 0.0±0.0 | 0.0±0.0 | 0.0±0.0 | 0.0±0.0 | 0.0±0.0 | 0.0±0.0 | 0.0±0.0 | 0.0±0.0 | 0.0±0.0 | 0.0±0.0 | 0.0±0.0 |
| *Direct Torque Cont.* - High Freq. (500Hz) | 0±0 | 44±0.794 | 42.9±0.53 | 63.6±2.37 | 53±2.15 | 97.8±0.387 | 49.9±2.36 | 9.18±7.69 | 10.8±5.42 | 81.1±4.24 | 32.2±1.27 |
| *Direct Torque Cont.* - Low Freq. (31.25Hz) | 70.4±6.52 | 100±1.22 | 100±1.26 | 87.2±1.35 | 100±0.204 | 100±0.0178 | 100±1.53e-06 | 83.1±10.2 | 67.2±3.05 | 100±1.79 | 69.2±1.38 |
| **EvoControl (Full State)** | **188±12** | **165±1.15** | **118±22.4** | **111±15.9** | 90.2±6.6 | **100±0.306** | 100±0 | **296±42.6** | **109±1.86** | **104±2.14** | **127±126** |
| **EvoControl (Residual State)** | **152±117** | **166±16.9** | **125±118** | **108±43** | **112±14.2** | 99.2±0.304 | 100±0 | **302±18.1** | **102±1.08** | **104±2.07** | **169±59.6** |
| **EvoControl (Target + Proprio.)** | **183±69.2** | **135±11.1** | **130±55.2** | **126±12.1** | **124±58.8** | 87.1±53.3 | 100±0 | **300±42.6** | **107±2.77** | **104±2.23** | **130±12.7** |
| **EvoControl (Target)** | **201±27.5** | **164±10.6** | **131±9.3** | **125±4.52** | **109±12.2** | 99.5±1.7 | 100±0 | **286±27** | 83.5±63.4 | **103±3.2** | **122±38** |
| **EvoControl (Learned Gains)** | **103±23.1** | 90±20.6 | **141±845** | **111±28.9** | **102±1.5** | 93.9±26 | 100±0 | **263±46.8** | **104±11.4** | **102±4.85** | **149±35.2** |
| **EvoControl (Delta Position)** | **168±29.9** | **123±75.2** | 95.9±40.8 | **104±9.23** | 94.2±8.68 | **100±0.491** | 100±0 | **300±61.6** | 59±5.93 | 99.7±7.07 | **160±12.4** |

Table 17: Ablation. Equivalent Number of Low-Level Environment Steps, with es_pop_size = 128. Normalized evaluation return $\mathcal{R}$ for the benchmark methods, across each environment. EvoControl on average achieves a higher normalized evaluation return than the baselines of *fixed controllers* and *direct torque control*. Results are averaged over 384 random seeds, with $\pm$ indicating 95% confidence intervals. Returns are normalized to a 0-100 scale, where 0 represents a random policy, and 100 represents the highest reward achieved by a non-EvoControl baseline in each environment. Scores bolded are greater than 100.

| Same PPO high-level alg. $\rho$ with a Low-Level Policy $\beta$ of | Ant | Halfcheetah | Hopper | Humanoid | Humanoid Standup | Inverted Double Pendulum | Inverted Pendulum | Pusher | Reacher | Reacher 1D | Walker2D |
|---|---|---|---|---|---|---|---|---|---|---|---|
| | $\mathcal{R}\uparrow$ | $\mathcal{R}\uparrow$ | $\mathcal{R}\uparrow$ | $\mathcal{R}\uparrow$ | $\mathcal{R}\uparrow$ | $\mathcal{R}\uparrow$ | $\mathcal{R}\uparrow$ | $\mathcal{R}\uparrow$ | $\mathcal{R}\uparrow$ | $\mathcal{R}\uparrow$ | $\mathcal{R}\uparrow$ |
| *Fixed Cont.* - PD Position | 100±9.19 | 73.2±0.419 | 86±0.878 | 89.5±2.72 | 78.4±1.31 | 99.7±0.0353 | 100±1.53e-06 | 100±8.67 | 100±0.928 | 86±2.76 | 100±0.935 |
| *Fixed Cont.* - PD Position Delta | 3.33±2.59 | 3.12±0.101 | 48.8±1.06 | 89±1.69 | 3.22±0.0153 | 70.2±1.95 | 100±1.53e-06 | 10.2±9.63 | 43.6±2.74 | 19.9±7.52 | 72.7±1.51 |
| *Fixed Cont.* - PD Int. Velocity | 8.12±2.69 | 2.82±0.107 | 42.2±1.94 | 57.8±0.929 | 0±0 | 52.2±1.66 | 100±1.53e-06 | 0±0 | 0±0 | 0.651±7.83 | 85.3±0.32 |
| *Fixed Cont.* - Random | 0.0±0.0 | 0.0±0.0 | 0.0±0.0 | 0.0±0.0 | 0.0±0.0 | 0.0±0.0 | 0.0±0.0 | 0.0±0.0 | 0.0±0.0 | 0.0±0.0 | 0.0±0.0 |
| *Direct Torque Cont.* - High Freq. (500Hz) | 4.16±15.4 | 66.5±2.22 | 46.6±1.49 | 54.8±1.15 | 75.7±2.04 | 93.4±0.598 | 49.6±2.28 | 14.2±5.96 | 33.5±5.21 | 83.8±4.27 | 41.1±1.75 |
| *Direct Torque Cont.* - Low Freq. (31.25Hz) | 53.9±7.67 | 100±1.74 | 100±3.31 | 100±1.87 | 100±0.397 | 100±0.029 | 100±1.53e-06 | 85.2±6.7 | 68.1±2.89 | 100±1.53 | 88.5±0.724 |
| **EvoControl (Full State)** | **328±88.2** | **148±68.4** | **105±2.16** | 87.1±8 | 97.1±13.4 | **101±0.0993** | 100±0 | **252±16.8** | **106±0.66** | **105±2.23** | **127±57.7** |
| **EvoControl (Residual State)** | **197±40.1** | **191±33** | 94.2±120 | **120±20.9** | **121±42** | 99.8±0.474 | 100±0 | **239±18.6** | **102±4.38** | **104±6.29** | **138±127** |
| **EvoControl (Target + Proprio.)** | **274±75.2** | **202±14.7** | **121±51.5** | **127±19.1** | **152±89.5** | 90.6±31.5 | 100±0 | **253±44.6** | **107±2.27** | **105±2.33** | **135±23.9** |
| **EvoControl (Target)** | **313±6.88** | **204±2.9** | **113±47.8** | **122±10.3** | **135±99** | 92.4±31.5 | 100±0 | **245±22.2** | **101±2.17** | **104±1.27** | **128±53.9** |
| **EvoControl (Learned Gains)** | **242±90.9** | **116±1.47** | **142±94.7** | **104±18.6** | **104±2.93** | 99.6±3.18 | 100±0 | **236±21.6** | **105±9.62** | **103±5.11** | **134±55.8** |
| **EvoControl (Delta Position)** | **296±14.1** | **151±50** | 61.5±287 | 98.8±4.74 | 93.3±24.5 | **101±0.13** | 100±0 | **238±19.7** | 60.2±24.1 | 95.6±19.8 | **123±41.4** |

Table 18: Ablation. Equivalent Number of Low-Level Environment Steps, with es_pop_size = 256. Normalized evaluation return $\mathcal{R}$ for the benchmark methods, across each environment. EvoControl on average achieves a higher normalized evaluation return than the baselines of *fixed controllers* and *direct torque control*. Results are averaged over 384 random seeds, with $\pm$ indicating 95% confidence intervals. Returns are normalized to a 0-100 scale, where 0 represents a random policy, and 100 represents the highest reward achieved by a non-EvoControl baseline in each environment. Scores bolded are greater than 100.

| Same PPO high-level alg. $\rho$ with a Low-Level Policy $\beta$ of | Ant | Halfcheetah | Hopper | Humanoid | Humanoid Standup | Inverted Double Pendulum | Inverted Pendulum | Pusher | Reacher | Reacher 1D | Walker2D |
|---|---|---|---|---|---|---|---|---|---|---|---|
| | $\mathcal{R}\uparrow$ | $\mathcal{R}\uparrow$ | $\mathcal{R}\uparrow$ | $\mathcal{R}\uparrow$ | $\mathcal{R}\uparrow$ | $\mathcal{R}\uparrow$ | $\mathcal{R}\uparrow$ | $\mathcal{R}\uparrow$ | $\mathcal{R}\uparrow$ | $\mathcal{R}\uparrow$ | $\mathcal{R}\uparrow$ |
| *Fixed Cont.* - PD Position | 100±5.67 | 60±0.402 | 47.3±1.17 | 100±1.78 | 96.2±0.628 | 100±0.0379 | 100±1.53e-06 | 100±6.56 | 100±0.959 | 85.4±2.66 | 100±0.253 |
| *Fixed Cont.* - PD Position Delta | 3±1.86 | 2.71±0.0777 | 27.6±0.461 | 98.3±1.56 | 3.03±0.024 | 96.3±0.84 | 100±1.53e-06 | 8.47±8.21 | 49.4±2.42 | 21.9±7.31 | 57.2±2.07 |
| *Fixed Cont.* - PD Int. Velocity | 6.16±1.79 | 2.46±0.0836 | 24.5±0.943 | 57.3±1.29 | 0±0 | 22±0.756 | 100±1.53e-06 | 0±0 | 0±0 | 3.33±7.57 | 66.1±0.156 |
| *Fixed Cont.* - Random | 0.0±0.0 | 0.0±0.0 | 0.0±0.0 | 0.0±0.0 | 0.0±0.0 | 0.0±0.0 | 0.0±0.0 | 0.0±0.0 | 0.0±0.0 | 0.0±0.0 | 0.0±0.0 |
| *Direct Torque Cont.* - High Freq. (500Hz) | 28.7±13.2 | 62.2±1.3 | 58.7±0.756 | 65.7±2.62 | 75.1±2.23 | 99.5±0.0862 | 60.9±3.02 | 19.3±5.42 | 45.7±4.95 | 83.4±3.99 | 43.4±1.66 |
| *Direct Torque Cont.* - Low Freq. (31.25Hz) | 52.4±5.84 | 100±0.992 | 100±0.664 | 94±1.3 | 100±0.435 | 100±0.0472 | 100±1.53e-06 | 89.8±3.93 | 81.2±2.41 | 100±1.34 | 79.1±0.946 |
| **EvoControl (Full State)** | **302±40.8** | **144±17.5** | 69.9±18.2 | 97.1±14.5 | **109±19.4** | **101±0.524** | 100±0 | **206±15.8** | 89±64.6 | **103±2.1** | **123±3.79** |
| **EvoControl (Residual State)** | **138±28.7** | **162±22.8** | 37±3.31 | **124±49** | **129±37.4** | **101±0.561** | 100±0 | **215±21.9** | 99.9±7.45 | **103±2.89** | **128±94.1** |
| **EvoControl (Target + Proprio.)** | **281±39.5** | **130±7.09** | 90.1±11.2 | **125±42.8** | **141±70.8** | 99.2±5.02 | 76.9±99.4 | **205±11.8** | **104±3.39** | **103±2.14** | **118±30.7** |
| **EvoControl (Target)** | **241±42.1** | **150±19.8** | 75.1±38.2 | **124±21.5** | **118±8.06** | 99.7±2.18 | 100±0 | **198±21.5** | 75.7±56.7 | **102±2.93** | 92.7±30.3 |
| **EvoControl (Learned Gains)** | **197±86.2** | 96.3±13.9 | 59.1±50.9 | **111±21.7** | **107±1.18** | **101±0.202** | 100±0 | **190±45.5** | **104±0.793** | **102±2.65** | **135±40** |
| **EvoControl (Delta Position)** | **257±50.6** | **116±23.7** | **102±2.25** | 98.4±14.1 | 98.5±11.2 | **101±1.57** | 100±0 | **206±25.4** | 65.3±40.5 | 95±15 | **130±48.7** |

J.4.2. EQUIVALENT NUMBER OF SEQUENTIAL LOW-LEVEL ENVIRONMENT STEPS

Here we set the total number of *sequential* low-level environment steps for all the baselines to be the same. This is approach is different from setting the total number of low-level environment steps to be the same, as it acknowledges the more realistic scenario of performing parallel environment rollouts on modern GPUs. With parallelized rollouts, the computational bottleneck becomes the number of *sequential* steps within each environment, as the overhead of increasing the number of parallel environments is negligible compared to increasing the number of sequential steps—assuming a user has a sufficiently large GPU to perform parallelized rollouts of the environment. Such an assumption is common in practice (Huang et al., 2022), with many implementations of PPO and simulation environments natively supporting parallelized rollouts for the environment (Brockman et al., 2016b).

To set the total number of sequential low-level environment steps for all the baselines to be the same, we follow a similar setup as described in Appendix J.4.1, now only accounting for the sequential low-level environment steps that EvoControl uses. Specifically, we first compute the total number of sequential low-level steps that EvoControl uses, where we train EvoControl's high-level policy for 1M steps, and then now train the baseline methods for this increased number of equivalent high-level steps. Specifically if we consider a PD position baseline, originally training this for 1M high-level environment steps, with a lower-level PD position controller, operating at a higher frequency with $G = 16$, meaning that the number of low-level environment steps used in the environment are $1\text{M} \times 16 = 16M$. Here EvoControl, when the high-level is trained for 1M steps, the lower-level policy also receives updates, therefore the total number of sequential low-level environment steps used within the training of EvoControl is $K \times$ es_sub_generations $\times$ episode_length. As es_rollouts and es_pop_size are parallelized, they are not counted in the total number of sequential low-level steps, therefore we leave them as the default values as defined in Appendix G. We leave the other parameters the same as they were for the main results (that of $K = 8$, es_sub_generations $= 8$, episode_length $= 1000$). This leaves the input parameter of es_pop_size that we can vary. Therefore the total number of sequential low-level environment steps used by EvoControl is $1\text{M} \times 16 + 64K$, a fixed amount. Therefore, for the following experiments we train all the other non-EvoControl baselines for $1\text{M} \times 16 + 64K$ low-level environment steps, by specifically determining how many high-level steps this is by dividing by $G$ and using that as the input as the total number of high-level steps to train for each baseline.

For the results, as discussed, we vary es_pop_size $= \{16, 32, 64, 128, 256\}$ and re-run each non-EvoControl baseline with the equivalent number of sequential low-level steps as EvoControl—which means as EvoControl uses ES to update the low-level policy, the high-level policy now receives less equivalent updates compared to the high-level policy of the non-EvoControl baselines. We observe in Tables 19 to 23 that EvoControl even when limited to use the same number of sequential low-level environment steps as all the baselines, on average achieves a higher normalized evaluation return than all the baselines *fixed controllers* and *direct torque control*—which remains consistent with the results seen throughout.

Table 19: Ablation. Equivalent Number of Sequential Low-Level Environment Steps, with es_pop_size $= 16$. Normalized evaluation return $\mathcal{R}$ for the benchmark methods, across each environment. EvoControl on average achieves a higher normalized evaluation return than the baselines of *fixed controllers* and *direct torque control*. Results are averaged over 384 random seeds, with $\pm$ indicating 95% confidence intervals. Returns are normalized to a 0-100 scale, where 0 represents a random policy, and 100 represents the highest reward achieved by a non-EvoControl baseline in each environment. Scores bolded are greater than 100.

| Same PPO high-level alg. $\rho$ with a Low-Level Policy $\beta$ of | Ant | Halfcheetah | Hopper | Humanoid | Humanoid Standup | Inverted Double Pendulum | Inverted Pendulum | Pusher | Reacher | Reacher 1D | Walker2D |
|---|---|---|---|---|---|---|---|---|---|---|---|
| | $\mathcal{R}\uparrow$ | $\mathcal{R}\uparrow$ | $\mathcal{R}\uparrow$ | $\mathcal{R}\uparrow$ | $\mathcal{R}\uparrow$ | $\mathcal{R}\uparrow$ | $\mathcal{R}\uparrow$ | $\mathcal{R}\uparrow$ | $\mathcal{R}\uparrow$ | $\mathcal{R}\uparrow$ | $\mathcal{R}\uparrow$ |
| *Fixed Cont.* - PD Position | 100±7.29 | 69.3±0.723 | 93.7±1.28 | 100±2.75 | 100±0.194 | 99.9±0.0347 | 100±1.53e-06 | 100±8.47 | 100±1.39 | 85±2.93 | 82.8±0.176 |
| *Fixed Cont.* - PD Position Delta | 4.81±1.78 | 3.14±0.101 | 100±1.55 | 97.7±1.85 | 2.6±0.0436 | 53.9±2.1 | 100±1.53e-06 | 0±0 | 40.3±3.19 | 15.2±7.6 | 93.3±0.235 |
| *Fixed Cont.* - PD Int. Velocity | 4.8±1.82 | 2.8±0.106 | 75.2±0.875 | 84±1.34 | 0±0 | 48.8±1.93 | 87.8±1.97 | 0±0 | 0±0 | 0±0 | 84.1±3.36 |
| *Fixed Cont.* - Random | 0.0±0.0 | 0.0±0.0 | 0.0±0.0 | 0.0±0.0 | 0.0±0.0 | 0.0±0.0 | 0.0±0.0 | 0.0±0.0 | 0.0±0.0 | 0.0±0.0 | 0.0±0.0 |
| *Direct Torque Cont.* - High Freq. (500Hz) | 0±0 | 19.3±0.355 | 3.85±0.273 | 13.4±2.62 | 15.3±1.23 | 0±0 | 0±0 | 2.95±7.81 | 0±0 | 45.3±6.74 | 1.15±4.13 |
| *Direct Torque Cont.* - Low Freq. (31.25Hz) | 55.6±6.39 | 100±2.73 | 75.9±0.496 | 89.7±2.27 | 71.9±2.3 | 100±0.041 | 100±1.53e-06 | 75.4±12.8 | 59.2±3.7 | 100±1.94 | 100±0.384 |
| **EvoControl (Full State)** | **116±62.6** | **104±22** | **127±52.3** | 94.7±32.6 | 88.2±6.4 | **100±0.553** | 100±0 | **272±112** | 95.5±65.5 | **105±3.31** | **130±63.9** |
| **EvoControl (Residual State)** | **163±118** | **153±55** | 96.5±5.44 | 93.8±40.1 | 95.6±30.9 | 99.2±2.81 | 100±0 | **314±43.1** | **107±1.46** | **105±2.39** | **194±3.42** |
| **EvoControl (Target + Proprio.)** | **121±54.9** | 92.1±20.7 | **131±16.9** | **117±10.2** | **101±3.66** | 99.4±12.5 | 100±0 | **311±59.8** | **110±5.76** | **106±2.39** | **137±1.04** |
| **EvoControl (Target)** | **129±78.2** | **115±59.4** | **133±2.96** | **110±13.1** | 87.8±5.32 | **100±2.15** | 100±0 | **303±53.2** | 71.3±46.9 | **104±4.91** | **143±382** |
| **EvoControl (Learned Gains)** | 46.8±54.2 | 58.6±8.22 | **128±74.8** | 82.6±7.06 | 91.1±8.77 | 75.6±139 | 100±0 | **228±41.4** | **103±13.9** | 97.3±1.76 | **111±307** |
| **EvoControl (Delta Position)** | **117±33.8** | **104±57.3** | **154±41.2** | 84.9±23.5 | 67.6±56.7 | 98.1±24.1 | 100±0 | **285±52.1** | 58.6±6.77 | 99.7±3.16 | **135±68.1** |

Table 20: Ablation. Equivalent Number of Sequential Low-Level Environment Steps, with es_pop_size = 32. Normalized evaluation return $\mathcal{R}$ for the benchmark methods, across each environment. EvoControl on average achieves a higher normalized evaluation return than the baselines of *fixed controllers* and *direct torque control*. Results are averaged over 384 random seeds, with $\pm$ indicating 95% confidence intervals. Returns are normalized to a 0-100 scale, where 0 represents a random policy, and 100 represents the highest reward achieved by a non-EvoControl baseline in each environment. Scores bolded are greater than 100.

| Same PPO high-level alg. $\rho$ with a Low-Level Policy $\beta$ of | Ant | Halfcheetah | Hopper | Humanoid | Humanoid Standup | Inverted Double Pendulum | Inverted Pendulum | Pusher | Reacher | Reacher 1D | Walker2D |
|---|---|---|---|---|---|---|---|---|---|---|---|
| | $\mathcal{R}\uparrow$ | $\mathcal{R}\uparrow$ | $\mathcal{R}\uparrow$ | $\mathcal{R}\uparrow$ | $\mathcal{R}\uparrow$ | $\mathcal{R}\uparrow$ | $\mathcal{R}\uparrow$ | $\mathcal{R}\uparrow$ | $\mathcal{R}\uparrow$ | $\mathcal{R}\uparrow$ | $\mathcal{R}\uparrow$ |
| *Fixed Cont.* - PD Position | 100±5.89 | 60.8±0.558 | 89.5±1.2 | 89.2±3.27 | 100±1.83 | 99.8±0.0361 | 100±1.53e-06 | 100±8.47 | 100±1.8 | 85±2.93 | 76.8±0.193 |
| *Fixed Cont.* - PD Position Delta | 2.14±1.8 | 2.77±0.0891 | 100±1.93 | 100±1.7 | 2.73±0.0726 | 53.9±2.1 | 100±1.53e-06 | 0±0 | 41±3.24 | 15.2±7.6 | 89.6±0.171 |
| *Fixed Cont.* - PD Int. Velocity | 1.82±1.66 | 2.47±0.0937 | 73±0.882 | 85±1.21 | 0±0 | 48.8±1.93 | 86.4±2.01 | 0±0 | 0±0 | 0±0 | 81.3±3.18 |
| *Fixed Cont.* - Random | 0.0±0.0 | 0.0±0.0 | 0.0±0.0 | 0.0±0.0 | 0.0±0.0 | 0.0±0.0 | 0.0±0.0 | 0.0±0.0 | 0.0±0.0 | 0.0±0.0 | 0.0±0.0 |
| *Direct Torque Cont.* - High Freq. (500Hz) | 0±0 | 16.9±0.325 | 3.54±0.253 | 8.28±2.31 | 21.9±2.06 | 0±0 | 0±0 | 2.95±7.81 | 0±0 | 45.3±6.74 | 0±0 |
| *Direct Torque Cont.* - Low Freq. (31.25Hz) | 45.1±4.76 | 100±1.7 | 79.3±1.22 | 93.7±2.87 | 69.6±2.38 | 100±0.041 | 100±1.53e-06 | 73.1±12.7 | 59.1±3.75 | 100±1.94 | 100±0.637 |
| **EvoControl (Full State)** | **150±46.9** | **147±10** | **132±27.3** | 97.8±33.7 | **105±17.5** | 100±0.451 | 100±0 | **305±32.8** | **115±2.56** | **105±1.89** | **138±75.6** |
| **EvoControl (Residual State)** | **155±102** | **121±11.3** | 93.7±4.42 | **116±19** | **121±9.93** | 100±1.45 | 100±0 | **323±60.3** | 94±66.9 | **105±3.69** | **161±327** |
| **EvoControl (Target + Proprio.)** | **156±33.6** | **111±49.3** | **141±20.5** | **126±12.3** | **117±22** | 99.3±0.367 | 100±0 | **324±40.8** | 58.1±2.89 | **106±2.35** | **137±79.6** |
| **EvoControl (Target)** | **164±56.5** | **116±2.59** | **136±27.6** | **129±4.94** | **110±16.7** | 98.7±12.2 | 100±0 | **291±104** | 88.7±62.8 | **105±0.546** | **139±23** |
| **EvoControl (Learned Gains)** | 86.3±84.9 | 59.9±11.8 | **162±171** | **105±18.5** | **103±21.7** | 99.1±12.1 | 100±0 | **303±57.5** | **110±6.3** | **102±4.43** | **155±160** |
| **EvoControl (Delta Position)** | **156±78.1** | **101±35.5** | **111±49.3** | **101±7.07** | 99.7±8.03 | 100±3.02 | 100±0 | **302±28.8** | 58.9±5.31 | **102±9.45** | **127±60.5** |

Table 21: Ablation. Equivalent Number of Sequential Low-Level Environment Steps, with es_pop_size = 64. Normalized evaluation return $\mathcal{R}$ for the benchmark methods, across each environment. EvoControl on average achieves a higher normalized evaluation return than the baselines of *fixed controllers* and *direct torque control*. Results are averaged over 384 random seeds, with $\pm$ indicating 95% confidence intervals. Returns are normalized to a 0-100 scale, where 0 represents a random policy, and 100 represents the highest reward achieved by a non-EvoControl baseline in each environment. Scores bolded are greater than 100.

| Same PPO high-level alg. $\rho$ with a Low-Level Policy $\beta$ of | Ant | Halfcheetah | Hopper | Humanoid | Humanoid Standup | Inverted Double Pendulum | Inverted Pendulum | Pusher | Reacher | Reacher 1D | Walker2D |
|---|---|---|---|---|---|---|---|---|---|---|---|
| | $\mathcal{R}\uparrow$ | $\mathcal{R}\uparrow$ | $\mathcal{R}\uparrow$ | $\mathcal{R}\uparrow$ | $\mathcal{R}\uparrow$ | $\mathcal{R}\uparrow$ | $\mathcal{R}\uparrow$ | $\mathcal{R}\uparrow$ | $\mathcal{R}\uparrow$ | $\mathcal{R}\uparrow$ | $\mathcal{R}\uparrow$ |
| *Fixed Cont.* - PD Position | 100±5.05 | 64.2±0.518 | 91.8±1.13 | 98±2.75 | 100±0.123 | 99.9±0.03 | 100±1.53e-06 | 100±8.47 | 100±1.8 | 85±2.93 | 82.2±0.371 |
| *Fixed Cont.* - PD Position Delta | 3.6±1.66 | 2.9±0.0928 | 100±1.55 | 100±1.79 | 2.49±0.0711 | 53.8±1.57 | 100±1.53e-06 | 0±0 | 41±3.24 | 15.2±7.6 | 99.7±0.285 |
| *Fixed Cont.* - PD Int. Velocity | 4±1.56 | 2.57±0.0976 | 75.2±0.875 | 90.1±1.48 | 0±0 | 49.7±1.55 | 86.5±2 | 0±0 | 0±0 | 0±0 | 93.8±3.03 |
| *Fixed Cont.* - Random | 0.0±0.0 | 0.0±0.0 | 0.0±0.0 | 0.0±0.0 | 0.0±0.0 | 0.0±0.0 | 0.0±0.0 | 0.0±0.0 | 0.0±0.0 | 0.0±0.0 | 0.0±0.0 |
| *Direct Torque Cont.* - High Freq. (500Hz) | 0±0 | 18.2±0.332 | 1.31±0.536 | 39.3±4.2 | 11.4±0.92 | 0±0 | 0±0 | 4.36±7.75 | 0±0 | 45.3±6.74 | 0±0 |
| *Direct Torque Cont.* - Low Freq. (31.25Hz) | 48.2±5.32 | 100±1.64 | 77.5±0.571 | 95.6±2.64 | 74.5±1.78 | 100±0.0311 | 100±1.53e-06 | 75.4±12.8 | 59.1±3.75 | 100±1.94 | 100±0.592 |
| **EvoControl (Full State)** | **175±5.49** | **163±6.17** | **148±32.6** | **115±25.4** | **103±27.3** | **101±0.306** | 100±0 | **348±3.02** | **114±5.95** | **106±2.17** | **166±85.1** |
| **EvoControl (Residual State)** | **143±73.5** | **163±32.4** | **170±162** | **144±7.81** | **116±10.2** | 99.4±0.305 | 100±0 | **349±8.24** | **108±1.2** | **105±3.87** | **228±27.1** |
| **EvoControl (Target + Proprio.)** | **183±124** | **126±57.2** | **164±48.3** | **137±13** | **114±12.7** | 87.3±53.4 | 100±0 | **335±48.2** | **114±2.93** | **106±2.3** | **157±51.3** |
| **EvoControl (Target)** | **180±67.6** | **176±12.8** | **132±18.4** | **138±6.19** | **126±66** | 99.7±1.7 | 100±0 | **326±41.6** | 91.2±72.9 | **105±3.25** | **157±79** |
| **EvoControl (Learned Gains)** | **107±26.7** | 91.2±24.8 | **198±195** | **110±29.7** | **104±1.37** | 94.1±26.1 | 100±0 | **291±19** | **112±6.83** | **104±1.37** | **211±37.8** |
| **EvoControl (Delta Position)** | **180±18.7** | **117±25.1** | **131±36.6** | **112±9.84** | 59.4±30.5 | **101±0.492** | 100±0 | **342±70** | 61.7±9.32 | **101±7.19** | **159±36.7** |

Table 22: Ablation. Equivalent Number of Sequential Low-Level Environment Steps, with es_pop_size = 128. Normalized evaluation return $\mathcal{R}$ for the benchmark methods, across each environment. EvoControl on average achieves a higher normalized evaluation return than the baselines of *fixed controllers* and *direct torque control*. Results are averaged over 384 random seeds, with $\pm$ indicating 95% confidence intervals. Returns are normalized to a 0-100 scale, where 0 represents a random policy, and 100 represents the highest reward achieved by a non-EvoControl baseline in each environment. Scores bolded are greater than 100.

| Same PPO high-level alg. $\rho$ with a Low-Level Policy $\beta$ of | Ant | Halfcheetah | Hopper | Humanoid | Humanoid Standup | Inverted Double Pendulum | Inverted Pendulum | Pusher | Reacher | Reacher 1D | Walker2D |
|---|---|---|---|---|---|---|---|---|---|---|---|
| | $\mathcal{R}\uparrow$ | $\mathcal{R}\uparrow$ | $\mathcal{R}\uparrow$ | $\mathcal{R}\uparrow$ | $\mathcal{R}\uparrow$ | $\mathcal{R}\uparrow$ | $\mathcal{R}\uparrow$ | $\mathcal{R}\uparrow$ | $\mathcal{R}\uparrow$ | $\mathcal{R}\uparrow$ | $\mathcal{R}\uparrow$ |
| *Fixed Cont.* - PD Position | 100±5.68 | 67.5±0.452 | 82.2±1.12 | 100±2.82 | 100±0.151 | 99.9±0.03 | 100±1.53e-06 | 100±8.47 | 100±1.8 | 85.2±2.87 | 75.5±0.396 |
| *Fixed Cont.* - PD Position Delta | 3.75±1.68 | 3.04±0.0975 | 100±1.17 | 92.9±1.59 | 2.4±0.064 | 53.8±1.57 | 100±1.53e-06 | 0±0 | 41.1±3.24 | 15.2±7.6 | 92.9±0.185 |
| *Fixed Cont.* - PD Int. Velocity | 1.81±1.6 | 2.7±0.103 | 67.2±0.812 | 81.4±1.29 | 0±0 | 49.7±1.55 | 86.5±2 | 0±0 | 0±0 | 0±0 | 86.5±2.67 |
| *Fixed Cont.* - Random | 0.0±0.0 | 0.0±0.0 | 0.0±0.0 | 0.0±0.0 | 0.0±0.0 | 0.0±0.0 | 0.0±0.0 | 0.0±0.0 | 0.0±0.0 | 0.0±0.0 | 0.0±0.0 |
| *Direct Torque Cont.* - High Freq. (500Hz) | 0±0 | 18.8±0.359 | 1.27±0.479 | 10±1.85 | 15.9±0.789 | 0±0 | 0±0 | 2.95±7.81 | 0±0 | 45.3±6.74 | 0±0 |
| *Direct Torque Cont.* - Low Freq. (31.25Hz) | 36.8±4.92 | 100±2.11 | 65.2±0.505 | 90.4±2.1 | 87.4±1.02 | 100±0.0311 | 100±1.53e-06 | 73.1±12.7 | 60.6±3.71 | 100±1.94 | 100±0.416 |
| **EvoControl (Full State)** | **258±60.9** | **124±13.1** | **157±62.4** | 99.4±13.6 | 93.4±5.77 | **101±0.0993** | 100±0 | **346±3.75** | **115±3.34** | **106±2.28** | **184±45.5** |
| **EvoControl (Residual State)** | **166±86.1** | **180±23.9** | **128±163** | **147±17.9** | **140±91.5** | 99.8±0.474 | 100±0 | **340±30.4** | **109±4.07** | **104±5.95** | **211±39.8** |
| **EvoControl (Target + Proprio.)** | **241±45.1** | **198±4.9** | **172±68.9** | **139±13** | **134±106** | 90.6±37.6 | 100±0 | **363±125** | **116±2.5** | **106±2.07** | **163±68.3** |
| **EvoControl (Target)** | **218±53.3** | **197±5.49** | **165±93.1** | **132±20.7** | **134±99.5** | 92.4±31.5 | 100±0 | **335±40.4** | **109±2.03** | **105±1.27** | **155±21.1** |
| **EvoControl (Learned Gains)** | **165±85.1** | **107±10.1** | **170±174** | **124±11.7** | 98.1±7.44 | 99.7±3.18 | 100±0 | **309±49.7** | **112±8.66** | **104±5.15** | **194±87.8** |
| **EvoControl (Delta Position)** | **216±20.1** | **125±27** | **114±153** | **115±12.1** | 94.4±21.9 | **101±0.13** | 100±0 | **334±16.5** | 59.4±6.88 | 95.9±21.7 | **190±74.7** |

Table 23: Ablation. Equivalent Number of Sequential Low-Level Environment Steps, with es_pop_size = 256. Normalized evaluation return $\mathcal{R}$ for the benchmark methods, across each environment. EvoControl on average achieves a higher normalized evaluation return than the baselines of *fixed controllers* and *direct torque control*. Results are averaged over 384 random seeds, with $\pm$ indicating 95% confidence intervals. Returns are normalized to a 0-100 scale, where 0 represents a random policy, and 100 represents the highest reward achieved by a non-EvoControl baseline in each environment. Scores bolded are greater than 100.

| Same PPO high-level alg. $\rho$ with a Low-Level Policy $\beta$ of | Ant | Halfcheetah | Hopper | Humanoid | Humanoid Standup | Inverted Double Pendulum | Inverted Pendulum | Pusher | Reacher | Reacher 1D | Walker2D |
|---|---|---|---|---|---|---|---|---|---|---|---|
| | $\mathcal{R}\uparrow$ | $\mathcal{R}\uparrow$ | $\mathcal{R}\uparrow$ | $\mathcal{R}\uparrow$ | $\mathcal{R}\uparrow$ | $\mathcal{R}\uparrow$ | $\mathcal{R}\uparrow$ | $\mathcal{R}\uparrow$ | $\mathcal{R}\uparrow$ | $\mathcal{R}\uparrow$ | $\mathcal{R}\uparrow$ |
| *Fixed Cont.* - PD Position | 100±6.25 | 63.2±0.599 | 89.3±1.2 | 92.6±2.05 | 100±0.208 | 99.9±0.03 | 100±1.53e-06 | 100±8.47 | 100±1.77 | 85±2.93 | 81.7±0.384 |
| *Fixed Cont.* - PD Position Delta | 2.57±1.74 | 2.86±0.092 | 100±1.84 | 100±1.78 | 2.54±0.0542 | 53.8±1.57 | 100±1.53e-06 | 0±0 | 40.8±3.23 | 15.2±7.6 | 96.3±0.219 |
| *Fixed Cont.* - PD Int. Velocity | 2.46±1.51 | 2.55±0.0967 | 73.1±0.844 | 86.6±1.33 | 0±0 | 49.7±1.55 | 86.5±2 | 0±0 | 0±0 | 0±0 | 93.7±2.77 |
| *Fixed Cont.* - Random | 0.0±0.0 | 0.0±0.0 | 0.0±0.0 | 0.0±0.0 | 0.0±0.0 | 0.0±0.0 | 0.0±0.0 | 0.0±0.0 | 0.0±0.0 | 0.0±0.0 | 0.0±0.0 |
| *Direct Torque Cont.* - High Freq. (500Hz) | 0±0 | 18±0.353 | 0.882±0.533 | 7.93±1.94 | 10.1±0.357 | 0±0 | 0±0 | 4.36±7.75 | 0±0 | 45.3±6.74 | 0±0 |
| *Direct Torque Cont.* - Low Freq. (31.25Hz) | 63±7.07 | 100±1.54 | 70.6±0.548 | 85.6±1.41 | 80±1.73 | 100±0.0311 | 100±1.53e-06 | 75.4±12.8 | 58.8±3.74 | 100±1.94 | 100±1.06 |
| **EvoControl (Full State)** | **285±52.7** | **161±19.5** | **190±40.5** | **126±9.29** | **105±29.9** | **101±0.522** | 100±0 | **349±27.4** | 97.9±70.9 | **106±2.16** | **169±58.1** |
| **EvoControl (Residual State)** | **161±49.5** | **188±1.91** | 95.7±5.8 | **153±13.7** | **175±160** | **100±0.559** | 100±0 | **351±54.3** | **112±1.46** | **105±2.97** | **204±60.3** |
| **EvoControl (Target + Proprio.)** | **293±108** | **134±21.4** | **202±134** | **165±36.2** | **140±76.2** | 98.8±5 | 76.9±99.4 | **338±45** | **114±3.76** | **106±2.18** | **179±30.8** |
| **EvoControl (Target)** | **248±25.5** | **165±31.1** | **175±86.6** | **152±31.8** | **150±148** | 99.3±2.17 | 100±0 | **356±14.6** | 95.6±23.8 | **105±2.87** | **148±31.3** |
| **EvoControl (Learned Gains)** | **214±12.7** | **111±11.2** | **128±69** | **125±72.2** | **102±8.52** | **100±0.201** | 100±0 | **327±81.4** | **115±0.874** | **105±2.67** | **229±56.9** |
| **EvoControl (Delta Position)** | **281±76.8** | **141±31.4** | **202±848** | **121±8.27** | 96.3±16.3 | **101±1.56** | 100±0 | **342±12.7** | 71.9±44.6 | 97.4±15.4 | **173±32.4** |

## J.5. Ablation: No annealing with PD Controller

Here we conduct an ablation of removing the gradual annealing with a PD controller throughout training. Specifically, to do this we set $\alpha = 0$, therefore meaning that within EvoControl the high-level policy $\rho$ output latent action $a_k$ directly goes into the initially un-trained low-level neural network policy $\beta$. We observe in Table 24, that EvoControl as presented in the main paper, the inclusion of annealing with a PD controller throughout learning does help EvoControl to achieve a higher normalized return on average compared to no annealing with a PD controller, as shown in the ablation. This provides empirical evidence for it's inclusion, which could be explained by the intuitive arguments presented in the paper, of helping the higher-level policy $\rho$ to learn a stable policy using an initial goal-tracking sub-policy of a PD controller, and then switch to an improved learned low-level controller throughout training. We also note, that although EvoControl performs well with the inclusion of the annealed PD controller, not having the PD controller, it also performs acceptably compared to the baselines. However, for best performance we recommend users to use the annealing with a PD controller.

Table 24: Main table of results (Table 3), with the ablation of EvoControl with no PD controller annealing (by setting $\alpha = 0$). Normalized evaluation return $\mathcal{R}$ for the benchmark methods, across each environment. EvoControl on average achieves a higher normalized evaluation return than the baselines of *fixed controllers* and *direct torque control*. Results are averaged over 384 random seeds, with $\pm$ indicating 95% confidence intervals. Returns are normalized to a 0-100 scale, where 0 represents a random policy, and 100 represents the highest reward achieved by a non-EvoControl baseline in each environment. Scores bolded are greater than 100.

| Same PPO high-level alg. $\rho$ with a Low-Level Policy $\beta$ of | Ant | Halfcheetah | Hopper | Humanoid | Humanoid Standup | Inverted Double Pendulum | Inverted Pendulum | Pusher | Reacher | Reacher 1D | Walker2D |
|---|---|---|---|---|---|---|---|---|---|---|---|
| | $\mathcal{R}\uparrow$ | $\mathcal{R}\uparrow$ | $\mathcal{R}\uparrow$ | $\mathcal{R}\uparrow$ | $\mathcal{R}\uparrow$ | $\mathcal{R}\uparrow$ | $\mathcal{R}\uparrow$ | $\mathcal{R}\uparrow$ | $\mathcal{R}\uparrow$ | $\mathcal{R}\uparrow$ | $\mathcal{R}\uparrow$ |
| *Fixed Cont.* - PD Position | 100±6.56 | 61.2±0.441 | 91.6±1.23 | 100±2.96 | 100±0.974 | 99.9±0.03 | 100±1.53e-06 | 100±8.47 | 100±1.8 | 85.2±2.87 | 75.7±0.633 |
| *Fixed Cont.* - PD Position Delta | 2.4±1.91 | 2.76±0.0888 | 100±1.35 | 96.6±1.71 | 2.96±0.0397 | 53.8±1.57 | 100±1.53e-06 | 0±0 | 40.9±3.23 | 15.2±7.6 | 90.2±0.239 |
| *Fixed Cont.* - PD Int. Velocity | 3.59±1.78 | 2.46±0.0932 | 74.7±0.903 | 83.4±1.13 | 0±0 | 49.7±1.55 | 86.5±2 | 0±0 | 0±0 | 0±0 | 85.9±2.55 |
| *Fixed Cont.* - Random | 0.0±0.0 | 0.0±0.0 | 0.0±0.0 | 0.0±0.0 | 0.0±0.0 | 0.0±0.0 | 0.0±0.0 | 0.0±0.0 | 0.0±0.0 | 0.0±0.0 | 0.0±0.0 |
| *Direct Torque Cont.* - High Freq. (500Hz) | 0±0 | 17.2±0.316 | 1.42±0.533 | 10.4±2.19 | 10.3±0.586 | 0±0 | 0±0 | 1.34±7.89 | 2.08±5.84 | 45.3±6.74 | 0±0 |
| *Direct Torque Cont.* - Low Freq. (31.25Hz) | 54.5±7.15 | 100±1.21 | 72±0.64 | 98±2.55 | 80.6±2.56 | 100±0.0311 | 100±1.53e-06 | 73.2±12.9 | 59.2±3.72 | 100±1.94 | 100±2.68 |
| **EvoControl (Full State)** | **368±73.2** | **157±18.3** | **274±20.6** | **123±19** | **116±18.4** | **101±0.859** | 100±0 | **362±12.5** | **114±7.51** | **106±2.75** | **203±137** |
| **EvoControl (Residual State)** | **182±16.1** | **182±6.31** | **101±5.54** | **170±18.2** | **212±145** | 99.2±1.25 | 100±0 | **375±94.8** | **106±25.8** | **104±3.42** | **205±57.4** |
| **EvoControl (Target + Proprio.)** | **319±35.1** | **168±14.7** | **171±155** | **165±7.19** | **165±150** | 99.7±1.19 | 100±0 | **353±28.4** | 96.8±78.6 | **105±4.71** | **178±63.7** |
| **EvoControl (Target)** | **293±87** | **162±23.5** | **283±250** | **164±21.2** | **205±147** | 99.6±0.949 | 100±0 | **353±43.4** | **112±1.73** | **105±1.65** | **188±88.2** |
| **EvoControl (Learned Gains)** | **266±104** | **113±10.5** | **206±302** | **150±15.1** | **117±2.44** | 99.5±2.48 | 100±0 | **330±17.8** | **116±1.62** | **105±2.45** | **196±118** |
| **EvoControl (Delta Position)** | **362±47.7** | **133±34.6** | **225±82.9** | **119±18** | **105±4.64** | **101±0.394** | 100±0 | **267±30.2** | 65.5±21 | 99.1±12.7 | **183±34.4** |
| Ablation - No PD controller annealing ($\alpha = 0$) - **EvoControl (Full State)** | **359±87** | **148±27.8** | **213±142** | **121±17.3** | **116±17.1** | **101±0.51** | 100±0 | **349±15** | 65.5±16 | **104±4.52** | **218±39.7** |
| Ablation - No PD controller annealing ($\alpha = 0$) - **EvoControl (Residual State)** | **164±195** | **121±27.7** | **160±181** | **166±28.2** | **122±25.6** | 99.7±1.47 | 100±0 | **366±86.2** | 68.4±43.5 | **102±7** | **185±109** |
| Ablation - No PD controller annealing ($\alpha = 0$) - **EvoControl (Target + Proprio.)** | **294±46** | **148±3.82** | **242±193** | **173±18.4** | **165±147** | 89.3±43.8 | 100±0 | **357±44.2** | 63.2±12.6 | **104±5.84** | **188±106** |
| Ablation - No PD controller annealing ($\alpha = 0$) - **EvoControl (Target)** | **298±31.5** | **145±71.4** | **121±109** | **174±8.98** | **128±16.6** | 68.7±78.1 | **100±0.0223** | **356±109** | 74.4±41.3 | 91.6±39.9 | **195±74.4** |
| Ablation - No PD controller annealing ($\alpha = 0$) - **EvoControl (Learned Gains)** | **255±58.2** | **140±19.9** | 95.7±127 | **156±23.9** | **119±5.95** | **100±0.554** | 100±0 | **340±49.4** | 87.9±40.1 | 92.7±4.95 | **179±117** |
| Ablation - No PD controller annealing ($\alpha = 0$) - **EvoControl (Delta Position)** | **328±110** | **151±67.6** | **255±116** | **127±2.02** | **120±8.73** | **101±0.365** | 100±0 | **334±30.9** | 60±13.3 | 96.3±13.3 | **236±10.8** |

## J.6. Ablation: Main Results for More High-level Steps

In the following, we increase the high-level number of steps that each baseline is trained for. Initially in the main results presented in the paper we trained all the results for 1M high-level $\rho$ policy steps. Therefore we ask the question, how do all the results compare if we run all the baselines for the high-level steps of 100M and 1B high-level steps. We tabulate these results in Tables 25 to 27.

Table 25: Additional Experiment. Training all the baselines for a larger amount of high-level $\rho$ policy steps of 10M steps. Normalized evaluation return $\mathcal{R}$ for the benchmark methods, across each environment. EvoControl on average achieves a higher normalized evaluation return than the baselines of *fixed controllers* and *direct torque control*. Results are averaged over 384 random seeds, with $\pm$ indicating 95% confidence intervals. Returns are normalized to a 0-100 scale, where 0 represents a random policy, and 100 represents the highest reward achieved by a non-EvoControl baseline in each environment. Scores bolded are greater than 100.

| Same PPO high-level alg. $\rho$ with a Low-Level Policy $\beta$ of | Ant | Halfcheetah | Hopper | Humanoid | Humanoid Standup | Inverted Double Pendulum | Inverted Pendulum | Pusher | Reacher | Reacher 1D | Walker2D |
| --- | --- | --- | --- | --- | --- | --- | --- | --- | --- | --- | --- |
| | $\mathcal{R}\uparrow$ | $\mathcal{R}\uparrow$ | $\mathcal{R}\uparrow$ | $\mathcal{R}\uparrow$ | $\mathcal{R}\uparrow$ | $\mathcal{R}\uparrow$ | $\mathcal{R}\uparrow$ | $\mathcal{R}\uparrow$ | $\mathcal{R}\uparrow$ | $\mathcal{R}\uparrow$ | $\mathcal{R}\uparrow$ |
| *Fixed Cont.* - PD Position | 60.8±2.01 | 68.9±0.858 | 100±1.23 | 100±0.281 | 100±0.308 | 100±0.033 | 89.6±1.48 | 100±4.99 | 93.8±2.48 | 84.9±2.56 | 100±0.775 |
| *Fixed Cont.* - PD Position Delta | 0±0 | 4.13±0.107 | 45.4±0.0546 | 86.5±1.22 | 1.4±0.0198 | 98.2±0.485 | 100±1.53e-06 | 0.226±7.09 | 65.5±1.68 | 25.7±7.17 | 46.5±0.0729 |
| *Fixed Cont.* - PD Int. Velocity | 0±0 | 4.15±0.145 | 45.9±0.0263 | 58.8±0.93 | 1.59±0.0629 | 13.4±0.43 | 99.6±0.244 | 0±0 | 0±0 | 9.68±7.5 | 53±0.887 |
| *Fixed Cont.* - Random | 0.0±0.0 | 0.0±0.0 | 0.0±0.0 | 0.0±0.0 | 0.0±0.0 | 0.0±0.0 | 0.0±0.0 | 0.0±0.0 | 0.0±0.0 | 0.0±0.0 | 0.0±0.0 |
| *Direct Torque Cont.* - High Freq. (500Hz) | 0.36±4.11 | 70±1.51 | 35±1.66 | 42.1±1.2 | 73.4±1.91 | 97.8±0.148 | 42.6±1.48 | 10.7±4.39 | 36.9±3.9 | 83.8±3.45 | 29.8±1.18 |
| *Direct Torque Cont.* - Low Freq. (31.25Hz) | 100±4.06 | 100±1.39 | 94.6±3.91 | 96.3±0.355 | 79.5±2.01 | 100±0.0336 | 100±1.53e-06 | 41.7±3.83 | 100±1.33 | 100±1.66 | 74.8±2.44 |
| **EvoControl (Full State)** | **208±4.54** | **231±1.11** | **187±0.629** | **104±0.802** | **118±0.172** | **102±8.45e-05** | 100±1.53e-06 | **223±3.06** | 97.1±0.738 | 99.7±1.4 | **222±1.12** |
| **EvoControl (Residual State)** | **113±3.74** | **208±1.23** | **128±4.11** | **124±0.865** | **228±4.5** | **101±0.0953** | 100±1.53e-06 | **228±3.03** | **103±0.646** | **101±0.634** | **150±0.501** |
| **EvoControl (Target + Proprio.)** | **178±3.88** | **230±0.842** | **177±0.467** | **169±0.922** | **206±6.11** | 83.6±1.65 | 100±0.000679 | **222±2.92** | 96.7±0.865 | 97.8±1.2 | **226±1.36** |
| **EvoControl (Target)** | **174±4.38** | **231±0.617** | **185±0.815** | **148±1.19** | **254±6.5** | 87.3±1.89 | 100±1.53e-06 | **224±2.84** | 92.7±1.08 | 99.5±1.04 | **203±1.72** |
| **EvoControl (Learned Gains)** | **147±4.81** | **194±1.13** | **172±0.682** | **110±1.06** | **112±0.506** | **102±0.0572** | 100±1.53e-06 | **200±4.62** | **102±0.651** | 99.8±0.836 | **186±1.55** |
| **EvoControl (Delta Position)** | **187±4.08** | **209±1.55** | **185±0.188** | 98.6±0.824 | **115±0.389** | **102±6.05e-05** | 100±1.53e-06 | **155±5.18** | 93.4±1.03 | 98.4±0.777 | **179±0.635** |

Table 26: Additional Experiment. Training all the baselines for a larger amount of high-level $\rho$ policy steps of 100M steps. Normalized evaluation return $\mathcal{R}$ for the benchmark methods, across each environment. EvoControl on average achieves a higher normalized evaluation return than the baselines of *fixed controllers* and *direct torque control*. Results are averaged over 384 random seeds, with $\pm$ indicating 95% confidence intervals. Returns are normalized to a 0-100 scale, where 0 represents a random policy, and 100 represents the highest reward achieved by a non-EvoControl baseline in each environment. Scores bolded are greater than 100.

| Same PPO high-level alg. $\rho$ with a Low-Level Policy $\beta$ of | Ant | Halfcheetah | Hopper | Humanoid | Humanoid Standup | Inverted Double Pendulum | Inverted Pendulum | Pusher | Reacher | Reacher 1D | Walker2D |
| --- | --- | --- | --- | --- | --- | --- | --- | --- | --- | --- | --- |
| | $\mathcal{R}\uparrow$ | $\mathcal{R}\uparrow$ | $\mathcal{R}\uparrow$ | $\mathcal{R}\uparrow$ | $\mathcal{R}\uparrow$ | $\mathcal{R}\uparrow$ | $\mathcal{R}\uparrow$ | $\mathcal{R}\uparrow$ | $\mathcal{R}\uparrow$ | $\mathcal{R}\uparrow$ | $\mathcal{R}\uparrow$ |
| *Fixed Cont.* - PD Position | 62.5±1.4 | 38.6±0.485 | 85.1±2.69 | 100±0.289 | 36±2.01 | 100±0.19 | 98.9±0.418 | 100±4.58 | 100±1.97 | 85.5±2.5 | 75.7±2.56 |
| *Fixed Cont.* - PD Position Delta | 2.34±0.431 | 4.14±0.0897 | 53.9±0.362 | 94.8±1.17 | 1.09±0.0471 | 99.9±0.25 | 100±1.53e-06 | 85.6±5.35 | 78.1±1.84 | 20.4±7.39 | 44.3±0.0682 |
| *Fixed Cont.* - PD Int. Velocity | 2.61±1.03 | 4.08±0.132 | 24.9±2.14 | 68.4±1.68 | 35±1.03 | 34.6±2.79 | 44.6±4.01 | 37.7±6.42 | 0±0 | 12.1±7 | 52.4±1.14 |
| *Fixed Cont.* - Random | 0.0±0.0 | 0.0±0.0 | 0.0±0.0 | 0.0±0.0 | 0.0±0.0 | 0.0±0.0 | 0.0±0.0 | 0.0±0.0 | 0.0±0.0 | 0.0±0.0 | 0.0±0.0 |
| *Direct Torque Cont.* - High Freq. (500Hz) | 29.8±4.11 | 100±1.82 | 77.1±2.23 | 75.1±2.87 | 100±2.88 | 86.5±1.06 | 63.2±2.86 | 9.06±6.19 | 84±2.74 | 65.3±6.04 | 72.7±2.65 |
| *Direct Torque Cont.* - Low Freq. (31.25Hz) | 100±3.6 | 60.6±1.53 | 100±5.62 | 96.3±0.501 | 77.7±1.17 | 99.9±0.281 | 100±1.53e-06 | 72.5±4.49 | 92.7±3.2 | 100±2.19 | 100±3.95 |
| **EvoControl (Full State)** | **188±3.19** | **191±0.581** | **227±.36** | **159±1.64** | **199±3.11** | **100±3.24e-05** | 100±1.53e-06 | **278±3.5** | **102±0.708** | **102±0.577** | **264±1.06** |
| **EvoControl (Residual State)** | **109±3.06** | **151±2.44** | **186±1.95** | **163±0.807** | **194±6.19** | **100±0.000688** | 100±1.53e-06 | **284±3.14** | **104±1.47** | **102±0.826** | **211±3.09** |
| EvoControl (Target + Proprio.) | **155±3.15** | **192±0.605** | **216±1.79** | **192±1** | **241±4.83** | 96.7±0.226 | 100±0.0112 | **283±2.86** | **103±0.782** | 96.8±1.3 | **257±1.58** |
| EvoControl (Target) | **147±2.77** | **192±0.499** | **213±2.3** | **189±1.66** | **261±2.19** | 96.4±0.314 | 98.9±0.791 | **279±3.23** | 96.9±1.03 | 88.1±2.05 | **260±0.723** |
| **EvoControl (Learned Gains)** | **117±3.68** | **151±1.37** | **215±0.793** | **123±0.874** | 63.6±2.6 | **100±4.2e-05** | 100±1.53e-06 | **259±4.17** | **103±0.844** | **101±0.848** | **180±2.09** |
| **EvoControl (Delta Position)** | **180±3.45** | **186±1.03** | **225±1.32** | **122±1.86** | **111±0.265** | **100±3.4e-05** | 100±1.53e-06 | **253±5.79** | **104±0.65** | **102±0.586** | **207±2.45** |

Table 27: Additional Experiment. Training all the baselines for a larger amount of high-level $\rho$ policy steps of 1B steps. Normalized evaluation return $\mathcal{R}$ for the benchmark methods, across each environment. EvoControl on average achieves a higher normalized evaluation return than the baselines of *fixed controllers* and *direct torque control*. Results are averaged over 384 random seeds, with $\pm$ indicating 95% confidence intervals. Returns are normalized to a 0-100 scale, where 0 represents a random policy, and 100 represents the highest reward achieved by a non-EvoControl baseline in each environment. Scores bolded are greater than 100.

| Same PPO high-level alg. $\rho$ with a Low-Level Policy $\beta$ of | Ant | Halfcheetah | Hopper | Humanoid | Humanoid Standup | Inverted Double Pendulum | Inverted Pendulum | Pusher | Reacher | Reacher 1D | Walker2D |
|---|---|---|---|---|---|---|---|---|---|---|---|
| | $\mathcal{R}\uparrow$ | $\mathcal{R}\uparrow$ | $\mathcal{R}\uparrow$ | $\mathcal{R}\uparrow$ | $\mathcal{R}\uparrow$ | $\mathcal{R}\uparrow$ | $\mathcal{R}\uparrow$ | $\mathcal{R}\uparrow$ | $\mathcal{R}\uparrow$ | $\mathcal{R}\uparrow$ | $\mathcal{R}\uparrow$ |
| *Fixed Cont.* - PD Position | 45.4±2.72 | 29.8±0.662 | 65.8±1.03 | 100±0.542 | 100±0.959 | 98.1±0.743 | 98.2±0.595 | 100±1.6 | 100±1.84 | 100±2.82 | 50.2±2.42 |
| *Fixed Cont.* - PD Position Delta | 1.63±0.459 | 4.2±0.0834 | 60.9±0.057 | 90.9±1.02 | 1.13±0.0683 | 100±0.353 | 100±1.53e-06 | 46.5±5.03 | 76.6±1.77 | 22.1±8.77 | 33.2±0.0273 |
| *Fixed Cont.* - PD Int. Velocity | 4.86±0.755 | 0±0 | 17.2±2.89 | 56.5±1.92 | 36.5±0.488 | 83.6±2.23 | 19.6±3.86 | 30.4±2.04 | 0±0 | 6.56±8.56 | 4.81±1.27 |
| *Fixed Cont.* - Random | 0.0±0.0 | 0.0±0.0 | 0.0±0.0 | 0.0±0.0 | 0.0±0.0 | 0.0±0.0 | 0.0±0.0 | 0.0±0.0 | 0.0±0.0 | 0.0±0.0 | 0.0±0.0 |
| *Direct Torque Cont.* - High Freq. (500Hz) | 40.8±3.72 | 0±0 | 100±1.74 | 86.3±2.36 | 52.9±3.39 | 93.8±0.622 | 95.3±0.751 | $\pm$ | 88.6±2.27 | 82.9±6.85 | $\pm$ |
| *Direct Torque Cont.* - Low Freq. (31.25Hz) | 100±3.25 | 100±0.983 | $\pm$ | 93.4±0.18 | 74.4±1.41 | 95.9±1.17 | 98.4±0.557 | 86.9±2.08 | $\pm$ | $\pm$ | 100±0.646 |
| **EvoControl (Full State)** | **220±4.55** | **197±0.54** | **285±0.424** | **159±0.844** | **185±1.02** | **101±5.28e-05** | **100±1.53e-06** | **106±1.63** | **104±0.61** | **120±0.664** | **196±0.435** |
| **EvoControl (Residual State)** | **134±3.24** | **149±2.34** | **270±1.21** | **159±1.79** | **365±0.986** | **101±0.000788** | **100±1.88e-06** | **111±1.31** | 95.3±2.41 | **120±0.727** | **155±0.924** |
| **EvoControl (Target + Proprio.)** | **182±3.71** | **194±0.564** | **269±0.707** | **183±1.01** | **340±4.46** | 96.5±0.391 | **100±1.88e-06** | **114±1.14** | 98.7±1.19 | **112±1.43** | **181±0.835** |
| **EvoControl (Target)** | **174±4.07** | **194±0.527** | **269±0.71** | **180±0.805** | **355±1.42** | 97.5±0.179 | **100±1.53e-06** | **113±1.22** | 96.5±1.21 | **112±1.5** | **198±0.428** |
| **EvoControl (Learned Gains)** | 87.3±1.98 | 39.3±0.435 | 80.7±0.277 | **103±0.745** | 83.1±2.83 | **101±0.0651** | 99.9±0.204 | 76.9±3.27 | 94.7±2.51 | **103±2.44** | 78.9±3.05 |
| **EvoControl (Delta Position)** | **215±3.81** | **195±0.499** | **269±1.15** | **121±0.526** | **158±1.55** | **101±4.84e-05** | **100±1.53e-06** | **112±1.35** | **108±0.478** | **120±0.667** | **167±0.592** |

## J.7. Main Table of Results Additional Metrics

For the main table of results presented in the paper (Table 3), we also provide un-normalized results in Table 28 and the time taken to train the policies in Table 29. Regarding the training time, the wall-clock time for EvoControl can be substantially improved, as the PPO implementation we use to train the high-level policies is implemented in Jax, and was pre-compiled, across a batch of environments, using Jax based environments (Brax). Whereas the ES could further be compiled, however for our implementation it was not, and only the population of rollouts was compiled in Jax. Ideally for optimal performance the entire ES step could be compiled in Jax leading to speed improvements, however given that all the baselines, including EvoControl could finish training their policies within a time interval of approximately one hour, further optimization was not necessary.

Table 28: Un-normalized evaluation return $R$ for the benchmark methods, across each environment. EvoControl on average achieves a higher evaluation return $R$ than the baselines of *fixed controllers* and *direct torque control*. Results are averaged over 384 random seeds, with $\pm$ indicating 95% confidence intervals.

| Same PPO high-level alg. $\rho$ with a Low-Level Policy $\beta$ of | Ant | Halfcheetah | Hopper | Humanoid | Humanoid Standup | Inverted Double Pendulum | Inverted Pendulum | Pusher | Reacher | Reacher 1D | Walker2D |
|---|---|---|---|---|---|---|---|---|---|---|---|
| | $\mathcal{R}\uparrow$ | $\mathcal{R}\uparrow$ | $\mathcal{R}\uparrow$ | $\mathcal{R}\uparrow$ | $\mathcal{R}\uparrow$ | $\mathcal{R}\uparrow$ | $\mathcal{R}\uparrow$ | $\mathcal{R}\uparrow$ | $\mathcal{R}\uparrow$ | $\mathcal{R}\uparrow$ | $\mathcal{R}\uparrow$ |
| *Fixed Cont.* - PD Position | 1.47e+03±31 | 1.09e+03±8.22 | 1.05e+03±12.7 | 3.52e+03±74 | 1.88e+05±1.43e+03 | 9.21e+03±1.97 | 1e+03±0 | -2.36e+03±26.2 | -46.6±2.44 | -20.3±2.08 | 895±7.13 |
| *Fixed Cont.* - PD Position Delta | 1.01e+03±9.04 | 3.6±1.66 | 1.14e+03±13.9 | 3.43e+03±42.9 | 4.51e+04±58.3 | 6.18e+03±103 | 1e+03±0 | -2.73e+03±26.6 | -127±4.38 | -70.9±5.5 | 1.06e+03±2.69 |
| *Fixed Cont.* - PD Int. Velocity | 1.01e+03±8.43 | -2.14±1.74 | 876±9.33 | 3.1e+03±28.2 | 4.05e+04±17.5 | 5.91e+03±102 | 898±15.1 | -2.67e+03±26.2 | -229±7.85 | -84.7±5.62 | 1.01e+03±28.7 |
| *Fixed Cont.* - Random | 0.0±0.0 | 0.0±0.0 | 0.0±0.0 | 0.0±0.0 | 0.0±0.0 | 0.0±0.0 | 0.0±0.0 | 0.0±0.0 | 0.0±0.0 | 0.0±0.0 | 0.0±0.0 |
| *Direct Torque Cont.* - High Freq. (500Hz) | 920±41.1 | 273±5.89 | 119±5.5 | 1.28e+03±54.8 | 5.59e+04±862 | 722±1.23 | 106±3.72 | -2.67e+03±24.4 | -179±7.92 | -49.1±4.87 | -36.9±24.9 |
| *Direct Torque Cont.* - Low Freq. (31.25Hz) | 1.26e+03±33.8 | 1.82e+03±22.5 | 848±6.61 | 3.47e+03±63.6 | 1.59e+05±3.76e+03 | 9.22e+03±2.05 | 1e+03±0 | -2.44e+03±39.9 | -102±5.04 | -9.55±1.41 | 1.17e+03±30.1 |
| **EvoControl (Full State)** | 2.74e+03±347 | 2.88e+03±341 | 2.94e+03±212 | 4.08e+03±474 | 2.11e+05±2.71e+04 | 9.28e+03±56.5 | 1e+03±0 | -1.55e+03±38.7 | -27.6±10.2 | -5.44±1.99 | 2.32e+03±1.54e+03 |
| **EvoControl (Residual State)** | 1.86e+03±76.2 | 3.35e+03±118 | 1.15e+03±57.3 | 5.27e+03±456 | 3.52e+05±2.13e+05 | 9.17e+03±82.2 | 1e+03±0 | -1.51e+03±293 | -38.5±35 | -6.69±2.47 | 2.35e+03±646 |
| **EvoControl (Target + Proprio.)** | 2.51e+03±166 | 3.08e+03±274 | 1.88e+03±1.6e+03 | 5.15e+03±180 | 2.83e+05±2.21e+05 | 9.2e+03±78.3 | 1e+03±0 | -1.58e+03±88 | -50.9±107 | -5.74±3.4 | 2.05e+03±716 |
| **EvoControl (Target)** | 2.38e+03±412 | 2.96e+03±438 | 3.03e+03±2.68e+03 | 5.11e+03±530 | 3.42e+05±2.16e+05 | 9.2e+03±62.4 | 1e+03±0 | -1.58e+03±134 | -30.8±2.35 | -5.78±1.19 | 2.15e+03±993 |
| **EvoControl (Learned Gains)** | 2.26e+03±492 | 2.06e+03±195 | 2.23e+03±3.12e+03 | 4.77e+03±378 | 2.13e+05±3.58e+03 | 9.19e+03±163 | 1e+03±0 | -1.65e+03±55 | -24.9±2.19 | -6±1.77 | 2.25e+03±1.32e+03 |
| **EvoControl (Delta Position)** | 2.71e+03±226 | 2.43e+03±644 | 2.43e+03±856 | 4e+03±450 | 1.95e+05±6.82e+03 | 9.27e+03±26 | 1e+03±0 | -1.84e+03±93.5 | -93.4±28.5 | -10.2±9.16 | 2.1e+03±387 |

Table 29: Time Taken to train in minutes for each baseline against each environment, for the main table of results in Table 3. All the baselines including EvoControl can train their policies on average within an hour. Results are averaged over 384 random seeds, with ± indicating 95% confidence intervals. As the random baseline does not perform any training, we put a placeholder of 0 for it.

| Same PPO high-level alg. $\rho$ with a Low-Level Policy $\beta$ of | Ant | Halfcheetah | Hopper | Humanoid | Humanoid Standup | Inverted Double Pendulum | Inverted Pendulum | Pusher | Reacher | Reacher 1D | Walker2D |
|---|---|---|---|---|---|---|---|---|---|---|---|
| | $\mathcal{R}\uparrow$ | $\mathcal{R}\uparrow$ | $\mathcal{R}\uparrow$ | $\mathcal{R}\uparrow$ | $\mathcal{R}\uparrow$ | $\mathcal{R}\uparrow$ | $\mathcal{R}\uparrow$ | $\mathcal{R}\uparrow$ | $\mathcal{R}\uparrow$ | $\mathcal{R}\uparrow$ | $\mathcal{R}\uparrow$ |
| *Fixed Cont.* - PD Position | 0.671±0.0112 | 1.15±0.0309 | 0.886±0.0451 | 0.574±0.281 | 1.36±0.297 | 0.536±0.0173 | 0.503±0.011 | 0.799±0.285 | 0.587±0.00373 | 0.519±0.00276 | 1.06±0.0154 |
| *Fixed Cont.* - PD Position Delta | 0.668±0.0118 | 1.08±0.00826 | 0.879±0.0303 | 0.576±0.285 | 1.19±0.283 | 0.542±0.00226 | 0.475±0.00815 | 0.847±0.332 | 0.595±0.0164 | 0.473±0.0101 | 1.16±0.0178 |
| *Fixed Cont.* - PD Int. Velocity | 0.662±0.0253 | 1.06±0.00543 | 0.888±0.0169 | 0.574±0.3 | 1.11±0.303 | 0.53±0.0161 | 0.506±0.0181 | 0.679±0.27 | 0.573±0.0182 | 0.503±0.0125 | 1.2±0.0349 |
| *Fixed Cont.* - Random | 0±0 | 0±0 | 0±0 | 0±0 | 0±0 | 0±0 | 0±0 | 0±0 | 0±0 | 0±0 | 0±0 |
| *Direct Torque Cont.* - High Freq. (500Hz) | 0.597±0.0392 | 0.963±0.0137 | 0.738±0.0213 | 0.513±0.273 | 1.06±0.277 | 0.457±0.00473 | 0.424±0.0121 | 0.679±0.291 | 0.585±0.02 | 0.501±0.00844 | 0.899±0.0231 |
| *Direct Torque Cont.* - Low Freq. (31.25Hz) | 0.663±0.041 | 1.14±0.0165 | 0.908±0.00972 | 0.582±0.277 | 1.4±0.281 | 0.583±0.016 | 0.513±0.0115 | 0.917±0.274 | 0.736±0.0267 | 0.58±0.0301 | 1.09±0.0476 |
| **EvoControl (Full State)** | 12.8±0.0925 | 23.3±0.11 | 14.8±0.0829 | 20.4±0.505 | 77.3±0.481 | 8.33±0.0986 | 6.93±0.0438 | 31.4±1.34 | 5.74±0.0747 | 4.6±0.0386 | 25.4±0.226 |
| **EvoControl (Residual State)** | 11.2±0.0999 | 23.2±0.106 | 14.3±0.0904 | 18±0.466 | 76.1±0.407 | 7.94±0.0885 | 7±0.0656 | 31.1±0.725 | 5.84±0.331 | 4.59±0.00888 | 23.7±0.112 |
| **EvoControl (Target + Proprio.)** | 12.7±0.0899 | 23.1±0.0987 | 14.5±0.103 | 19.5±0.499 | 76.7±0.205 | 8.25±0.0224 | 6.89±0.00579 | 31.5±0.256 | 5.71±0.231 | 4.55±0.0514 | 24.1±0.154 |
| **EvoControl (Target)** | 11.4±0.0776 | 22.9±0.156 | 14.5±0 | 19.4±0.467 | 76.7±0.399 | 8.33±0.139 | 7.09±0.0406 | 30.7±1.27 | 5.66±0.119 | 4.55±0.0235 | 24±0.279 |
| **EvoControl (Learned Gains)** | 12.9±0.106 | 23.1±0.0689 | 13.3±1.06 | 20.3±0.517 | 76.8±0.173 | 8.09±0.0643 | 7.13±0.0462 | 27.4±1.52 | 5.64±0.16 | 4.51±0.0625 | 25.3±0.34 |
| **EvoControl (Delta Position)** | 12.8±0.124 | 22.3±0.162 | 14.8±0.175 | 20.5±0.479 | 73.2±0.733 | 8.39±0.0537 | 6.93±0.0197 | 29.2±1.25 | 5.72±0.151 | 4.59±0.0536 | 25.7±0.0329 |

## J.8. Learning Curves for All Baselines

We provide the learning curves for all environments presented in the main table of results in the paper, that of Table 3. Specifically, we provide these plots in Figures 5 to 14. We observe that EvoControl on average consistently outperforms the non-EvoControl baselines and achieves a higher evaluation return.

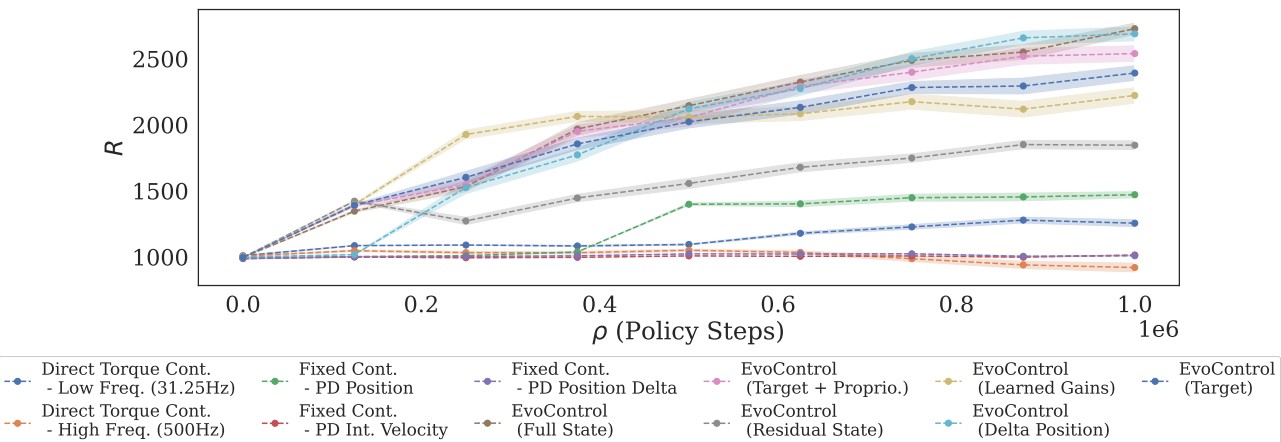

Figure 5: Evaluation return $R$ versus $\rho$ policy steps on Ant, for main table of results Table 3.

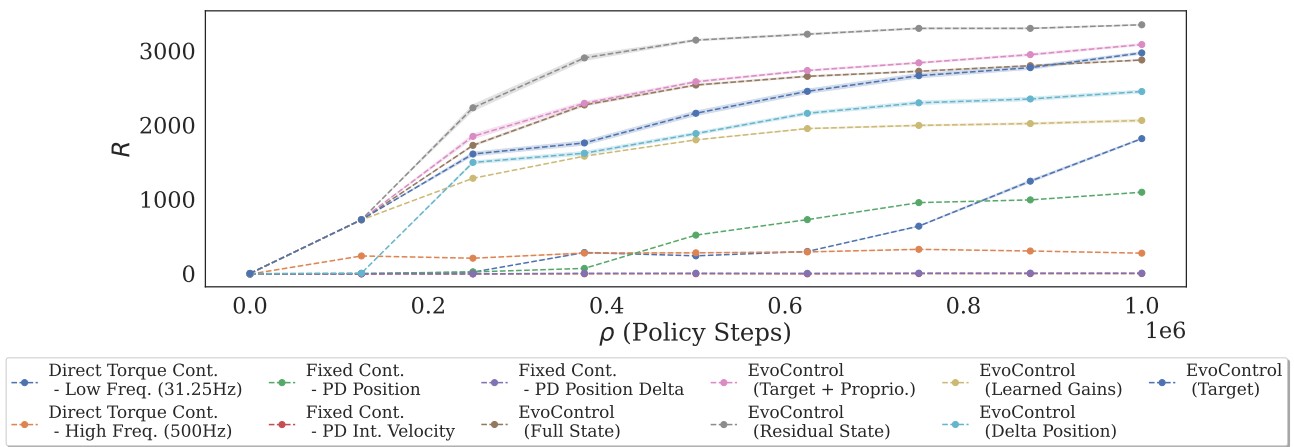

Figure 6: Evaluation return $R$ versus $\rho$ policy steps on Halfcheetah, for main table of results Table 3.

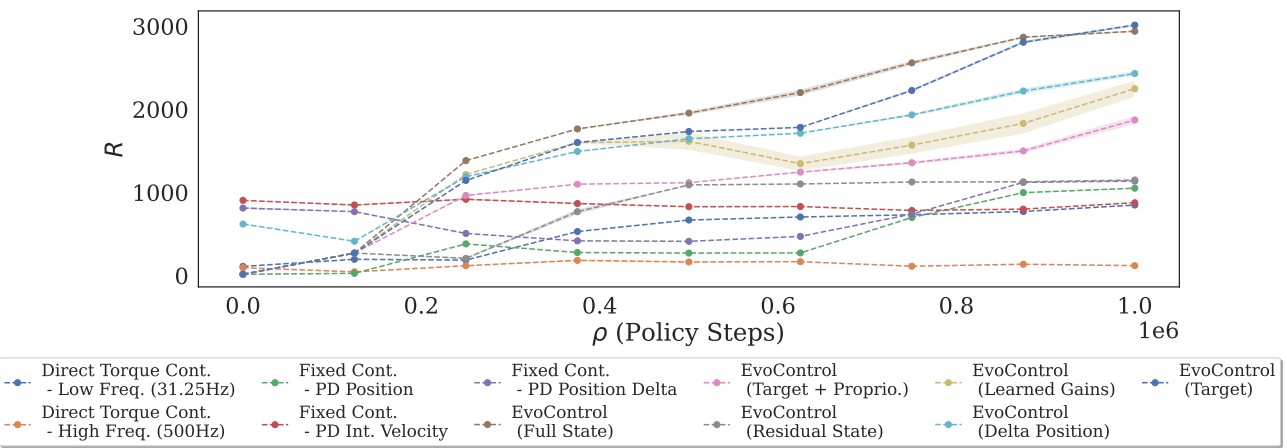

Figure 7: Evaluation return $R$ versus $\rho$ policy steps on Hopper, for main table of results Table 3.

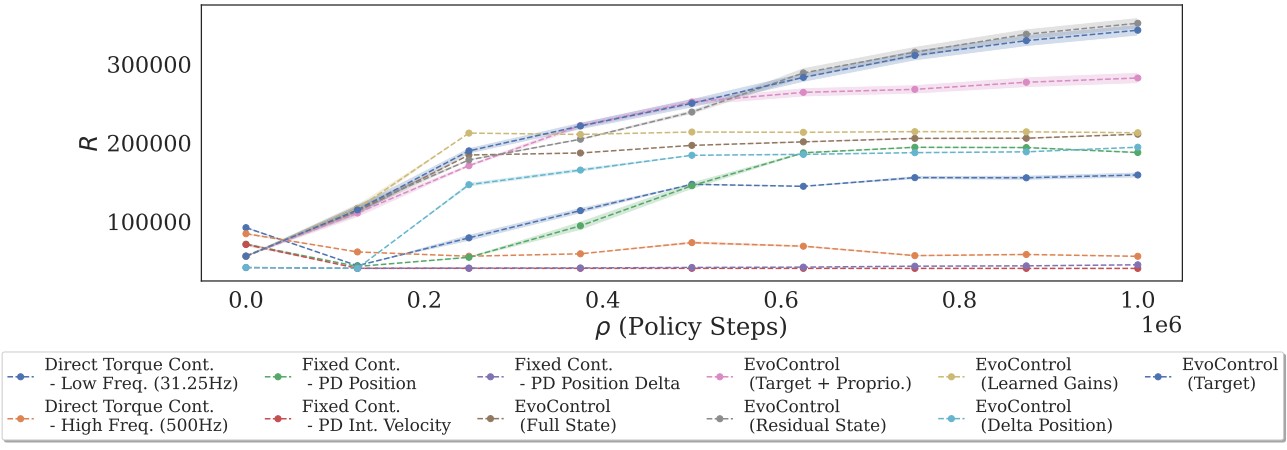

Figure 8: Evaluation return $R$ versus $\rho$ policy steps on Humanoid Standup, for main table of results Table 3.

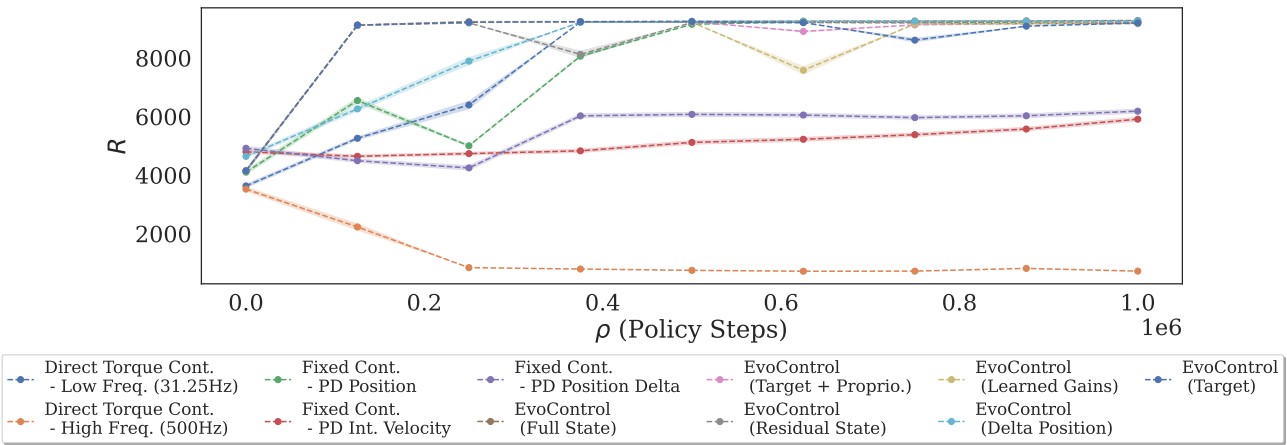

Figure 9: Evaluation return $R$ versus $\rho$ policy steps on Inverted Double Pendulum, for main table of results Table 3.

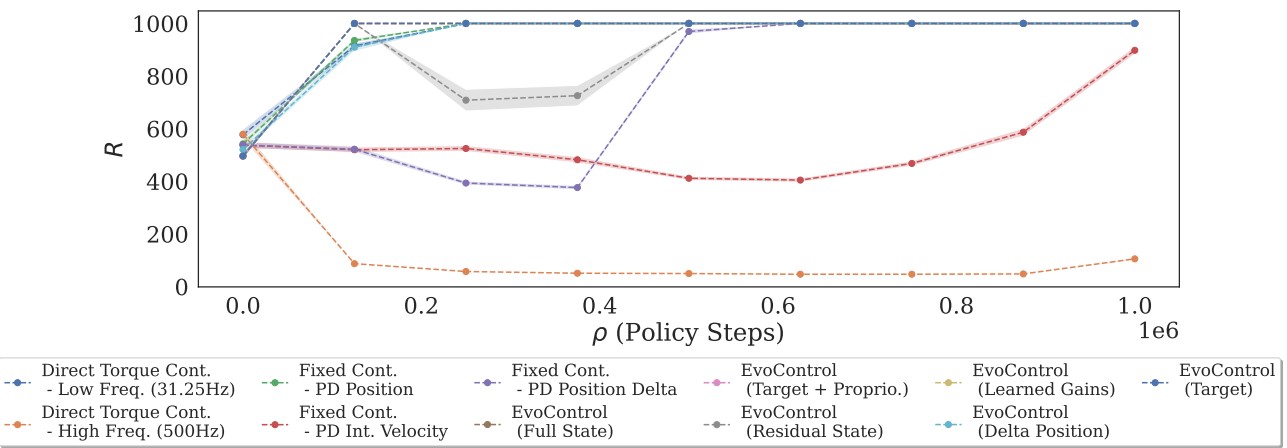

Figure 10: Evaluation return $R$ versus $\rho$ policy steps on Inverted Pendulum, for main table of results Table 3.

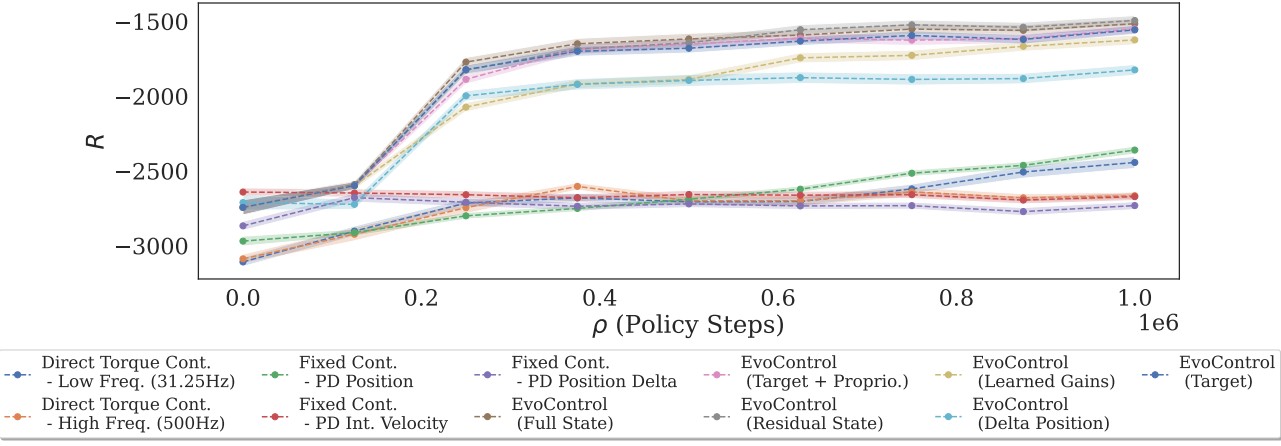

Figure 11: Evaluation return $R$ versus $\rho$ policy steps on Pusher, for main table of results Table 3.

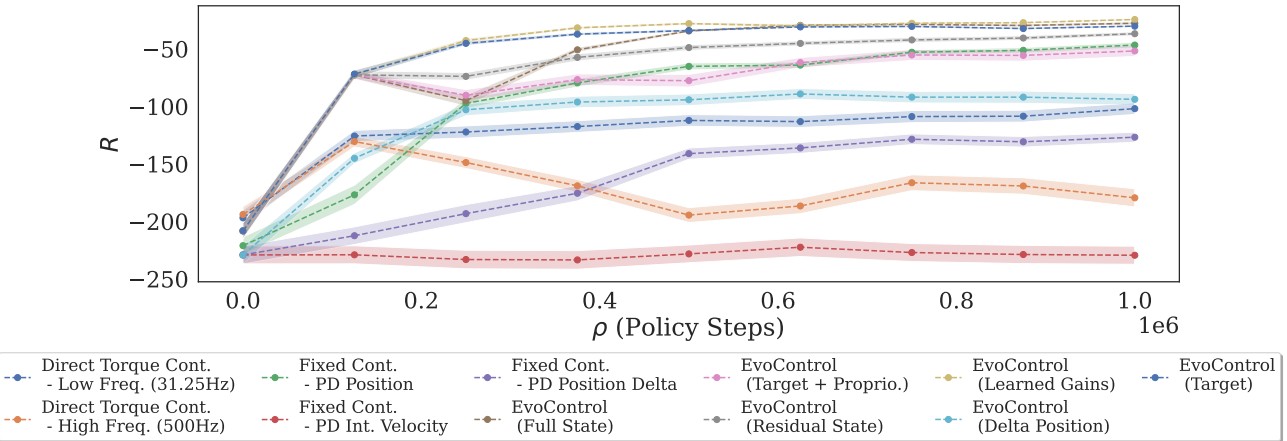

Figure 12: Evaluation return $R$ versus $\rho$ policy steps on Reacher, for main table of results Table 3.

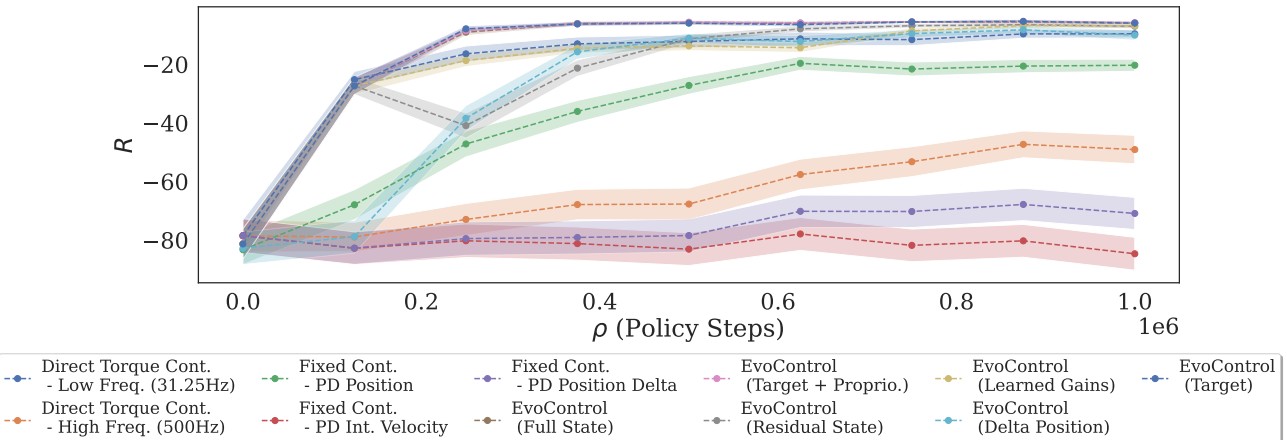

Figure 13: Evaluation return $R$ versus $\rho$ policy steps on Reacher 1D, for main table of results Table 3.

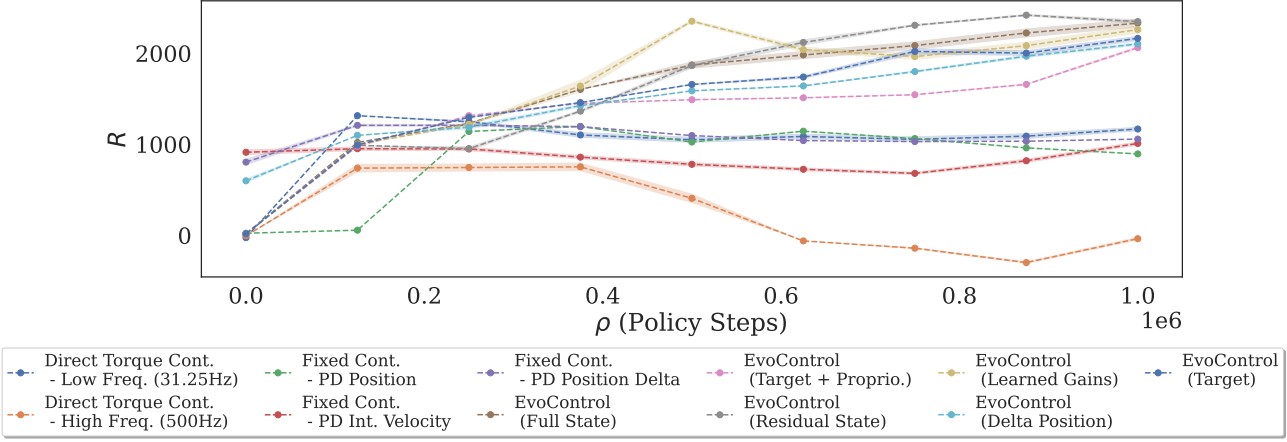

Figure 14: Evaluation return $R$ versus $\rho$ policy steps on Walker2D, for main table of results Table 3.

## J.9. Ablation: Training High-Level Policy with Evolution Strategies

In the following we perform an ablation whereby we train the high-level policy with Evolution Strategies (ES) rather than PPO. We provide two variations: first, using ES to train the high-level policy with the same fixed low-level controllers for all non-EvoControl baselines (Table 30), and second, within EvoControl modifying training the high-level policy with ES instead of PPO.

First, we replace the high-level PPO policy, keeping the same high-level policy architecture and network as originally used in the main paper, and instead using the same ES as was used to train the low-level policy. For this we provide two versions, A) that of training the high-level policy with the same number of high-level $\rho$ steps as the core baselines presented in the paper, with 1M $\rho$ steps—labelled *ES (1M $\rho$ steps)*. Specifically, this uses $es\_rollouts = 1, K = 8, es\_pop\_size = 256, es\_sub\_generations = 8$. Second, B) training the high-level policy with the same number of environment steps as was used to train the low-level policy within EvoControl—labelled *ES (same as EvoControl low-level)*. Specifically this uses more low-level environment steps than A), with $es\_rollouts = 16, K = 8, es\_pop\_size = 512, es\_sub\_generations = 8$. The results are tabulated alongside the original main table of results for ease of comparison in Table 30. We observe on average that EvoControl still outperforms the competitive baselines, and on average achieves a higher normalized evaluation return $\mathcal{R}$ than the baselines of *fixed-controllers* and *direct torque control*.

Table 30: Ablation. Training the high-level policy with ES instead of PPO. Normalized evaluation return $\mathcal{R}$ for the benchmark methods, across each environment. EvoControl on average achieves a higher normalized evaluation return than the baselines of *fixed controllers* and *direct torque control*. Results are averaged over 384 random seeds, with $\pm$ indicating 95% confidence intervals. Returns are normalized to a 0-100 scale, where 0 represents a random policy, and 100 represents the highest reward achieved by a non-EvoControl baseline in each environment. Scores bolded are greater than 100.

| Method Name | High-level $\rho$ with | Low-level $\beta$ with | Ant $\mathcal{R}\uparrow$ | Halfcheetah $\mathcal{R}\uparrow$ | Hopper $\mathcal{R}\uparrow$ | Humanoid $\mathcal{R}\uparrow$ | Humanoid Standup $\mathcal{R}\uparrow$ | Inverted Double Pendulum $\mathcal{R}\uparrow$ | Inverted Pendulum $\mathcal{R}\uparrow$ | Pusher $\mathcal{R}\uparrow$ | Reacher $\mathcal{R}\uparrow$ | Reacher 1D $\mathcal{R}\uparrow$ | Walker2D $\mathcal{R}\uparrow$ |
|---|---|---|---|---|---|---|---|---|---|---|---|---|---|
| *Fixed Cont. - PD Position* | PPO | PD Position | 100±6.25 | 63.2±0.599 | 89.3±1.2 | 92.6±2.05 | 100±0.208 | 99.9±0.03 | 100±1.53e-06 | 100±8.47 | 100±1.77 | 85±2.93 | 81.7±0.384 |
| *Fixed Cont. - PD Position Delta* | PPO | PD Position Delta | 2.57±1.74 | 2.86±0.092 | 100±1.84 | 100±1.78 | 2.54±0.0542 | 53.8±1.57 | 100±1.53e-06 | 0±0 | 40.8±3.23 | 15.2±7.6 | 96.3±0.219 |
| *Fixed Cont. - PD Int. Velocity* | PPO | PD Int. Velocity | 2.46±1.51 | 2.55±0.0967 | 73.1±0.844 | 86.6±1.33 | 0±0 | 49.7±1.55 | 86.5±2 | 0±0 | 0±0 | 0±0 | 93.7±2.77 |
| *Fixed Cont. - Random* | Random | Direct Torque | 0.0±0.0 | 0.0±0.0 | 0.0±0.0 | 0.0±0.0 | 0.0±0.0 | 0.0±0.0 | 0.0±0.0 | 0.0±0.0 | 0.0±0.0 | 0.0±0.0 | 0.0±0.0 |
| *Direct Torque Cont. - High Freq. (500Hz)* | PPO | Direct Torque | 0±0 | 18±0.353 | 0.882±0.533 | 7.93±1.94 | 10.1±0.357 | 0±0 | 0±0 | 4.36±7.75 | 0±0 | 45.3±6.74 | 0±0 |
| *Direct Torque Cont. - Low Freq. (31.25Hz)* | PPO | Direct Torque | 63±7.07 | 100±1.54 | 70.6±0.548 | 85.6±1.41 | 80±1.73 | 100±0.0311 | 100±1.53e-06 | 75.4±12.8 | 58.8±3.74 | 100±1.94 | 100±1.06 |
| **EvoControl (Full State)** | PPO | ES | **368±73.2** | **157±18.3** | **274±20.6** | **123±19** | **116±18.4** | **101±0.859** | **100±0** | **362±12.5** | **114±7.51** | **106±2.75** | **203±137** |
| **EvoControl (Residual State)** | PPO | ES | **182±16.1** | **182±6.31** | **101±5.54** | **170±18.2** | **212±145** | 99.2±1.25 | **100±0** | **375±94.8** | **106±25.8** | **104±3.42** | **205±57.4** |
| **EvoControl (Target + Proprio.)** | PPO | ES | **319±35.1** | **168±14.7** | **171±155** | **165±7.19** | **165±150** | 99.7±1.19 | **100±0** | **353±28.4** | 96.8±78.6 | **105±4.71** | **178±63.7** |
| **EvoControl (Target)** | PPO | ES | **293±87** | **162±23.5** | **283±250** | **164±21.2** | **205±147** | 99.6±0.949 | **100±0** | **353±43.4** | **112±1.73** | **105±1.65** | **188±88.2** |
| **EvoControl (Learned Gains)** | PPO | ES | **266±104** | **113±10.5** | **206±302** | **150±15.1** | **117±2.44** | 99.5±2.48 | **100±0** | **330±17.8** | **116±1.62** | **105±2.45** | **196±118** |
| **EvoControl (Delta Position)** | PPO | ES | **362±47.7** | **133±34.6** | **225±82.9** | **119±18** | **105±4.64** | **101±0.394** | **100±0** | **267±30.2** | 65.5±21 | 99.1±12.7 | **183±34.4** |
| *Ablation: Fixed Cont. - PD Position* | ES (1M $\rho$ steps) | PD Position | 136±1.02 | 52.5±0.881 | 104±1.1 | 154±0.149 | 118±0.205 | 101±0.0186 | 100±1.08e-06 | 390±1.33 | 69.1±0.37 | 86.7±0.179 | 179±1.18 |
| *Ablation: Fixed Cont. - PD Position Delta* | ES (1M $\rho$ steps) | PD Position Delta | 36.1±0.185 | 4.48±0.00862 | 133±0.535 | 137±0.738 | 4.61±0.00116 | 97.7±0.782 | 100±0.0594 | 279±3.12 | 52±0.174 | 12.2±0.83 | 126±0.512 |
| *Ablation: Fixed Cont. - PD Int. Velocity* | ES (1M $\rho$ steps) | PD Int. Velocity | 128±2.31 | 67.6±0.85 | 96±0.941 | 156±0.381 | 120±0.21 | 97.9±1.22 | 55.2±1.81 | 256±1.19 | 61.2±0.179 | 59.7±0.869 | 130±1.73 |
| *Ablation: Fixed Cont. - Random* | Random | Direct Torque | 0.0±0.0 | 0.0±0.0 | 0.0±0.0 | 0.0±0.0 | 0.0±0.0 | 0.0±0.0 | 0.0±0.0 | 0.0±0.0 | 0.0±0.0 | 0.0±0.0 | 0.0±0.0 |
| *Ablation: Direct Torque Cont. - High Freq. (500Hz)* | ES (1M $\rho$ steps) | Direct Torque | 224±2.01 | 135±0.509 | 117±1.05 | 159±1.2 | 118±0.891 | 102±0.323 | 100±1.08e-06 | 344±0.985 | 73.1±0.509 | 104±0.17 | 189±3.02 |
| *Ablation: Direct Torque Cont. - Low Freq. (31.25Hz)* | ES (1M $\rho$ steps) | Direct Torque | 198±3.12 | 119±0.556 | 118±0.349 | 168±0.497 | 115±0.863 | 100±0.0525 | 100±1.08e-06 | 305±2.54 | 70.2±0.267 | 105±0.113 | 184±2.28 |
| *Ablation: Fixed Cont. - PD Position* | ES (EvoControl ll steps) | PD Position | 220±0.748 | 52.5±0.881 | 104±1.1 | 158±0.395 | 119±0.158 | 101±0.0186 | 100±1.08e-06 | 390±1.33 | 69.1±0.37 | 86.7±0.179 | 152±1.38 |
| *Ablation: Fixed Cont. - PD Position Delta* | ES (EvoControl neuro. steps) | PD Position Delta | 36.2±0.172 | 4.48±0.00862 | 133±0.535 | 143±0.337 | 4.6±0.00358 | 97.7±0.782 | 100±0.0594 | 279±3.12 | 52±0.174 | 12.2±0.83 | 122±0.452 |
| *Ablation: Fixed Cont. - PD Int. Velocity* | ES (EvoControl neuro. steps) | PD Int. Velocity | 82.7±1.97 | 67.6±0.85 | 96±0.941 | 172±0.689 | 118±0.124 | 97.9±1.22 | 55.2±1.81 | 256±1.19 | 61.2±0.179 | 59.7±0.869 | 140±2.56 |
| *Ablation: Fixed Cont. - Random* | Random | Direct Torque | 0.0±0.0 | 0.0±0.0 | 0.0±0.0 | 0.0±0.0 | 0.0±0.0 | 0.0±0.0 | 0.0±0.0 | 0.0±0.0 | 0.0±0.0 | 0.0±0.0 | 0.0±0.0 |
| *Ablation: Direct Torque Cont. - High Freq. (500Hz)* | ES (EvoControl neuro. steps) | Direct Torque | 288±3.08 | 135±0.509 | 117±1.05 | 156±1.02 | 116±0.851 | 102±0.323 | 100±1.08e-06 | 344±0.985 | 73.1±0.509 | 104±0.17 | 169±2.98 |
| *Ablation: Direct Torque Cont. - Low Freq. (31.25Hz)* | ES (EvoControl neuro. steps) | Direct Torque | 223±1.56 | 119±0.556 | 118±0.349 | 158±0.574 | 119±0.314 | 100±0.0525 | 100±1.08e-06 | 305±2.54 | 70.2±0.267 | 105±0.113 | 180±2.17 |

Second, within EvoControl we perform the ablation of training the high-level policy with ES instead of PPO, using the same high-level policy architecture as used in the main paper. Again, we provide two versions, A) tha of training the A) that of training the high-level policy with the same number of high-level $\rho$ steps as the core baselines presented in the paper, with 1M $\rho$ steps—labelled *ES (1M $\rho$ steps)*. Specifically, this uses $es\_rollouts = 1, K = 8, es\_pop\_size = 256, es\_sub\_generations = 8$. Second, B) training the high-level policy with the same number of environment steps as was used to train the low-level policy within EvoControl—labelled *ES (same as EvoControl low-level)*. Specifically this uses more low-level environment steps than A), with $es\_rollouts = 16, K = 8, es\_pop\_size = 512, es\_sub\_generations = 8$. The results are tabulated alongside the original main table of results for ease of comparison in Table 31. Crucially we observe that on average EvoControl using PPO to train the high-level outperforms (on average achieves a higher normalized evaluation return $\mathcal{R}$) training the high-level with ES, confirming the main framework and EvoControl method presented in the paper, and the advantages of the unique combination of a high-level PPO learned policy with a ES-learned low-level policy. Furthermore, we also observe that on average these variations of EvoControl outperform the respective non-EvoControl baselines of *fixed-controllers* and *direct torque control*.

Table 31: Ablation. Training the high-level policy with ES instead of PPO for EvoControl. Normalized evaluation return $\mathcal{R}$ for the benchmark methods, across each environment. EvoControl on average achieves a higher normalized evaluation return than the baselines of *fixed controllers* and *direct torque control*. Results are averaged over 384 random seeds, with ± indicating 95% confidence intervals. Returns are normalized to a 0-100 scale, where 0 represents a random policy, and 100 represents the highest reward achieved by a non-EvoControl baseline in each environment. Scores bolded are greater than 100.

| Method Name | High-level $\rho$ with | Low-level $\beta$ with | Ant $\mathcal{R}\uparrow$ | Halfcheetah $\mathcal{R}\uparrow$ | Hopper $\mathcal{R}\uparrow$ | Humanoid $\mathcal{R}\uparrow$ | Humanoid Standup $\mathcal{R}\uparrow$ | Inverted Double Pendulum $\mathcal{R}\uparrow$ | Inverted Pendulum $\mathcal{R}\uparrow$ | Pusher $\mathcal{R}\uparrow$ | Reacher $\mathcal{R}\uparrow$ | Reacher 1D $\mathcal{R}\uparrow$ | Walker2D $\mathcal{R}\uparrow$ |
|---|---|---|---|---|---|---|---|---|---|---|---|---|---|
| *Fixed Cont.* - PD Position | PPO | PD Position | 100±6.25 | 63.2±0.599 | 89.3±1.2 | 92.6±2.05 | 100±0.208 | 99.9±0.03 | 100±1.53e-06 | 100±8.47 | 100±1.77 | 85±2.93 | 81.7±0.384 |
| *Fixed Cont.* - PD Position Delta | PPO | PD Position Delta | 2.57±1.74 | 2.86±0.092 | 100±1.84 | 100±1.78 | 2.54±0.0542 | 53.8±1.57 | 100±1.53e-06 | 0±0 | 40.8±3.23 | 15.2±7.6 | 96.3±0.219 |
| *Fixed Cont.* - PD Int. Velocity | PPO | PD Int. Velocity | 2.46±1.51 | 2.55±0.0967 | 73.1±0.844 | 86.6±1.33 | 0±0 | 49.7±1.55 | 86.5±2 | 0±0 | 0±0 | 0±0 | 93.7±2.77 |
| *Fixed Cont.* - Random | Random | Direct Torque | 0.0±0.0 | 0.0±0.0 | 0.0±0.0 | 0.0±0.0 | 0.0±0.0 | 0.0±0.0 | 0.0±0.0 | 0.0±0.0 | 0.0±0.0 | 0.0±0.0 | 0.0±0.0 |
| *Direct Torque Cont.* - High Freq. (500Hz) | PPO | Direct Torque | 0±0 | 18±0.353 | 0.882±0.533 | 7.93±1.94 | 10.1±0.357 | 0±0 | 0±0 | 4.36±7.75 | 0±0 | 45.3±6.74 | 0±0 |
| *Direct Torque Cont.* - Low Freq. (31.25Hz) | PPO | Direct Torque | 63±7.07 | 100±1.54 | 70.6±0.548 | 85.6±1.41 | 80±1.73 | 100±0.0311 | 100±1.53e-06 | 75.4±12.8 | 58.8±3.74 | 100±1.94 | 100±1.06 |
| **EvoControl (Full State)** | PPO | ES | **368±73.2** | **157±18.3** | **274±20.6** | **123±19** | **116±18.4** | **101±0.859** | **100±0** | **362±12.5** | **114±7.51** | **106±2.75** | **203±137** |
| **EvoControl (Residual State)** | PPO | ES | **182±16.1** | **182±6.31** | **101±5.54** | **170±18.2** | **212±145** | 99.2±1.25 | **100±0** | **375±94.8** | **106±25.8** | **104±3.42** | **205±57.4** |
| **EvoControl (Target + Proprio.)** | PPO | ES | **319±35.1** | **168±14.7** | **171±155** | **165±7.19** | **165±150** | 99.7±1.19 | **100±0** | **353±28.4** | 96.8±78.6 | **105±4.71** | **178±63.7** |
| **EvoControl (Target)** | PPO | ES | **293±87** | **162±23.5** | **283±250** | **164±21.2** | **205±147** | 99.6±0.949 | **100±0** | **353±43.4** | **112±1.73** | **105±1.65** | **188±88.2** |
| **EvoControl (Learned Gains)** | PPO | ES | **266±104** | **113±10.5** | **206±302** | **150±15.1** | **117±2.44** | 99.5±2.48 | **100±0** | **330±17.8** | **116±1.62** | **105±2.45** | **196±118** |
| **EvoControl (Delta Position)** | PPO | ES | **362±47.7** | **133±34.6** | **225±82.9** | **119±18** | **105±4.64** | **101±0.394** | **100±0** | **267±30.2** | 65.5±21 | 99.1±12.7 | **183±34.4** |
| **Ablation: EvoControl (Full State)** | ES (1M $\rho$ steps) | ES | **151±42.9** | **134±6.93** | 99.6±26.5 | **115±33.3** | **110±18.1** | 99.6±1.66 | **100±0** | **320±84.2** | 55.4±16.4 | 42.2±56.3 | **144±72.4** |
| **Ablation: EvoControl (Residual State)** | ES (1M $\rho$ steps) | ES | **136±135** | 99.4±21.3 | 90.9±2.96 | **138±13** | **115±3.68** | 67.7±38.9 | 17.5±66.1 | **292±22.3** | 52.3±11.5 | 2.98±2.12 | **117±34.9** |
| **Ablation: EvoControl (Target + Proprio.)** | ES (1M $\rho$ steps) | ES | **130±13.9** | **151±32.5** | 80.4±47.6 | **121±20.7** | **107±22.8** | 44±52.2 | 23±48.6 | **296±74.9** | 53.9±12.4 | 7.38±17.6 | **126±24.4** |
| **Ablation: EvoControl (Target)** | ES (1M $\rho$ steps) | ES | **135±7.57** | **101±19** | 75.6±222 | **123±14.3** | **103±22** | 65.4±40.5 | 51.1±42.4 | **342±46.1** | 52.3±9.81 | 26±40.1 | **130±42.2** |
| **Ablation: EvoControl (Learned Gains)** | ES (1M $\rho$ steps) | ES | **174±38.9** | 89±60.3 | 67.2±160 | **111±23.9** | **113±2.31** | 46.7±23.3 | 76.4±101 | **252±101** | 53.8±8.36 | 26.2±29.4 | **113±18.4** |
| **Ablation: EvoControl (Delta Position)** | ES (1M $\rho$ steps) | ES | **156±15.7** | **130±9.55** | **132±261** | **119±17.6** | **107±17.1** | 90.6±40.5 | **100±0** | **322±12.6** | 54.9±11.8 | 31.9±72.6 | **112±98.9** |
| **Ablation: EvoControl (Full State)** | ES (EvoControl ll steps) | ES | **176±6.59** | **143±12.4** | 94.4±15.6 | **123±24.3** | **108±13.4** | 95.8±9.58 | **100±0** | **334±52.9** | 55.4±16.4 | 15.7±28.3 | **153±43.5** |
| **Ablation: EvoControl (Residual State)** | ES (EvoControl ll steps) | ES | **169±51.8** | **154±41.9** | 91.1±8.64 | **131±44.1** | **142±58** | 42.9±69.4 | 85.8±59.4 | **300±50.1** | 52.3±10.1 | 4.56±8.08 | **142±21.8** |
| **Ablation: EvoControl (Target + Proprio.)** | ES (EvoControl ll steps) | ES | **153±13** | **137±19.3** | 94.6±0.0 | **122±26.4** | **102±23.2** | 62.7±18.8 | 43.8±65.7 | **291±16.4** | 53.9±12.3 | 27.3±52.4 | **122±26** |
| **Ablation: EvoControl (Target)** | ES (EvoControl ll steps) | ES | **152±61** | **131±109** | 63.5±77.1 | **120±22.9** | 99.6±25.1 | 48±24.7 | 20.4±15 | **313±74.8** | 54.1±14.7 | 18.8±25.4 | **116±11.3** |
| **Ablation: EvoControl (Learned Gains)** | ES (EvoControl ll steps) | ES | **145±166** | 90.5±35.2 | 13.9±80.3 | **126±12.6** | **107±21.8** | 71.8±18.3 | 97±12.8 | **237±99.7** | 53.8±13.9 | 45.9±19.8 | 76.1±155 |
| **Ablation: EvoControl (Delta Position)** | ES (EvoControl ll steps) | ES | **168±60** | **140±13.5** | **147±27** | **118±8.28** | **107±8.46** | 95.8±16.6 | **100±0** | **315±103** | 55.6±13.2 | 50±83.2 | **172±67.5** |

## J.10. Rollout Trajectory Plots of High-level Action for Baselines

In the following we plot rollout trajectories, including the high-level latent action $a_k$ over one evaluation episode in the Reacher 1D environment for all baselines. We take the trained policies, as trained in the main table of results in the paper (Table 3), and evaluate them for one random seed to produce the rollout trajectory plot. To facilitate simpler comparison, we use the same random seed across the baseline trajectory plots. Moreover, we use the Reacher 1D environment as it is straightforward to plot and understand what an optimal policy $\pi$ should do. In this case the initial reacher arm starts in a random position (as defined on a circle $q_0 \in (-\pi, \pi)$), with a random velocity, and its goal is to move the arm to a randomly sampled goal location (goal $q_{\text{goal}} \in (-\pi, \pi)$)—in the trajectory plots we plot the goal location with the red line, which is constant throughout the episode. Therefore an optimal policy is one that moves the arm to the goal location, quickly and keeps it there to maximize reward, where the reward is defined as $-||\text{xy\_tip\_of\_arm}(q_0) - \text{xy\_tip\_of\_arm}(q_{\text{goal}})||_2^2$. We provide the plots for each respective baseline in Figures 15 to 25. We observe that the EvoControl variants on average consistently outperform the other non-EvoControl baselines and achieve a higher evaluation return $R$—which is also given on each plot.

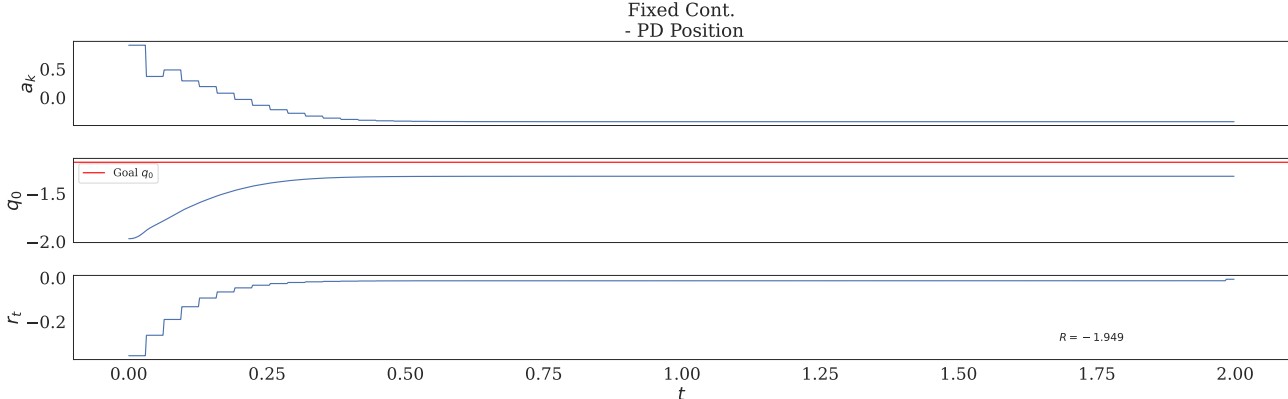

Figure 15: Evaluation Trajectory Rollout for Reacher 1D, for baseline *Fixed Cont.* - PD Position. Environment runs at 500Hz, with an 1,000 low-level environment steps, corresponding to a episode duration of 2.0 seconds. Where $a_k$ is the high-level policy $\rho$ latent action, $q_0$ is the reacher arm's angle in radians, with the red-line indicating the random goal $q_{\text{goal}}$ for the episode, $r_t$ the instantaneous reward and $R$ the total return for the episode.

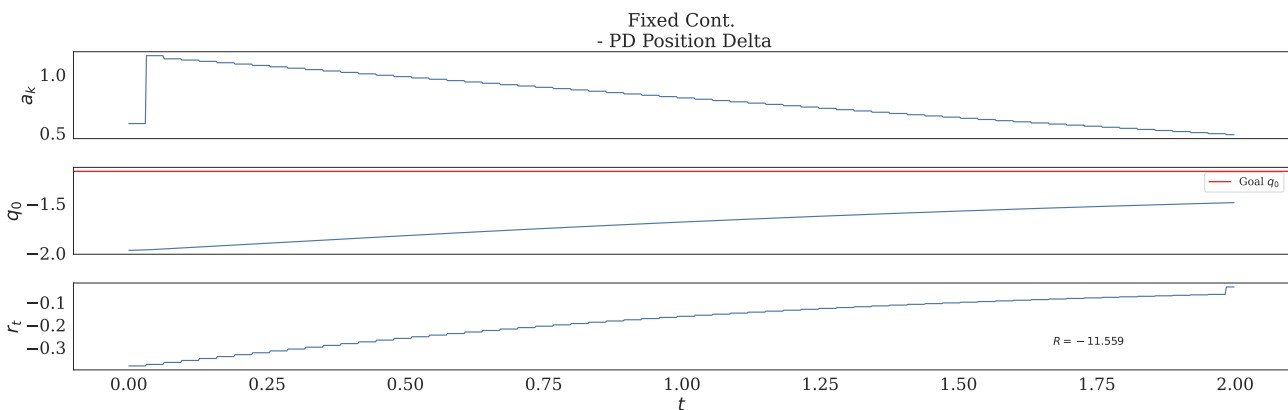

Figure 16: Evaluation Trajectory Rollout for Reacher 1D, for baseline *Fixed Cont.* - PD Position Delta. Environment runs at 500Hz, with an 1,000 low-level environment steps, corresponding to a episode duration of 2.0 seconds. Where $a_k$ is the high-level policy $\rho$ latent action, $q_0$ is the reacher arm's angle in radians, with the red-line indicating the random goal $q_{\text{goal}}$ for the episode, $r_t$ the instantaneous reward and $R$ the total return for the episode.

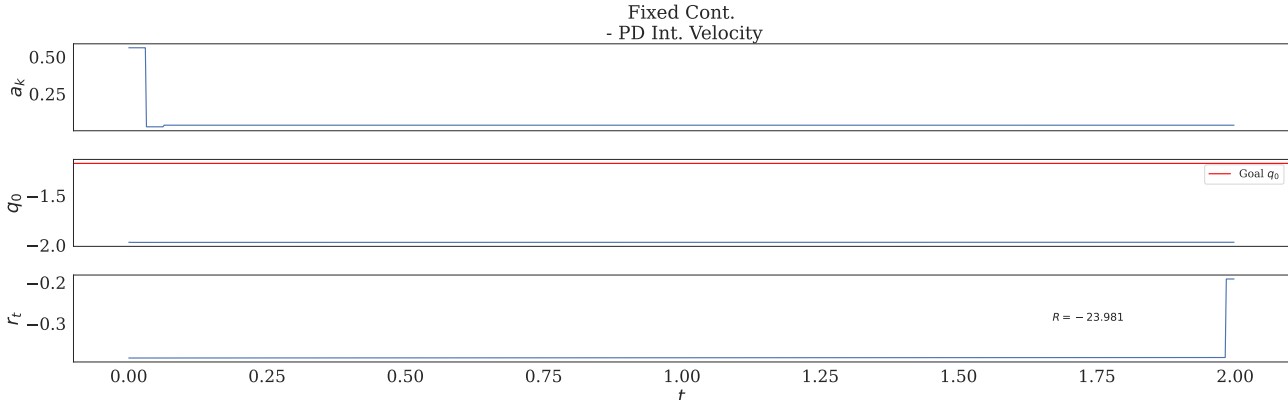

Figure 17: Evaluation Trajectory Rollout for Reacher 1D, for baseline *Fixed Cont.* - PD Int. Velocity. Environment runs at 500Hz, with an 1,000 low-level environment steps, corresponding to a episode duration of 2.0 seconds. Where $a_k$ is the high-level policy $\rho$ latent action, $q_0$ is the reacher arm's angle in radians, with the red-line indicating the random goal $q_{\text{goal}}$ for the episode, $r_t$ the instantaneous reward and $R$ the total return for the episode.

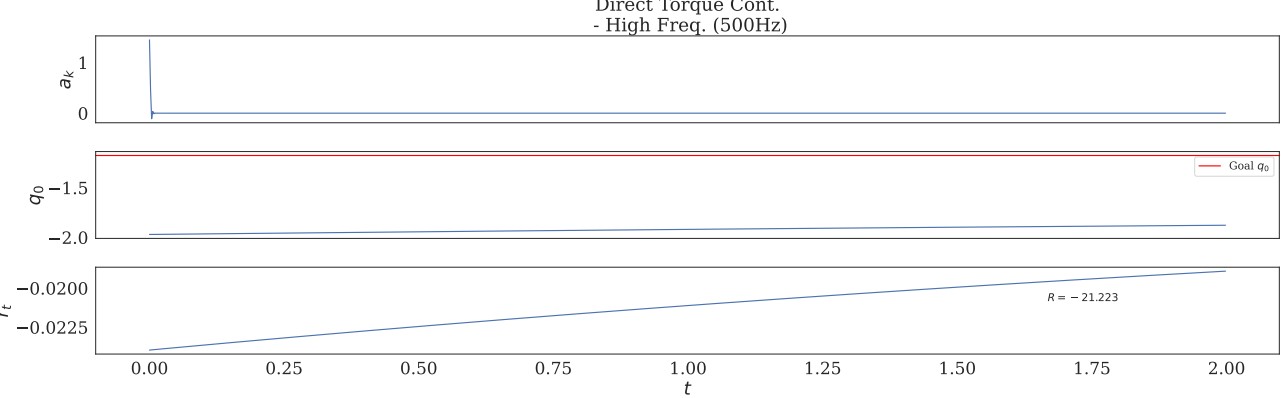

Figure 18: Evaluation Trajectory Rollout for Reacher 1D, for baseline *Direct Torque Cont.* - High Freq. (500Hz). Environment runs at 500Hz, with an 1,000 low-level environment steps, corresponding to a episode duration of 2.0 seconds. Where $a_k$ is the high-level policy $\rho$ latent action, $q_0$ is the reacher arm's angle in radians, with the red-line indicating the random goal $q_{\text{goal}}$ for the episode, $r_t$ the instantaneous reward and $R$ the total return for the episode.

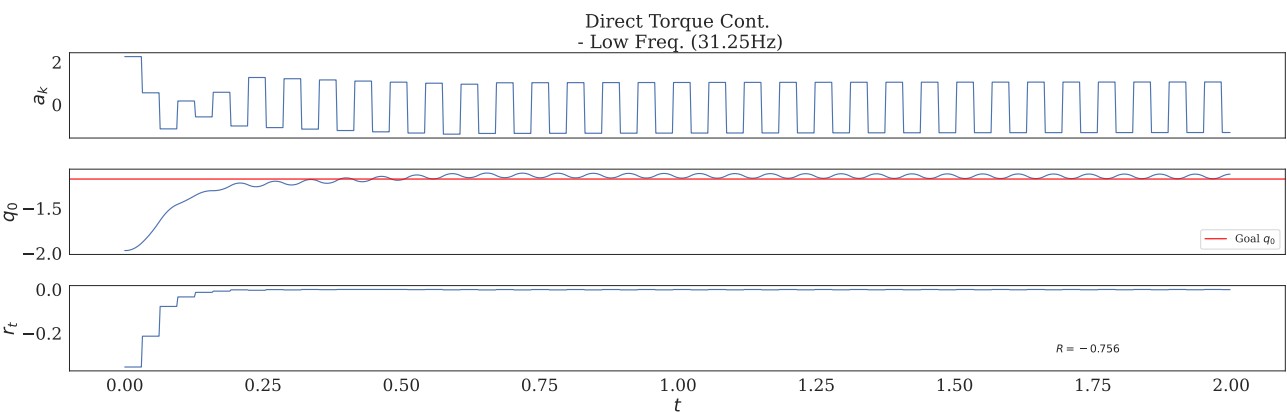

Figure 19: Evaluation Trajectory Rollout for Reacher 1D, for baseline *Direct Torque Cont. - Low Freq. (31.25Hz)*. Environment runs at 500Hz, with an 1,000 low-level environment steps, corresponding to a episode duration of 2.0 seconds. Where $a_k$ is the high-level policy $\rho$ latent action, $q_0$ is the reacher arm's angle in radians, with the red-line indicating the random goal $q_{\text{goal}}$ for the episode, $r_t$ the instantaneous reward and $R$ the total return for the episode.

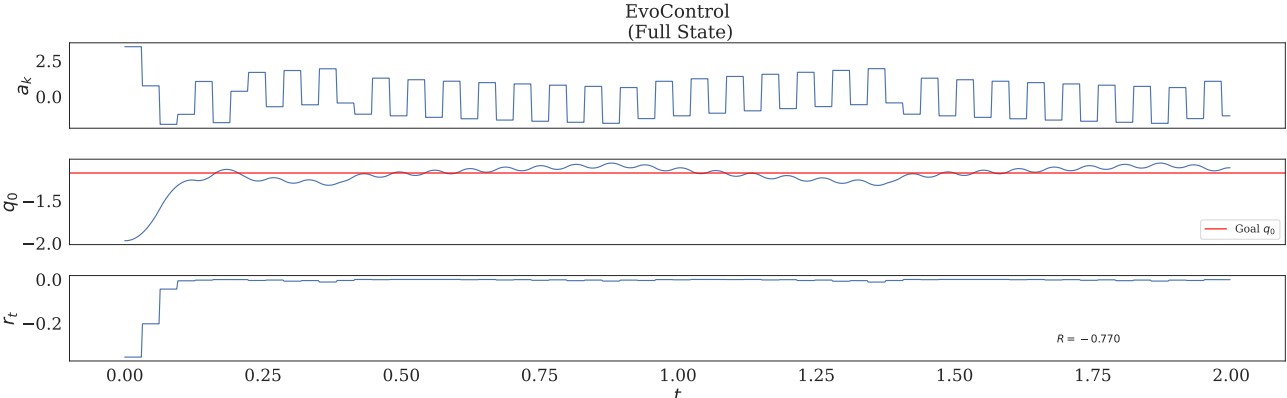

Figure 20: Evaluation Trajectory Rollout for Reacher 1D, for baseline *EvoControl* (Full State). Environment runs at 500Hz, with an 1,000 low-level environment steps, corresponding to a episode duration of 2.0 seconds. Where $a_k$ is the high-level policy $\rho$ latent action, $q_0$ is the reacher arm's angle in radians, with the red-line indicating the random goal $q_{\text{goal}}$ for the episode, $r_t$ the instantaneous reward and $R$ the total return for the episode.

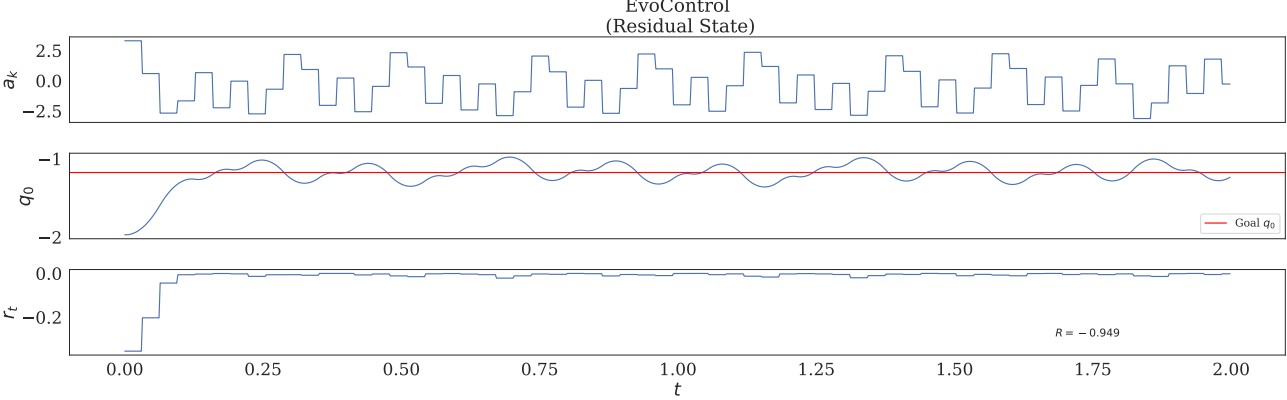

Figure 21: Evaluation Trajectory Rollout for Reacher 1D, for baseline *EvoControl* (Residual State). Environment runs at 500Hz, with an 1,000 low-level environment steps, corresponding to a episode duration of 2.0 seconds. Where $a_k$ is the high-level policy $\rho$ latent action, $q_0$ is the reacher arm's angle in radians, with the red-line indicating the random goal $q_{\text{goal}}$ for the episode, $r_t$ the instantaneous reward and $R$ the total return for the episode.

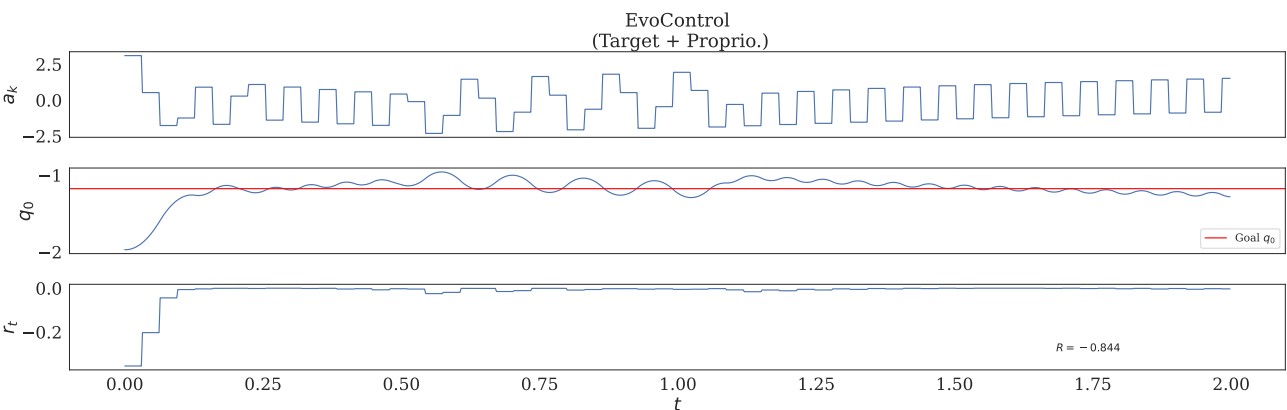

Figure 22: Evaluation Trajectory Rollout for Reacher 1D, for baseline *EvoControl* (Target + Proprio.). Environment runs at 500Hz, with an 1,000 low-level environment steps, corresponding to a episode duration of 2.0 seconds. Where $a_k$ is the high-level policy $\rho$ latent action, $q_0$ is the reacher arm's angle in radians, with the red-line indicating the random goal $q_{\text{goal}}$ for the episode, $r_t$ the instantaneous reward and $R$ the total return for the episode.

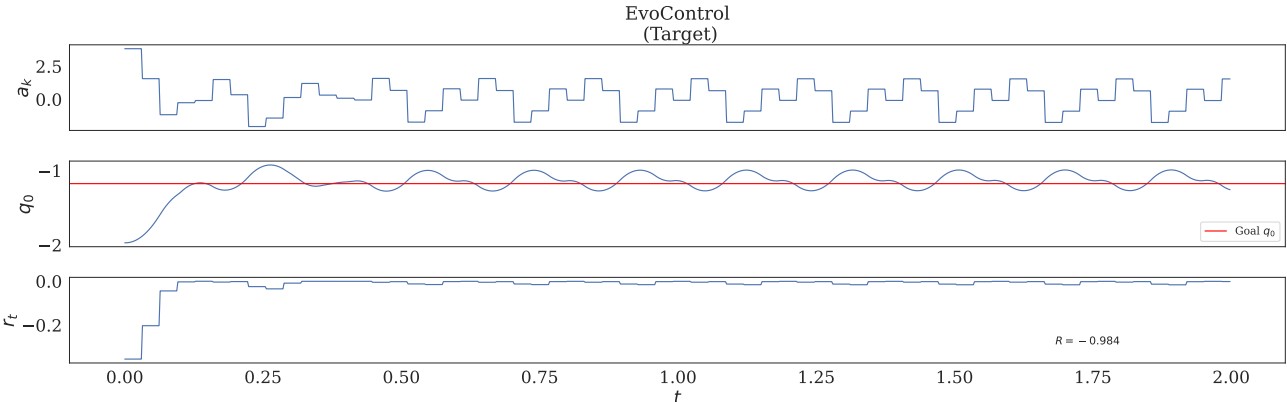

Figure 23: Evaluation Trajectory Rollout for Reacher 1D, for baseline *EvoControl* (Target). Environment runs at 500Hz, with an 1,000 low-level environment steps, corresponding to a episode duration of 2.0 seconds. Where $a_k$ is the high-level policy $\rho$ latent action, $q_0$ is the reacher arm's angle in radians, with the red-line indicating the random goal $q_{\text{goal}}$ for the episode, $r_t$ the instantaneous reward and $R$ the total return for the episode.

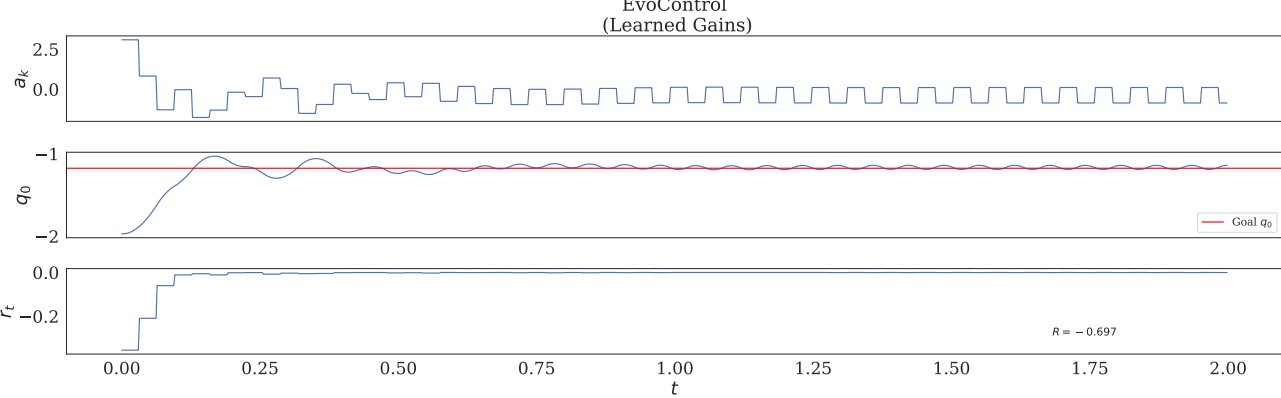

Figure 24: Evaluation Trajectory Rollout for Reacher 1D, for baseline *EvoControl* (Learned Gains). Environment runs at 500Hz, with an 1,000 low-level environment steps, corresponding to a episode duration of 2.0 seconds. Where $a_k$ is the high-level policy $\rho$ latent action, $q_0$ is the reacher arm's angle in radians, with the red-line indicating the random goal $q_{\text{goal}}$ for the episode, $r_t$ the instantaneous reward and $R$ the total return for the episode.

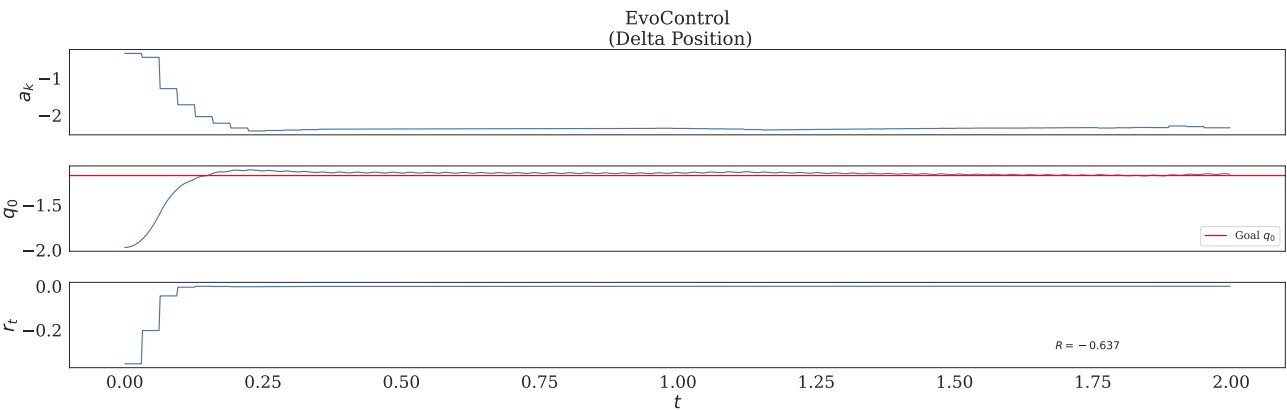

Figure 25: Evaluation Trajectory Rollout for Reacher 1D, for baseline *EvoControl* (Delta Position). Environment runs at 500Hz, with an 1,000 low-level environment steps, corresponding to a episode duration of 2.0 seconds. Where $a_k$ is the high-level policy $\rho$ latent action, $q_0$ is the reacher arm's angle in radians, with the red-line indicating the random goal $q_{\text{goal}}$ for the episode, $r_t$ the instantaneous reward and $R$ the total return for the episode.

## J.11. Ablation: Removing Communication Between the Layers in EvoControl

We performed a further ablation, which tests the hypothesis of if within EvoControl, whether there is useful communication between the high-level and low-level policy. Specifically for the EvoControl variants that we considered, in some variations the low-level policy receives only a restricted observation (just the joint positions of the robot, and not the random goal location if one exists), compared to receiving the full observation (which includes any random goal location if one exists for that environment).

Specifically, we consider two variants of EvoControl, where the low-level policy receives the full observation (*EvoControl - (Full State)*), and where the low-level policy only receives a restricted observation without any goal location—necessitating effective communication from the high-level to the low-level (*EvoControl (Target)*). We compare these on the Reacher 1D task, visualizing the rollouts for a random high-level policy, and a zero high-level policy, as observed in Figures 26 and 27. We observe that for both variations after the standard EvoControl training, removing the communication (by making the high-level policy either a random policy or a null policy) significantly reduces the performance (return), and leads to an unstable low-level policy, even when the low-level policy receives the full observation ($s_t$). Intuitively, it could be the case that the low-level policy learns a form of a PD controller, and the high-level policy, as trained initially with a PD position controller, could converge to treat the low-level policy as a form of PD controller.

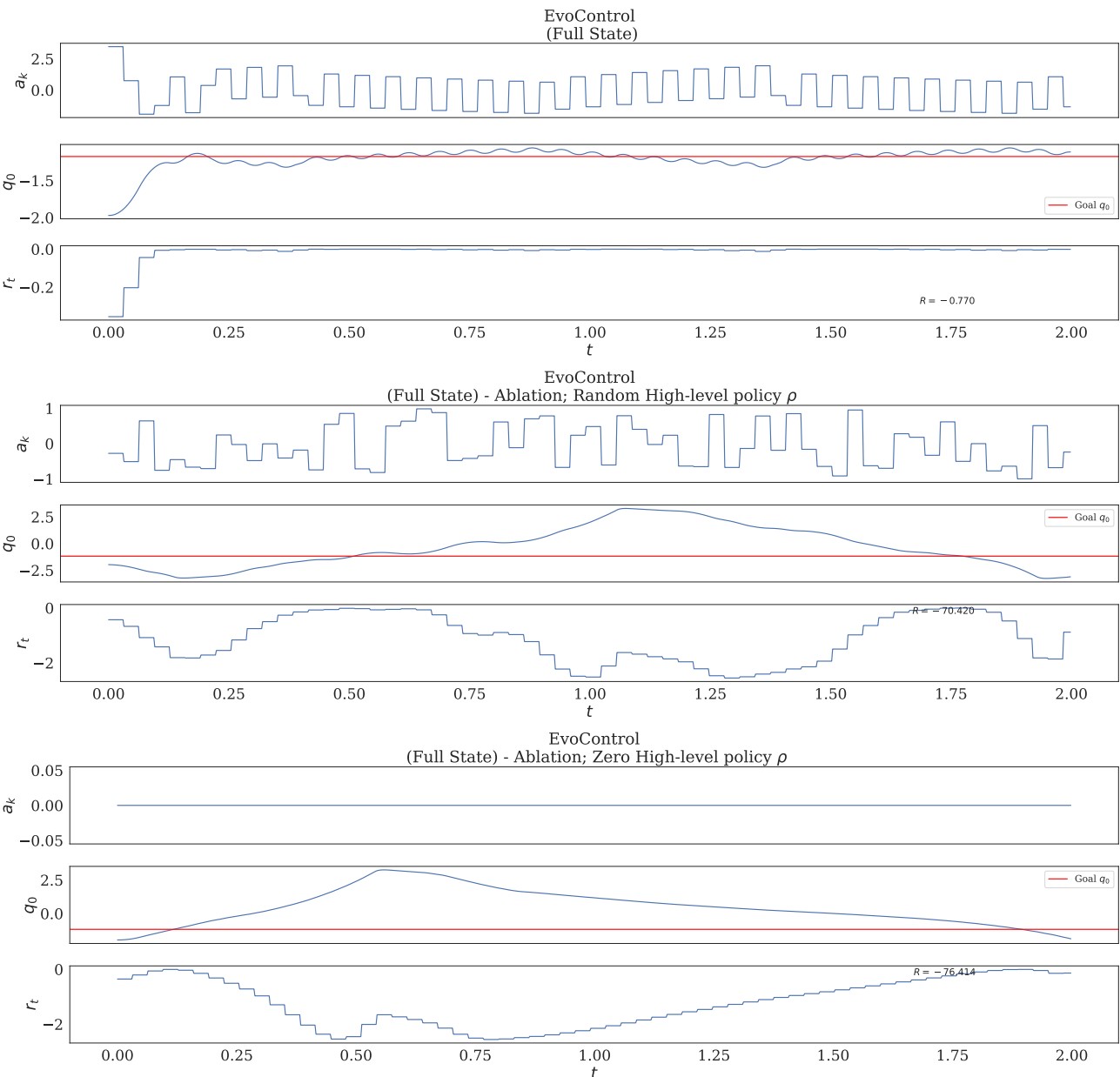

Figure 26: Evaluation Trajectory Rollout for Reacher 1D, for baseline *EvoControl* (Full State), with ablation of a random high-level and a null (zero action) policy. Here the observation for the low-level policy is $s_t, a_k, e_t, q_t, \dot{q}_t, t/T$. We observe that even with the low-level policy receiving the full observation it still relies on the communication from the high-level latent action $a_k$, and without it, the return significantly reduces. Environment runs at 500Hz, with an 1,000 low-level environment steps, corresponding to a episode duration of 2.0 seconds. Where $a_k$ is the high-level policy $\rho$ latent action, $q_0$ is the reacher arm's angle in radians, with the red-line indicating the random goal $q_{\text{goal}}$ for the episode, $r_t$ the instantaneous reward and $R$ the total return for the episode.

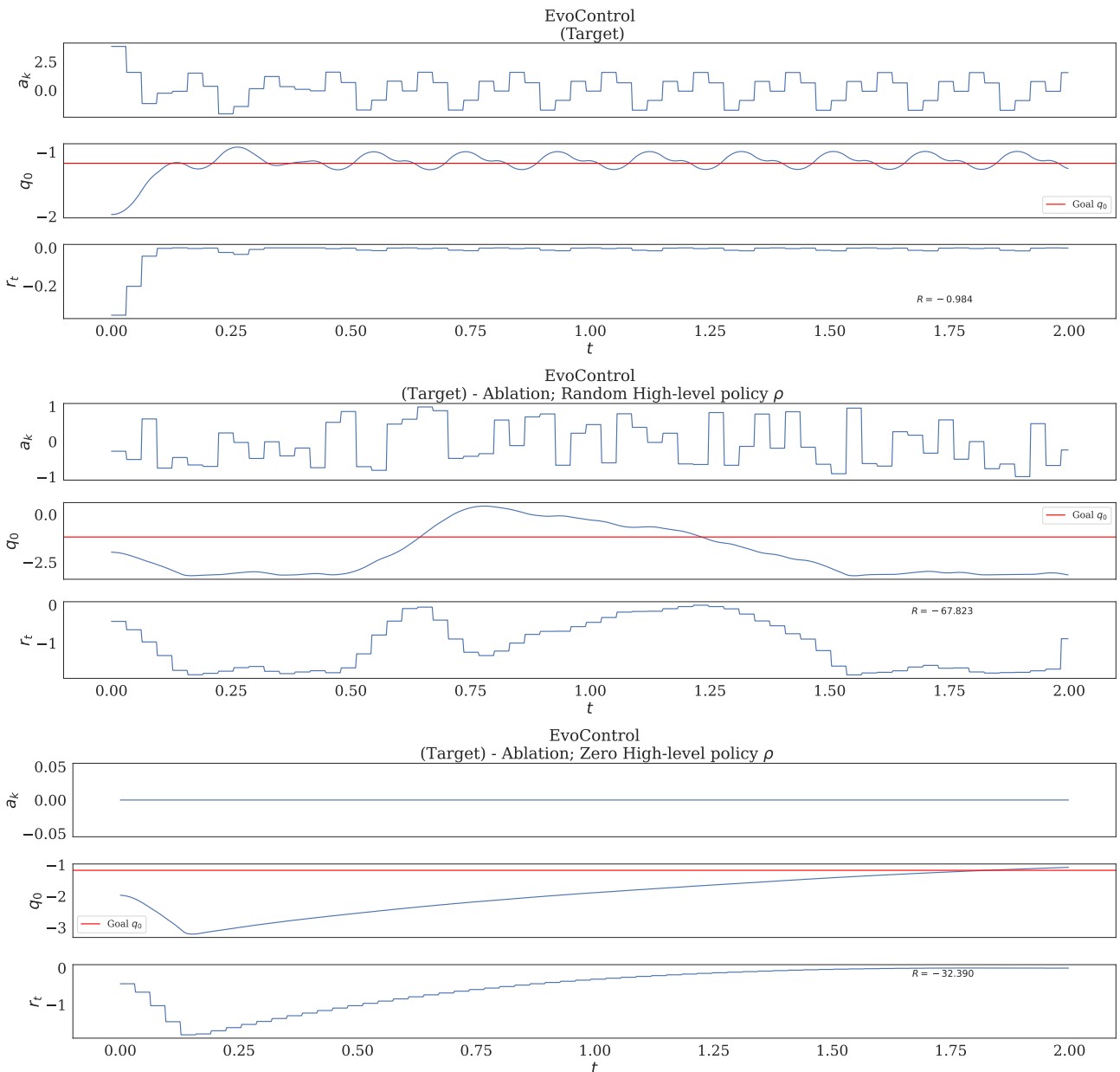

Figure 27: Evaluation Trajectory Rollout for Reacher 1D, for baseline *EvoControl* (Target), with ablation of a random high-level and a null (zero action) policy. Here the observation for the low-level policy is $a_k, q_t, \dot{q}_t, t/T$. We observe that the low-level policy relies on the communication from the high-level latent action $a_k$, and without it, the return significantly reduces. Environment runs at 500Hz, with an 1,000 low-level environment steps, corresponding to a episode duration of 2.0 seconds. Where $a_k$ is the high-level policy $\rho$ latent action, $q_0$ is the reacher arm's angle in radians, with the red-line indicating the random goal $q_{\text{goal}}$ for the episode, $r_t$ the instantaneous reward and $R$ the total return for the episode.

### J.12. Ablation on Varying Fixed Values of the Annealing Parameter $\alpha$

First, we clarify that in EvoControl, the annealing parameter $\alpha$ is gradually reduced from 1 to 0 during training. Instead, we analyze the impact of fixing $\alpha$ to a constant value throughout training.

By default, EvoControl anneals $\alpha$ over time, balancing control between the high-level and low-level controllers: $\alpha = 0$ means full control by the high-level policy, while $\alpha = 1$ means the low-level controller (initialized as a PD controller) has full control. We present this analysis in Table 32.

Table 32: Ablation of EvoControl with fixed $\alpha$ parameter. Normalized evaluation return $\mathcal{R}$ for the benchmark methods, across the Reacher 1D environment. EvoControl on average achieves a higher normalized evaluation return than the baselines of *fixed controllers* and *direct torque control*. Results are averaged over 384 random seeds, with $\pm$ indicating 95% confidence intervals. Returns are normalized to a 0-100 scale, where 0 represents a random policy, and 100 represents the highest reward achieved by a non-EvoControl baseline in each environment. Scores bolded are greater than 100.

| Same PPO high-level alg. $\rho$ with | Reacher 1D ($\alpha = 0.0$) $\mathcal{R} \uparrow$ | Reacher 1D ($\alpha = 0.1$) $\mathcal{R} \uparrow$ | Reacher 1D ($\alpha = 0.2$) $\mathcal{R} \uparrow$ | Reacher 1D ($\alpha = 0.4$) $\mathcal{R} \uparrow$ | Reacher 1D ($\alpha = 0.6$) $\mathcal{R} \uparrow$ | Reacher 1D ($\alpha = 0.8$) $\mathcal{R} \uparrow$ | Reacher 1D ($\alpha = 1.0$) $\mathcal{R} \uparrow$ |
|---|---|---|---|---|---|---|---|
| *Fixed Cont.* - PD Position | 85±2.93 | 85±2.93 | 85±2.93 | 85±2.93 | 85±2.93 | 85±2.93 | 85±2.93 |
| *Fixed Cont.* - PD Position Delta | 15.2±7.6 | 15.2±7.6 | 15.2±7.6 | 15.2±7.6 | 15.2±7.6 | 15.2±7.6 | 15.2±7.6 |
| *Fixed Cont.* - PD Int. Velocity | 0±0 | 0±0 | 0±0 | 0±0 | 0±0 | 0±0 | 0±0 |
| *Fixed Cont.* - Random | 0.0±0.0 | 0.0±0.0 | 0.0±0.0 | 0.0±0.0 | 0.0±0.0 | 0.0±0.0 | 0.0±0.0 |
| *Direct Torque Cont.* - High Freq. (500Hz) | 45.3±6.74 | 45.3±6.74 | 45.3±6.74 | 45.3±6.74 | 45.3±6.74 | 45.3±6.74 | 45.3±6.74 |
| *Direct Torque Cont.* - Low Freq. (31.25Hz) | 100±1.94 | 100±1.94 | 100±1.94 | 100±1.94 | 100±1.94 | 100±1.94 | 100±1.94 |
| **EvoControl (Full State)** | **105±1.25** | **106±0.781** | **106±0.765** | **105±1.31** | **105±0.822** | **105±1** | 89.7±2.79 |
| **EvoControl (Residual State)** | **105±1.03** | **105±0.916** | **105±0.893** | **105±0.828** | **104±1.04** | **106±0.754** | 89.7±2.79 |
| **EvoControl (Target + Proprio.)** | **105±0.947** | **106±0.693** | **105±1.17** | **106±1.6** | **106±0.844** | **105±0.733** | 89.7±2.79 |
| **EvoControl (Target)** | **104±0.943** | **105±1.02** | **105±1.02** | **104±1.29** | **104±1.62** | **105±0.935** | 89.7±2.79 |
| **EvoControl (Learned Gains)** | **100±1.66** | **105±1.68** | **105±0.869** | **102±1.19** | 94.1±2.69 | 91.7±2.8 | 89.7±2.79 |
| **EvoControl (Delta Position)** | **101±2.5** | **103±1.28** | **102±1.53** | **103±1.83** | **100±2.84** | 98.8±2.41 | 24.1±7.45 |

EvoControl consistently outperforms baselines across most $\alpha$ values, demonstrating the effectiveness of our hierarchical approach. Performance degrades for some EvoControl variants at $\alpha = 1.0$, indicating the importance of high-level policy guidance. This aligns with our annealing strategy, gradually reducing the low-level controller's influence as the high-level policy learns. The low-frequency torque control is competitive, but EvoControl achieves higher rewards, especially for lower $\alpha$ values, showcasing the benefits of learning the low-level control.

### J.13. PD Position Delta Action Scaling Sweep

We performed an additional experimental action scaling sweep of the baseline Fixed Controller - PD Position Delta. Specifically this baseline has a hyper-parameter of the action delta scaling parameter $\delta_c$. We swept over $\delta_c$ for the Reacher 1D environment and present the results in Table 33. We observe that a larger $\delta_c$ does improve the baseline's performance, however it still underperforms compared to EvoControl.

Table 33: Fixed Controller - PD Position Delta, action detla scaling parameter $\delta_c$ sweep. Normalized evaluation return $\mathcal{R}$ for the baselines across the Reacher 1D environment. We use the same normalization as the main table of results in the paper, that of Table 3.

| Same PPO high-level alg. $\rho$ with | Reacher 1D ($\delta_c = 0.05$) $\mathcal{R} \uparrow$ | Reacher 1D ($\delta_c = 0.1$) $\mathcal{R} \uparrow$ | Reacher 1D ($\delta_c = 0.2$) $\mathcal{R} \uparrow$ | Reacher 1D ($\delta_c = 0.4$) $\mathcal{R} \uparrow$ | Reacher 1D ($\delta_c = 0.6$) $\mathcal{R} \uparrow$ | Reacher 1D ($\delta_c = 0.8$) $\mathcal{R} \uparrow$ | Reacher 1D ($\delta_c = 1.0$) $\mathcal{R} \uparrow$ | Reacher 1D ($\delta_c = 2.0$) $\mathcal{R} \uparrow$ |
|---|---|---|---|---|---|---|---|---|
| *Fixed Cont.* - PD Position Delta | 15.2±7.6 | 33.6±6.97 | 57.3±5.59 | 76±4.12 | 81.8±3.64 | 83.2±3.54 | 83.7±3.43 | 84.5±3.28 |

### J.14. Safety Critical HalfCheetah

High-frequency control is especially valuable in safety-critical locomotion, where an agent must react instantaneously to unforeseen contacts. In the *Safety-Critical HalfCheetah* task (Appendix E.4) we insert a blocking capsule in 25% of episodes and expose only the measured contact force to the policy—the observation vector is of dimensionality 19. A successful controller must therefore (i) accelerate forward to maximise progress, yet (ii) immediately reverse direction upon detecting a collision in order to avoid the large penalty of $-200$ (c.f. Eq. equation 3). Such behaviour demands rapid, reflex-like adjustments that lower-frequency policies struggle to produce, thereby providing an empirical test-bed for Proposition 2.1. This enviornment is illustrated in Figure 28.

Table 34 reports the normalised evaluation return $\mathcal{R}$ ($\uparrow$)—scaled to $[0, 100]$ where 0 is a random policy and 100 is the best non-EvoControl baseline—averaged over 384 random seeds with 95% confidence intervals. EvoControl variants consistently outperform both *fixed* low-level controllers and *direct torque* control, even when the latter operates at 500 Hz. Notably, **EvoControl (Full State)** attains $120 \pm 13.9$, exceeding the best baseline by more than 20% despite using the same high-level PPO algorithm. These results corroborate our hypothesis that the ability to adapt low-level actions online at high frequency is crucial for safe and efficient locomotion in un-modelled environments.

Table 34: Normalised evaluation return $\mathcal{R}$ on the *Safety-Critical HalfCheetah*. EvoControl achieves significantly higher returns than both fixed high-frequency tracking controllers and direct torque control. Bold scores exceed the best baseline (100).

| Same PPO high-level algorithm $\rho$ with | Safety-Critical HalfCheetah $\mathcal{R}$ ($\uparrow$) |
|---|:---:|
| *Fixed Controller* – PD Position | $84.7 \pm 1.68$ |
| *Fixed Controller* – PD Position Delta | $0.0 \pm 0.0$ |
| *Fixed Controller* – Random | $0.0 \pm 0.0$ |
| *Direct Torque Control* – High Freq. (500 Hz) | $100 \pm 1.42$ |
| *Direct Torque Control* – Low Freq. (31.25 Hz) | $0.0 \pm 0.0$ |
| **EvoControl (Full State)** | $\mathbf{120 \pm 13.9}$ |
| **EvoControl (Residual State)** | $\mathbf{105 \pm 5.84}$ |
| **EvoControl (Target + Proprio.)** | $\mathbf{117 \pm 34.9}$ |
| **EvoControl (Target)** | $\mathbf{117 \pm 5.33}$ |

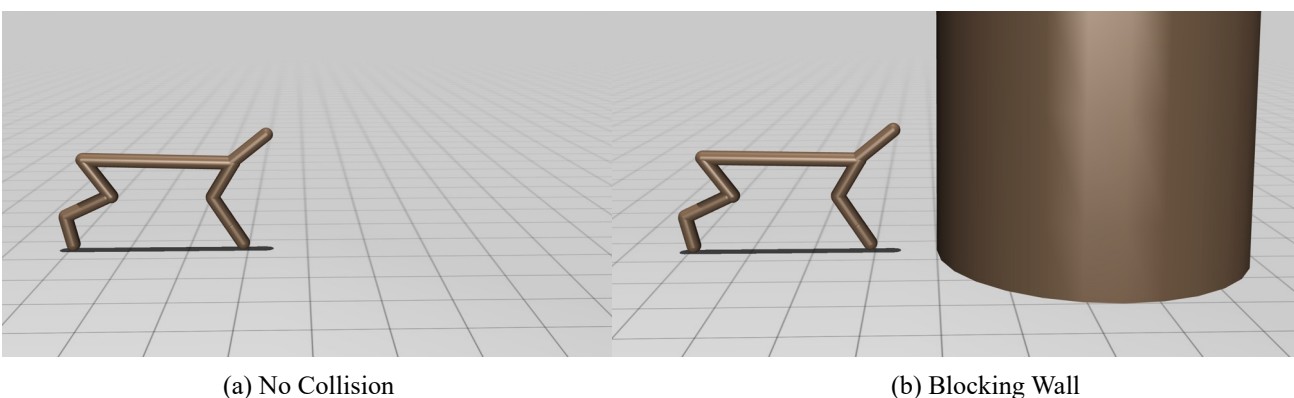

(a) No Collision              (b) Blocking Wall

Figure 28: **Safety Critical Halfcheetah Environment**—which is a halfcheetah enviornment (a), with a blocking wall that appears 25% of the episodes (b), and only contact force with wall can be observed, blocking the cheetahs goal path, and incurring a negative reward for collision with the wall. Therefore a good policy in this environment is to move forwards with the halfcheetah and if the wall collision is detected immediately retreat, otherwise continute forwards, therefore exemplifies the need for policies that take actions at a higher-frequency to achieve higher reward, for un-modelled safety critical applications.

### J.15. Comprehensive Baseline Comparison

To contextualise the gains of EVOCONTROL we benchmark it against a wide suite of strong controllers, including both *fixed* high–frequency tracking controllers and state-of-the-art model–free RL agents. We additionally train Soft Actor–Critic (SAC) and Proximal Policy Optimisation (PPO) with publicly available *Brax*[9] implementations at **high frequency** (500 Hz) and the **high-level low frequency** (31.25 Hz). Pre-trained policies from standard MuJoCo tasks cannot be used directly because our tasks require an order-of-magnitude faster actuation; instead, we retrain all methods from scratch under identical hyperparameters. We tabulate these results in Table 35.

**Normalisation and statistics.** Returns are *undiscounted* and linearly normalised to $[0, 100]$ such that $0$ corresponds to a random policy and $100$ to the *best* non-EVOCONTROL baseline in each environment. All numbers are means over $384$ random seeds; $\pm$ indicates the $95\%$ confidence interval.

Table 35: Normalised evaluation return $\mathcal{R}$ for all baselines and EVOCONTROL on the eight benchmark environments. Bold values exceed the best non-EVOCONTROL baseline ($\mathcal{R} > 100$).

| Same PPO high-level alg. $\rho$ with a Low-Level Policy $\beta$ of | Ant | Halfcheetah | Hopper | Inverted Double Pendulum | Inverted Pendulum | Reacher | Reacher 1D | Walker2D |
|---|---|---|---|---|---|---|---|---|
| | $\mathcal{R}\uparrow$ | $\mathcal{R}\uparrow$ | $\mathcal{R}\uparrow$ | $\mathcal{R}\uparrow$ | $\mathcal{R}\uparrow$ | $\mathcal{R}\uparrow$ | $\mathcal{R}\uparrow$ | $\mathcal{R}\uparrow$ |
| *Fixed Cont.* - PD Position | 100±6.56 | 61.2±0.441 | 88.1±1.18 | 99.9±0.03 | 100±2.86e-15 | 100±1.8 | 85.2±2.87 | 61.1±0.51 |
| *Fixed Cont.* - PD Position Delta | 2.4±1.91 | 2.76±0.0888 | 96.2±1.30 | 53.8±1.57 | 100±2.86e-15 | 40.9±3.23 | 15.2±7.6 | 72.7±0.19 |
| *Fixed Cont.* - PD Int. Velocity | 3.59±1.78 | 2.46±0.0932 | 71.8±0.87 | 49.7±1.55 | 86.5±2 | 0.0±0.0 | 0±0 | 69.3±2.06 |
| *Fixed Cont.* - PD : Position & $K_p$ | 3.55±2.54 | 16.7±0.151 | 100±1.00 | 97.5±0.751 | 100±2.86e-15 | 50.7±3.9 | 81.8±4.11 | 100±0.24 |
| *Fixed Cont.* - Random | 0.0±0.0 | 0.0±0.0 | 0.0±0.0 | 0.0±0.0 | 0.0±0.0 | 0.0±0.0 | 0.0±0.0 | 0.0±0.0 |
| *Direct Torque Cont.* - High Freq. (500Hz) | 0±0 | 17.2±0.316 | 1.37±0.51 | 10.3±0.586 | 0±0 | 0.97±5.72 | 45.3±6.74 | 0.0±0.0 |
| *Direct Torque Cont.* - Low Freq. (31.25Hz) | 54.5±7.15 | 100±1.21 | 69.2±0.62 | 80.6±2.56 | 100±0.0311 | 53.0±9.35 | 100±1.94 | 80.7±2.16 |
| **EvoControl (Full State)** | **368±10.6** | **157±1.1** | **263±1.46** | **101±0.0487** | 100±2.86e-15 | **114±0.973** | **106±0.936** | **163±3.72** |
| **EvoControl (Residual State)** | **182±8.58** | **182±1.02** | 97±0.51 | 99.2±0.054 | 100±2.86e-15 | **106±1.29** | **104±1.19** | **165±2.19** |
| **EvoControl (Target + Proprio.)** | **319±14.1** | **168±1.41** | **164±5.08** | 99.7±0.0417 | 100±2.86e-15 | 96.8±3.54 | **105±0.776** | **143±1.86** |
| **EvoControl (Target)** | **293±13.2** | **162±1.58** | **272±1.84** | 99.6±0.0377 | 100±2.86e-15 | **112±0.785** | **105±0.78** | **151±2.44** |
| **EvoControl (Learned Gains)** | **266±14.1** | **113±1.6** | **198±9.62** | 99.5±0.0947 | 100±2.86e-15 | **116±0.747** | **105±1.21** | **158±3.63** |
| **EvoControl (Delta Position)** | **362±12.8** | **133±1.82** | **216±2.88** | **101±0.0364** | 100±2.86e-15 | 65.5±3.71 | 99.1±2.44 | **147±1.90** |
| **SAC (Brax)** *Direct Torque Cont.* - High Freq. (500Hz) | 11.8±2.64 | 33.6±0.646 | **174±1.38** | 100±0 | 100±0 | 63.6±4.17 | **106±1.43** | **155±0.523** |
| **SAC (Brax)** *Direct Torque Cont.* - Low Freq. (31.25Hz) | 12.8±2.69 | 28.3±0.478 | 90.2±0.0343 | 100±0 | 100±0 | 63.9±4.09 | **105±2.08** | **156±0.41** |
| **PPO (Brax)** *Direct Torque Cont.* - High Freq. (500Hz) | 1.23±1.31 | 18.2±0.346 | 81±0.0225 | 100±0 | 100±0 | 62.1±4.13 | **104±1.54** | **114±2.67** |
| **PPO (Brax)** *Direct Torque Cont.* - Low Freq. (31.25Hz) | 3.14±1.1 | 18.2±0.346 | 81±0.0225 | 100±0 | 100±0 | 62.1±4.13 | **104±1.54** | **114±2.67** |

Across all eight environments EVOCONTROL attains the highest average normalised return, frequently $> 50\%$ above the strongest baseline. Crucially, even when sophisticated RL agents are granted the same 500 Hz control bandwidth, the population-based evolution of low-level reflexes in EVOCONTROL yields superior performance, underscoring the importance of closed-loop *online* adaptation at millisecond time-scales.

---

[9] https://github.com/google/brax

## K. Real-World Validation on a Franka Emika Panda

We validate *EvoControl* on a 7-DoF Franka Emika Panda robot equipped with a Robotiq 2F-85 gripper and a Robotiq FT-300 force-torque sensor, aiming to demonstrate two key aspects:

1. **Zero-shot sim-to-real transfer** on a tabletop manipulation task;

2. **Reduced collision forces** via high-frequency torque control, compared to a simpler position-based controller.

Figure 29 shows the overall setup in both MuJoCo simulation (left) and the physical robot (right).

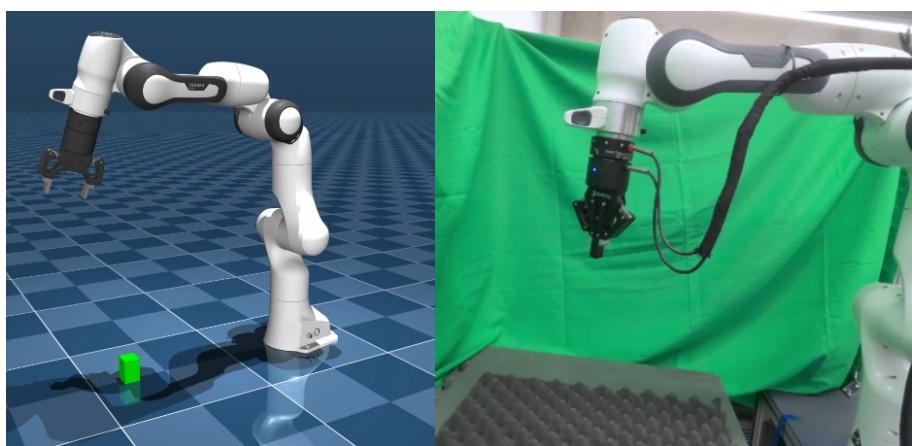

Figure 29: **Real-World Robot Setup.** *Left:* MuJoCo scene of our 7-DoF Franka Emika Panda robot equipped with a Robotiq 2F-85 gripper and a Robotiq FT-300 force-torque sensor. *Right:* The corresponding real hardware. We deploy two sim-to-real tasks to evaluate *EvoControl*.

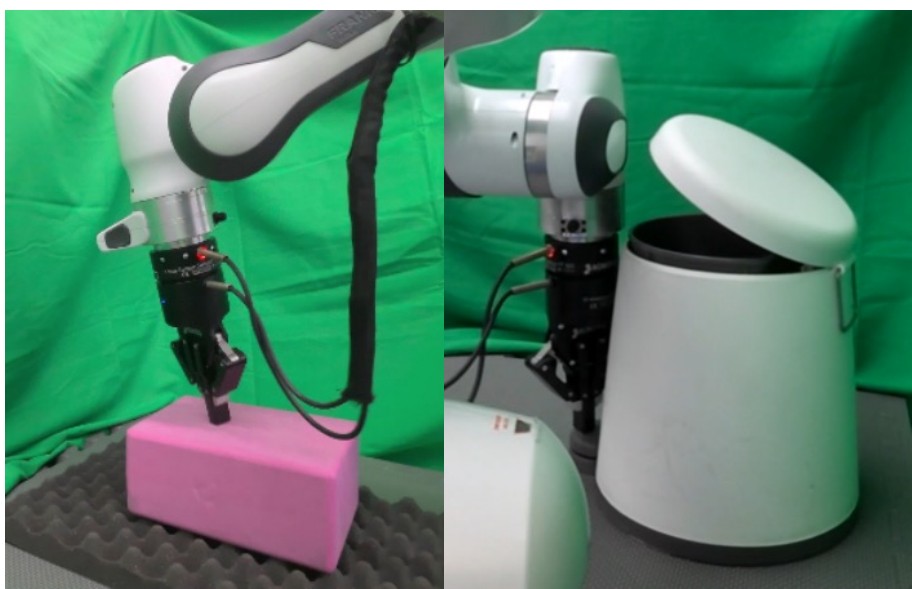

Figure 30: **Block Collision and Bin-Opening Tasks.** *Left:* The robot collides with a block while moving its end-effector downward, measuring contact forces. *Right:* Another example of applying high-frequency torque control for a soft bin-opening motion (qualitative demonstration).

**Task and Setup.** We evaluate two tasks to highlight both free-space tracking and collision handling:

1. **Sim2Real Goal Tracking:** The robot must track a randomly moving goal pose in free space. A shaped reward encourages precise positioning of the end-effector (tool center point, TCP), through a reward to minimize the L2 norm distance of the TCP to the goal location.

2. **Sim2Real Block Collision:** The same policy is tested with an enforced collision by commanding the end-effector to move downward into a block. We measure the average forces at the end-effector.

In both scenarios, the **low-level controller** operates at **200** Hz (torque control) while the **high-level policy** (PPO) runs at **10** Hz. We train each policy entirely in MuJoCo and deploy it zero-shot on the physical robot via the Franka Control Interface (FCI). The robot's built-in gravity compensation is enabled, allowing the learned controller to focus purely on torque-level corrections and collision handling. The same robot platform was also used and developed further in Zakka et al. (2025).

**Baseline Controllers.** We compare **EvoControl (Full State)** against two alternative low-level controllers, both paired with the same high-level PPO policy learning algorithm:

- *Fixed Controller - PD Position:* A hand-tuned PD loop that tracks the high-level's joint-position commands.

- *Direct Torque Control (Hi-Freq)*: An end-to-end torque policy trained purely with RL at high frequency.

While direct-torque RL can in principle match EvoControl's frequency, it often suffers from poor exploration and long-horizon instability, as shown in the main paper.

**Metrics.** We measure:

- **Avg. Tracking Error (m):** The distance between the goal pose and the end-effector over an episode.

- **Avg. Total Force (N):** The mean force magnitude recorded at the wrist sensor, capturing the compliance or stiffness of each approach.

**Results.** Table 36 presents the performance of each low-level controller across three conditions: (1) *Sim-Goal Tracking* (evaluation in simulation), (2) *Sim2Real Goal Tracking* (physical free-space tracking), and (3) *Sim2Real Block Collision* (intentional collision).

- **Accuracy vs. Compliance:** EvoControl maintains comparable or better tracking accuracy than the fixed PD controller while reducing collision forces.

- **Zero-Shot Transfer:** No additional fine-tuning was performed on real hardware; EvoControl exhibits stable behavior from the outset.

Table 36: Performance comparison of different low-level policies with a high-level PPO algorithm. EvoControl achieves superior tracking performance and lower force application compared to other methods. Direct torque control at high frequency fails to complete the task due to hitting joint limits. Bold values indicate the best performance in each metric.

| Same PPO high-level alg. $\rho$ with | Sim-Goal Tracking | Sim2Real Goal Tracking | | Sim2Real Block Push | |
|---|---|---|---|---|---|
| | $R \uparrow$ | Avg. Tracking Error (m) $\downarrow$ | Avg. Total Force (N) $\downarrow$ | Avg. Tracking Error (m) $\downarrow$ | Avg. Total Force (N) $\downarrow$ |
| *Fixed Cont. - PD Position* | 13,564 | 0.076 | 4.159 | 0.1537 | 9.056 |
| *Direct Torque Cont. High Freq.* | 10,413 | Did Not Finish (Hits joint limits) | | Did Not Finish (Hits joint limits) | |
| **EvoControl (Full State)** | 24,213 | 0.0261 | 4.331 | 0.1524 | 8.169 |

**Conclusion.** These real-robot results confirm that *EvoControl* can: Transfer directly from MuJoCo simulation to real hardware without additional tuning; Retain accuracy for free-space tracking while show initial insights into high-frequency torque controlled policies potentially having lower un-modelled interaction forces. Overall, this demonstrates EvoControl's practical viability for deployment to real robot setups to assist the future development of safe, adaptive, high-frequency torque control in real-world robotic tasks.

## L. Limitations & Future Work

While EvoControl demonstrates promising results for high-frequency continuous control, several limitations and avenues for future research warrant exploration.

- Still relies on the existence of a fixed-PD controller for the continuous-time control task. Although we demonstrate robustness to some degree of PD parameter misspecification (Table 6), the reliance on a PD controller as a starting point poses a limitation. In domains where designing a suitable PD controller is challenging or impossible (e.g., systems with non-actuated joints, highly nonlinear dynamics, or discrete action spaces), applying EvoControl in its current form may be difficult. We do perform an ablation where we show that EvoControl can still learn performant policies without the existence of a fixed-PD controller Appendix J.5, however other approaches to stabilize and initialize policy learning are promising directions for future work.

- EvoControl can require more computational complexity compared to only performing PPO, which can be readily parallelized in practice with modern accelerated compute platforms, and restricting EvoControl to use the same computational complexity, whilst still outperforming the baselines is also possible, Appendix J.4.

In addition, promising future directions include exploring more complex nested hierarchies, direct low-level to high-level information flow, and ensembles of policies.

## M. Reproducibility Statement

In the following we outline all the sections where the reader can find full information to fully reproduce all the main results. We also clearly state the following of the *assumptions* of the method in Appendix G, *experimental settings* in Appendices E to I, and the *limitations* of the work in Appendix L.

## N. Common Questions and Discussion

### N.1. Why Not Simply Pre-train with a PD Controller and Imitate?

One might consider using a PD position controller as a policy to collect rollout trajectories and then using imitation learning to train a neural network to replicate its behavior. However, this approach has limitations. The learned network would only be as capable as the PD controller itself, inheriting its limitations in expressiveness and inability to learn complex, high-frequency interaction behaviors. EvoControl, by directly optimizing the low-level policy with ES, aims to surpass the capabilities of the initial PD controller and discover more nuanced and adaptive control strategies. Furthermore, imitation learning requires a substantial amount of demonstration data, while EvoControl learns directly from the environment reward signal.

### N.2. Relationship to Pulse-Width Modulation

The benefits of high-frequency control, as highlighted by Proposition 2.1, share a conceptual similarity with Pulse-Width Modulation (PWM) in electrical engineering. In PWM, a high-frequency signal with varying pulse widths is used to effectively represent a lower-frequency analog signal. Similarly, in EvoControl, high-frequency actions generated by the low-level policy can represent and achieve the lower-frequency targets set by the high-level policy with greater precision and responsiveness compared to a fixed-frequency PD controller. While not a direct analogy, this parallel highlights the ability of high-frequency signals to enhance control and achieve desired outcomes more effectively.

### N.3. Discussion of Theoretical Analysis on Convergence of the Bi-Level Framework

While a formal proof is beyond our current scope, we have designed EvoControl to promote stable learning and convergence through the following mechanisms:

*Staged Training and Annealing*

- Alternating optimization of the high-level policy $\rho$ (using PPO) and the low-level policy $\beta$ (using evolutionary strategies) reduces non-stationarity.

- The annealing parameter $\alpha$ gradually transitions $\beta$ from a fixed PD controller to a learned neural network, maintaining stability.

*Convergence Properties of PPO and Evolutionary Strategies*

- **PPO:** Known for stable convergence due to its clipped surrogate objective (Schulman et al., 2017).

- **Evolutionary Strategies:** Robust in high-dimensional, non-convex spaces, effectively optimizing policies without gradient information (Salimans et al., 2017).

*Relation to Hierarchical Reinforcement Learning*

- Our framework aligns with HRL methods such as the Options framework (Sutton et al., 1999) and FeUdal Networks (Vezhnevets et al., 2017), which provide theoretical insights into hierarchical policies.

- Nachum et al. (2019) offer theoretical guarantees for HRL methods, supporting potential convergence in hierarchical structures.

These design choices contribute to the stability and reliability of EvoControl, aligning with established reinforcement learning principles.

### N.4. Comparison of EvoControl with Model Predictive Control (MPC)

EvoControl differs from MPC in key aspects:

*Learning-Based vs. Model-Based*

- **EvoControl:** Entirely learning-based, using neural networks trained through interaction with the environment, without explicit system models.

- **MPC:** Relies on explicit mathematical models to predict future states and optimize control actions.

*Policy Optimization vs. Online Optimization*

- **EvoControl:** Policies are optimized during training and map states to actions without online optimization during execution.

- **MPC:** Performs online optimization at each control step during execution.

*Hierarchical Structure*

- **EvoControl:** Employs a hierarchical policy with temporal abstraction, where a high-level policy guides a low-level policy at a higher frequency.

- **MPC:** Does not inherently incorporate hierarchical policies with different operating frequencies.

*Novelty and Significance*

- **Integration of PPO and Evolutionary Strategies:** Combining PPO for the high-level policy with evolutionary strategies for the low-level policy in a hierarchical setting is, to our knowledge, novel.

- **Annealing Strategy:** Transitioning from a fixed PD controller to a learned policy stabilizes training, a feature not present in MPC.

- **Empirical Performance:** Our experiments show superior performance in high-frequency control tasks compared to standard RL methods and fixed-controller baselines.

