# OpenReview forum: "EvoControl: Multi-Frequency Bi-Level Control for High-Frequency Continuous Control"
_ICML.cc/2025/Conference — ICML 2025 poster_

### Official Review · Reviewer_qwWU · 2025-03-08

**Overall Recommendation:** 2

**Summary:**

The authors propose EvoControl, a hierarchical, bi-level policy network for motor control. EvoControl separates low-level high-frequency control and high-level, low-frequency control, by training two separate policies which output actions at different frequencies. The high-level policy is trained with RL, facilitating exploration and credit assignment due to the stronger impact of action noise and the more straightforward correlation between actions and their outcomes (compared to training a single high-frequency policy). The low-level policy is instead trained with evolution strategies, an algorithm less susceptible to the credit assignment problem. Experiments show that EvoControl can achieve better reward in continuous control tasks, which benefit from higher frequency decision making. This feature is especially relevant where safety is a concern.

**Claims And Evidence:**

1) There exist CTMDPs in which higher frequency decisions lead to higher reward. This claim is clearly true, but do the environments considered for the evaluation have the characteristics requested by the constructive proof of prop. 2.1?

2) The baselines are rather ablations of EvoFormer's components. Some performance gains are completely unexpected: for example, the locomotion environments considered in this paper are practically considered "solved" with standard RL methods. Instead, EvoControl claims, for example, 5x performance improvement in Ant vs direct torque control (also, 2x in Walker2D, 4x in Hopper). These results need to be better motivated, because standard algorithms seem to use the capabilities of these robot nearly optimally. What does EvoControl discover that makes these robots run at unprecedented speeds?

**Essential References Not Discussed:**

The background and literature review sections are extensive. However, the authors should justify why they do not compare EvoControl to the cited hierarchical RL methods.

**Experimental Designs Or Analyses:**

The environments considered for evaluation (classic control environments) do not seem to be ideal to test the importance of high-frequency control, as they involve either periodic control (locomotion), balancing (pendulum) or reaching. The authors should explain why experiments on a real robot are only presented in the supplementary material.

**Methods And Evaluation Criteria:**

I am not confident about the correctness of the evaluations. The authors should provide better context, because it is counterintuitive that locomotion, pushing or balancing would benefit so much from more frequent control. The authors should verify that the high frequency control does not interfere with the physics simulator, generating unrealistic behavior. Also, results from standard algorithms should be used to put these results into perspective - the fact that they are only reported in normalized form does not allow to understand whether the baselines achieve good results. I suspect the baseline performance detailed in table 3 to be low compared to the state of the art in these environments.

**Other Comments Or Suggestions:**

No additional comments.

**Other Strengths And Weaknesses:**

The strength of the paper is combining ES with PPO to make up for the shortcomings of each algorithm. The main weaknesses of the paper are lack of comparison with other methods and lack of analysis of very surprising results.

**Questions For Authors:**

When the authors mention that they use MuJoCo Brax, what do they refer to? MuJoCo with GPU acceleration? The environments they are considering are very popular in the RL community, they should consider pretrained policies available online for comparison and verify that EvoControl really leads to an improvement in these environments. If the results stand, they should provide an analysis of the EvoControl policy.

**Relation To Broader Scientific Literature:**

The algorithm presented in this work should be compared with other forms of hierarchical control, as pairing a low-leve and a high-level controller is a common idea.

**Theoretical Claims:**

I have issues with prop. 2.1. The claim is trivial, clearly there exist processes in which taking decisions more often is an advantage. Imagine considering one environment where the reward is proportional to the number of clicks of a button, and at every decision step one can press the button or lift the finger. The authors, however, define an environment with certain requirements to write the proof, which does not seem to be related to any of the environments used in the experiments. What is the point of proving the existence of environments where higher decision frequency is beneficial, if then the requirements for such environments are not identified in any test?

---

> ### Author Rebuttal · Authors · 2025-04-01
>
> We sincerely thank the reviewer for their thoughtful and constructive feedback. Below, we address each concern:
> > do the environments considered for the evaluation have the characteristics requested by the constructive proof of prop. 2.1?
>
> Thank you; **Prop. 2.1** formally motivates that certain CTMDPs can yield higher returns by acting at higher frequencies. To concretely illustrate this, we use two safety-critical environments:
> 1. **Safety Critical Reacher (Sec. 5.2, Table 4; App. B.1)**–Here, unmodeled collision detection requires rapid reaction to avoid a penalty. Policies with slower control frequencies respond too late and accumulate significant penalties, leading to substantially lower returns.
> 2. **Safety Critical Halfcheetah** – We created a *new environment* that likewise penalizes collisions that need immediate corrective actions. We observe that policies acting faster achieve higher returns, consistent with Prop. 2.1– https://imgur.com/a/LNBypMy.
>
> > The authors should provide better context, because it is counterintuitive that locomotion, pushing or balancing would benefit so much from more frequent control....results from standard algorithms should be used to put these results into perspective
>
> **All our benchmark environments are non-standard** because we increase the control frequency to 500Hz (episode length = 1000 steps, i.e., 2 seconds of real time) and remove the typical control-cost term. Each original Gym MuJoCo environment is typically controlled at 12.5–100Hz. Therefore, our modified settings differ substantially from standard benchmarks. We made these changes to research within robotics, the question of whether we can remove the lower-level fixed controllers and instead directly control the motor torques at high frequency. What benefits (e.g., faster reactions, finer control) this higher-frequency control can enable, while revealing the challenges standard learning algorithms face in these now longer-horizon tasks. EvoControl addresses precisely these high-frequency demands.
> * **Table 3 (main paper)** reports relative policy performance at 1M high-level steps.
> * We include **explicit non-normalized results** of Table 3 in *Appendix J.7, Table 25*, alongside extended training for all baselines up to 1 billion steps in App. J.6.
>
> We now include this in Sec. 5.
> > What does EvoControl discover that makes these robots run at unprecedented speeds?
>
> In these high-frequency environments, standard policy learners struggle with exploration over long horizons (Peng, 2017)—EvoControl matches the exploration efficiency of temporally abstract goal-reaching methods (Fig. 2) and converges faster to high-reward policies **(App J.10, Figure 18)**. Rollouts (App. J.10, Fig. 18) show EvoControl often applies maximal torques to reach goals quickly while retaining precise control. This likely arises as direct high-frequency policy learning algorithms struggle with *less efficient exploration* and *slower convergence* (**Figure 2, Sec. 5.2**), and due to this, can get stuck in suboptimal policies (L 238, Left Col.) (App. J.6 & J.4), and low-frequency policies using a *lower-level goal based policies* struggle to learn accurate yet refined high-frequency motor movement, and rely on tuned fixed low-level controllers. We now include this in Sec. 5.2.
> > results from standard algorithms should be used to put these results into perspective ... consider pretrained policies
>
> Thank you; we now **include four additional standard baselines** using publicly available Brax code of Soft Actor-Critic and PPO, for both high and low frequency control. We observe that EvoControl still outperforms, https://imgur.com/a/yVFuNKk. Pretrained policies for standard MuJoCo tasks cannot be used because our tasks differ significantly (see above). Instead, we use official high-performance Brax code to ensure the baselines are well-optimized.
> >  experiments on a real robot are only presented in the supplementary material
>
> We appreciate this point and will incorporate real-robot results directly into the main paper (using our extra page) to showcase practical feasibility of EvoControl.
>
> > justify why they do not compare to ... hierarchical RL
>
> In hierarchical RL, the most relevant method is HIRO, an off-policy method that assigns state-based goals to a low-level policy, rewarded by sub-goal completion. This closely resembles our *“fixed-controllers”*, which also receive state-based goals (L192). However, as shown in Table 4, sub-goal methods cannot handle the high-frequency reactions needed in safety-critical tasks. HIRO can also be unstable; with the authors’ hyperparameters and our training budget, it failed to converge and performed near-random, highlighting the need for extensive tuning.
>
> > MuJoCo Brax
>
> Yes, MuJoCo with GPU acceleration (MuJoCo XLA (MJX)).
>
> ----
> *We hope that all of the reviewers’ concerns have been addressed and, if so, they would consider updating their score. We’d be happy to engage in further discussions.*

---

### Official Review · Reviewer_J6Rg · 2025-03-13

**Overall Recommendation:** 4

**Summary:**

The paper learns high-frequency control with a two-level structure. A high-level policy working at a lower frequency is trained with PPO. A low-level high-frequency policy is obtained with evolutionary algorithms. The two-level design works better than directly training a single high-frequency control policy with RL. A learned low-level policy achieves better results than using a fixed PD controller and is robust to PD parameters. Notably, the proposed method is better in tasks that require adaptive reactions to the environment.

**Claims And Evidence:**

The claim that using two-level control policies can both handle long horizons and achieve fine-grained control performance is reasonable and is supported by clear evidence.

**Essential References Not Discussed:**

I did not notice any missing essential related works.

**Experimental Designs Or Analyses:**

1. For Table 3, it is better to report the raw returns without normalization, so that readers can better compare the results with other methods on the benchmark.
2. The experiments on the safety critical reacher task are appreciated since the task requires high-frequency close-loop control to get good performance. However, the task has only one degree of freedom to control. I think experiments on similar safety-critical tasks with more complex dynamics would be more valuable for practical use.

**Methods And Evaluation Criteria:**

The proposed method generally makes sense, but it is not clear whether it directly suits control applications. Specifically, what is the inference speed of the low-level policy? Can it be deployed at its desired frequency in the real world?

**Other Comments Or Suggestions:**

In line 83, CTMDP occurs without explanation.

**Other Strengths And Weaknesses:**

None.

**Questions For Authors:**

None.

**Relation To Broader Scientific Literature:**

The key contributions of the paper are mostly related to RL techniques for long horizon problems, such as hierarchical RL.

**Theoretical Claims:**

I did not check the detailed proofs, but the claim seems correct from the proof sketch.

---

> ### Author Rebuttal · Authors · 2025-04-01
>
> We sincerely thank the reviewer for their thoughtful and constructive feedback. Below, we address each concern:
>
> > not clear whether it directly suits control applications. Specifically, what is the inference speed of the low-level policy? Can it be deployed at its desired frequency in the real world?
>
> Thank you for highlighting this point. We confirm that EvoControl can indeed be deployed at its desired frequency in real-world settings. In **Appendix J.14 (p. 51)**, we provide a **real-robot validation** using a 7-DoF Franka Emika Panda on a tabletop manipulation task, where the high-level policy operated at 10Hz and the low-level policy at 200Hz in a **zero-shot sim-to-real transfer**, https://imgur.com/a/Eq2Dewy, https://imgur.com/a/9yUONbW.
>
> We measured the low-level network’s inference time to be **64 microseconds ($\mu$s)** on average (over 1,000 samples), corresponding to a potential 15 kHz operating frequency—well beyond typical robotic control rates. Our real-world experiments yielded three main findings:
> 1. *Stable and Fast:* The low-level network infers rapidly (64 µs per step), comfortably enabling deployment at 200 Hz or higher. In practice, the main bottleneck would typically be observation latency (on the order of a few milliseconds). Thus, the low-level controller itself is not the limiting factor for overall reactivity.
> 2. *Better Collision Handling:* Compared to a tuned PD controller, EvoControl produced *lower contact forces*, highlighting the benefits of learned high-frequency torque control for safety.
> 3. *No Fine-Tuning Required:* The policies trained in *MuJoCo (XLA)* transferred directly to the real robot without additional adjustments.
>
> These results strongly support EvoControl’s practicality in real-world environments, clearly demonstrating that the high-frequency policy can indeed be deployed at the desired rate despite practical observation latency constraints.
>
>
> > For Table 3, it is better to report the raw returns without normalization, so that readers can better compare the results with other methods on the benchmark.
>
> Thank you; that is a great suggestion. We already include the raw returns without normalization for Table 3 in Appendix J.7, Table 25.
>
> Allow us to kindly clarify that **all our benchmark environments** are non-standard because we increase the control frequency to 500Hz (episode length = 1000 steps, i.e., 2 seconds of real time) and remove the typical control-cost term (Appendix E). Each original Gym MuJoCo environment is typically controlled at 12.5–100Hz. Therefore, our modified settings differ substantially from standard benchmarks. We made these changes to research within robotics, the question of *whether* we can remove the lower-level fixed controllers and instead directly control the motor torques at high frequency. What benefits (e.g., faster reactions, finer control) this higher-frequency control can enable, while revealing the challenges standard learning algorithms face in these now longer-horizon tasks. EvoControl addresses precisely these high-frequency demands.
>
> >  I think experiments on similar safety-critical tasks with more complex dynamics would be more valuable for practical use.
>
> Thank you for the helpful suggestion. In response, we introduce a **new “Safety Critical Halfcheetah” environment** with six degrees of freedom and an observation dimension of 19, exhibiting significantly more complex dynamics than our previous 1D safety task. Specifically, we add a blocking wall in 25% of the episodes, with collision force only observable upon contact. This wall blocks the cheetah’s forward path and incurs a penalty for any impact. As a result, the policy must quickly retreat if it detects a collision; otherwise, it continues forward. This design underscores the importance of higher-frequency actions for fast responses in unmodeled safety-critical situations, following a similar setup to "Safety Critical Reacher". Our full results (at https://imgur.com/a/LNBypMy) show that *EvoControl* still outperforms the baselines in this more challenging safety-critical scenario.
>
> > In line 83, CTMDP occurs without explanation.
>
> Thank you for this typo; line 83 now reads "continuous-time Markov decision processes (CTMDPs)".
>
> ---
>
> *We hope that all of the reviewers’ concerns have been addressed and, if so, they would consider updating their score. If any issues remain, we would be glad to discuss them further, especially in light of your current evaluation.*

---

> > ### Comment · Reviewer_J6Rg · 2025-04-02
> >
> > Thank you for the answers. I appreciate the real robot deployment results and the added "Safety Critical Halfcheetah" experiments. As my concerns are resolved, I would raise the score accordingly.

---

> > > ### Author Response · Authors · 2025-04-03
> > >
> > > Thank you for your positive feedback and for reevaluating our work so favorably. We are glad that the real robot deployment results and the new “Safety Critical Halfcheetah” experiments have addressed your concerns. Your input has been very valuable in strengthening our paper, and we appreciate the time and thought you invested in your review.

---

### Official Review · Reviewer_c2Dp · 2025-03-13

**Overall Recommendation:** 4

**Summary:**

The manuscript presents a novel bi-level optimization reinforcement learning method for high frequency control. The method combines a high-level low frequency policy with a low-level high frequency controller. Both controllers are learnable. The authors motivate the bi-level learning/optimization scheme. Overall, the manuscript, the experimental validation and result analysis are high quality. The results are also very encouraging. Finally, the authors provide motivation (and analytical proof; although trivial imho) for why a higher frequency controller is always more desirable if it can be learned effectively.

**Claims And Evidence:**

The authors back up their claims using both theoretical (although a bit weak) and empirical (quite strong) analysis.

**Essential References Not Discussed:**

I do not think that the authors missed any important reference.

**Experimental Designs Or Analyses:**

As I said before, the authors have really thought out the experimental analysis. The authors have included all important baselines that I could think of, and the experimental analysis is thorough.

**Methods And Evaluation Criteria:**

The evaluation pipeline makes sense and the criteria is rather rigorous. I appreciate the 128 rollouts for the metrics per policy (and 3 full training runs). Overall, every question that I had while reading the manuscript was answered later in the text.

**Other Comments Or Suggestions:**

I have no other comments.

**Other Strengths And Weaknesses:**

The proposed method is not very novel (I would say the main novelty is incremental), but the presentation, experimental results and analysis are of top quality.

**Questions For Authors:**

I honesty do not have any question left. Every question that arised while reading the manuscript, the authors replied to it later in the text.

**Relation To Broader Scientific Literature:**

Continuous control lies in the heart of many real-world systems, and having more accurate, robust and highly adaptive policies can be of great importance towards the widespread adoption of robots, and RL agents in general.

**Theoretical Claims:**

This might be the weakest point of the manuscript. Although I do not see anything wrong with the proof, it seems rather trivial for the specific case of MDPs that the authors assume. Also, the authors do not really motivate why such MDPs exist, and why they are important. Overall, the proof/claim here seems superficial. I would, personally, be happy with the paper without the claim/proof. Although, nothing wrong with keeping it.

---

> ### Author Rebuttal · Authors · 2025-04-01
>
> We thank the reviewer for their thoughtful and constructive feedback. Below, we address each concern:
> > authors do not really motivate why such MDPs exist, and why they are important
>
> ### (A) Why Such MDPs Exist
>
> Our Proposition 2.1 is meant to show that there are environments/continuous-time MDPs in which acting at higher frequency can strictly yield higher returns. While this might be theoretically straightforward, it underscores the practical reality that in certain tasks, reacting faster confers a distinct advantage. Indeed, when new or unmodeled information arrives between coarser time steps, lower-frequency controllers (or those that do not adapt quickly) risk missing events that incur large penalties or lose large rewards.
>
> Concretely, we focus on safety-critical tasks where immediate reaction to collisions or unexpected contacts is crucial. In these tasks, an undetected collision or a delayed response can rapidly cause catastrophic damage. The “Safety Critical Reacher” environment (Section 6.2 of our paper) exemplifies this scenario: when there is a sudden collision, the policy must quickly backtrack, or else it will suffer a considerable penalty for the extra collision force. A lower-frequency controller often cannot respond fast enough within that sub-interval.
>
> To further verify the existence of such MDPs, we introduce a **new “Safety Critical Halfcheetah” environment** with six degrees of freedom and an observation dimension of 19, exhibiting significantly more complex dynamics than our previous 1D safety task. Specifically, we add a blocking wall in 25% of the episodes, with collision force only observable upon contact. This wall blocks the cheetah’s forward path and incurs a penalty for any impact. As a result, the policy must quickly retreat if it detects a collision; otherwise, it continues forward. This design underscores the importance of higher-frequency actions for fast responses in unmodeled safety-critical situations, following a similar setup to "Safety Critical Reacher". Our full results (at https://imgur.com/a/LNBypMy) show that *EvoControl* still outperforms the baselines in this more challenging safety-critical scenario.
>
> **Table 33, Safety Critical Halfcheetah Results**
> |Same PPO high-level alg. ($\rho$) with|Safety Critical Halfcheetah ($\mathcal{R}$) ↑)|
> |---------------------------------------------|----------------------------------------------|
> |*Fixed Cont.* – PD Position|84.7±1.68|
> |*Fixed Cont.* – PD Position Delta|0±0|
> |*Fixed Cont.* – Random|0.0±0.0|
> |*Direct Torque Cont.* – High Freq. (500Hz)|100±1.42|
> |*Direct Torque Cont.* – Low Freq. (31.25Hz)|0±0|
> |**EvoControl (Full State)**|**120±13.9**|
> |**EvoControl (Residual State)**|**105±5.84**|
> |**EvoControl (Target + Proprio.)**|**117±34.9**|
> |**EvoControl (Target)**|**117±5.33**|
>
> ### (B) Why These Scenarios Are Important
>
> These environments mirror real robotic applications, such as:
> * *High-gain robotic arms near humans:* Where collisions must be minimized or mitigated the instant they occur, e.g., in collaborative assembly lines.
> * *Aerial vehicles under sudden gusts:* Hovering drones or helicopters must correct torque and thrust when wind forces change abruptly.
> * *Surgical robotics:* In certain procedures, micrometer-level slip or unintentional contact can cause tissue damage; responding at tens-of-milliseconds scale is vital.
>
> In all these cases, the penalty for a delayed response can be high; hence, faster control loops can yield higher returns.
>
> Furthermore, we validated the real-world feasibility of EvoControl on a 7-DoF Franka Emika Panda (Appendix J.14, p. 51), initially demonstrating superior collision handling and reduced contact forces compared to a tuned PD controller. Importantly, the policies trained in MuJoCo XLA transferred directly to the robot without requiring additional task-specific tuning, unlike fixed-PD controllers. These results underscore the practicality of using a learned fast controller in real-world scenarios.
> > proof/claim here seems superficial ... nothing wrong with keeping it.
>
> While the proof (Proposition 2.1) is simple, we keep it to clarify that high-frequency control is not just an intuitive guess but can be rigorously shown beneficial under specific conditions.
>
> > not very novel (I would say the main novelty is incremental)
> Though high-/low-frequency control is common, our approach:
> 1. Learns a neural fast controller (vs. fixed PD/sub-goal tracking only policies).
> 2. Combines on-policy PPO for the high-level and ES for the low-level, learning jointly, avoiding instability from direct high-frequency RL.
> 3. Demonstrates consistent real-world performance with minimal tuning.
>
> This synergy extends hierarchical RL to truly fast torque control while retaining exploration benefits from slower layers.
>
> ---
>
> *We hope that all of the reviewers’ concerns have been addressed. We’d be happy to engage in further discussions.*

---

> > ### Comment · Reviewer_c2Dp · 2025-04-07
> >
> > I thank the reviewers for the detailed rebuttal. I am happy with the response, and I will keep my score. I still believe this is incremental work, but the authors did an amazing job in providing detailed analysis and results which imho produces added value.

---

> > > ### Author Response · Authors · 2025-04-08
> > >
> > > Thank you once again for your positive feedback and for evaluating our work so favorably. We are delighted that you appreciated our *“amazing job in providing detailed analysis and results which … produces added value,”* as well as our focus on achieving more accurate, robust, and highly adaptive policies. Your perspective on the incremental nature of our contribution is duly noted, and we have done our best to clarify the unique aspects of our approach in the latest revision. Our key novelty is the *joint learning* of a slow (e.g., 30Hz) high-level policy (via PPO) and a fast (e.g., 500Hz) low-level proprioceptive controller (via ES), thus avoiding the instability common to direct high-frequency RL. Unlike standard sub-goal methods, our low-level controller optimizes the overall long-horizon episodic return. Moreover, EvoControl has been validated on a real robot with minimal tuning, enabling zero-shot transfer. We sincerely appreciate the time and thought you invested in reviewing our paper, and we are pleased that our work resonates with your vision of accelerating real-world adoption of robots and RL agents.

---

### Official Review · Reviewer_kbC8 · 2025-03-19

**Overall Recommendation:** 4

**Summary:**

The paper introduces a bi-level policy and training method for high-frequency and continuous-time control. The bi-level policy consists of a high-level policy that operates at a low frequency and issues a latent action. The low-level policy decides the final action based on the environment state and the latent action. For training stability, the low-level policy is a convex combination of a ​PD controller and a neural network where the combination gradually shifts towards the NN policy during the training. The high-level policy is trained with PPO and the low-level policy is trained with an evolution algorithm. The paper shows this approach can outperform both direct torque learning and fixed PD controller approaches and introduce exploration and robustness advantages over them.

**Claims And Evidence:**

All claims and information are very well supported by either relevant prior work or independently demonstrated ablation experiments. For instance,​ the challenges of policy gradient that are mentioned to motivate the evolutionary algorithm in Section 3.3 are verified empirically.

**Essential References Not Discussed:**

I am not aware of any critical​ references that are missed.

**Experimental Designs Or Analyses:**

The experiments and their analysis are mostly standard. The main concern I have is the choice to fix the number of high-level policy training steps in the comparisons instead of fixing the total environment interaction time (Line 295right). This may be specifically disadvantageous to direct torque high frequency. For this algorithm, the high-level policy takes orders of magnitudes less time to interact with the environment as each interaction is very short. I think fixing the total time and not the number of steps is more fair. If possible, including the results for this will be beneficial.

**Methods And Evaluation Criteria:**

The methods of the paper are in the ​vicinity of the ideas in the field and highly appropriate. The evaluations use both the standard benchmarks and customized tasks that further highlight the advantages of the algorithm.

**Other Comments Or Suggestions:**

Some typos:

1. line 093 right, the definition of R, subscripts are wrongly formatted.

**Other Strengths And Weaknesses:**

The paper does a tremendous job in supporting the findings with ablation experiments and evaluations. The paper is well written and the idea is novel and interesting. Please see questions and the comment for experimental design for potential weaknesses.

**Questions For Authors:**

1. Are there any experiments to show the proposed method has advantages over hierarchical RL methods with goal-reaching low-level policies, such as HIRO. I saw the experiments comparing with having fi​xed PD controllers as the low-level policy, but no comparisons to HIRO. Some comments or empirical results would be appreciated.

2. Have you tried the techniques that enable stable training of other hierarchical RL algorithms such as option-critic and HIRO to stabilize the training of EvoControl instead of the annealing? For example, different learning rates for the two levels to induce almost fixed behavior of one level when the other one is adjusting to its changes. I wonder why RL has failed to train the low-level policy here (Appendix J2) as opposed to those methods.

**Relation To Broader Scientific Literature:**

I am aware that there are multi-level hierarchical RL algorithms for control tasks. This paper's method is highly related to those approaches. The main new ability of the new approach is that the low-level policy is not limited to a fixed behavior as opposed to the previous methods where the low-level policy was trained to reach a goal state. This gives the agent a higher representation ability for more complex tasks.

**Theoretical Claims:**

The paper does not have much theory. The theoretical result seems correct from the proof sketch. I have not checked​ the details of the proof.

---

> ### Author Rebuttal · Authors · 2025-04-01
>
> We thank the reviewer for their thoughtful and constructive feedback, particularly their appreciation that all the claims are well supported and the paper does a tremendous job supporting the findings, with the idea being novel and interesting. Below, we address each concern:
> > fixing the total time and not the number of steps is more fair. If possible, including the results for this will be beneficial.
>
> This is an excellent point, we agree. We already provide results for all baselines in **Appenidx J.4** (also shown here: https://imgur.com/a/CBzbEBO) where we fix the total time—here implemented by fixing the total number of low-level policy steps, which for all our high-frequency environments, that take actions at 500Hz, is a duration of 2 seconds. This ensures that the direct torque controller is evaluated in an identical physical time window to EvoControl. These new results still show EvoControl outperforming baselines, including the high-frequency direct-torque policy, which continues to struggle with long-horizon credit assignment at 500 Hz.
>
> We will highlight these findings more prominently in the main paper’s Table 3 (as suggested) for camera-ready. This way, readers see both “fixed high-level steps” and “fixed total environment time” versions of the experiments.
> > Typos
>
> Thank you, we have now fixed these.
> > Are there any experiments to show the proposed method has advantages over hierarchical RL methods with goal-reaching low-level policies, such as HIRO.
>
> Yes, as HIRO closely resembles our *"fixed-controllers"* methods class (L192), where the low-level policies perform sub-goal state completion. Let us clarify that HIRO is an off-policy method that assigns state-based goals to a low-level policy, rewarded by sub-goal completion. However, as shown in Table 4, sub-goal attainment methods cannot handle the high-frequency reactions needed in safety-critical tasks.
>
> We did use an existing implementation of HIRO (from the authors) on our high-frequency environments, however, we found that it was too unstable, and did not converge to a meaningful policy (showing random performance). We likely believe that HIRO, being an off-policy method, may require extensive hyper-parameter tuning for each environment and task to work correctly, and the authors within the HIRO paper discuss the instability of the method during learning.
>
> Similarly, the Option-Critic framework uses a finite set of discrete options, each with learnable termination conditions. Because our tasks feature a continuous latent action (e.g., specifying a target joint position or velocity) and because we run on-policy PPO for the high-level layer, we found it non-trivial to combine discrete option-termination logic with continuous-time feedback.
> > Have you tried the techniques that enable stable training of other hierarchical RL algorithms such as option-critic and HIRO to stabilize the training of EvoControl instead of the annealing?
>
> Thank you for the insightful question. The techniques used to stabilize other hierarchical RL algorithms do not directly apply to EvoControl as:
> * HIRO is an off-policy method that relies on importance sampling and sub-goal relabeling of previously collected trajectories. By contrast, EvoControl’s high-level policy uses on-policy PPO, which discards old data once the policy updates. Hence, off-policy relabeling and importance sampling are not feasible in EvoControl.
> * Option-critic typically defines a finite set of discrete options, each with its own policy and termination condition. Its stability arises from learning when to switch or terminate these options. In EvoControl, the lower-level controller is a single, continuous neural policy (e.g., torque commands at 500 Hz)—there are no discrete “skills” to terminate, and the entire low-level training is through ES on raw episodic return.
>
> Consequently, we rely on annealing from a PD controller to the learned neural policy, which stabilizes our on-policy training and avoids the need for subgoal relabeling or discrete option terminations.
> > For example, different learning rates for the two levels to induce almost fixed behavior of one level when the other one is adjusting to its changes. I wonder why RL has failed to train the low-level policy here (Appendix J2) as opposed to those methods.
>
> That is a nice idea. We empirically tested a similar concept, where, when using RL to train the low-level policy, we take interleaved updates every so many steps, and also did the same with EvoControl. Interestingly taking frequent interleaved updates causes instability, and taking interleaved updates slightly less frequently improves learning (e.g., take an update of the higher-level policy for every 16 low-level policy updates), this could arise due to an update on one level has a larger effect on the overall policy than an update on the other layer.
>
> ---
> *We hope that all of the reviewers’ concerns have been addressed. We’d be happy to engage in further discussions.*

---

### Decision · Program_Chairs · 2025-05-01

**Decision:**

Accept (poster)

**Comment:**

This paper introduces EvoControl, a novel bi-level policy designed for high-frequency continuous control. The method utilizes a low-frequency high-level policy, trained with PPO, to generate latent actions. Subsequently, a high-frequency low-level policy, initially a convex combination of a PD controller and a neural network trained via an evolutionary algorithm, determines the final control action. This hierarchical approach demonstrates superior performance compared to direct learning methods and fixed PD controllers. Overall, the reviewers found the method interesting, although one questioned the novelty of the proposed bi-level policy. The authors generally addressed the reviewers' comments satisfactorily during the rebuttal, and three out of four reviewers, along with myself, believe the overall approach, experiments, and writing quality make the paper acceptable for the conference. While one reviewer (qwWU) raised concerns regarding the experiments and theoretical aspects that were not fully resolved by the authors' response, the majority opinion supports acceptance.